# Coupling of cell shape, matrix and tissue dynamics ensures embryonic patterning robustness

Prachiti Moghe ®[1,2,3], Roman Belousov ®[4], Takafumi Ichikawa ®[5,6], Chizuru Iwatani[7], Tomoyuki Tsukiyama ®[5,7], Anna Erzberger ®[4] ✉ & Takashi Hiiragi ®[1,5,6] ✉

Tissue patterning coordinates morphogenesis, cell dynamics and fate specification. Understanding how precision in patterning is robustly achieved despite inherent developmental variability during mammalian embryogenesis remains a challenge. Here, based on cell dynamics quantification and simulation, we show how salt-and-pepper epiblast and primitive endoderm (PrE) cells pattern the inner cell mass of mouse blastocysts. Coupling cell fate and dynamics, PrE cells form apical polarity-dependent actin protrusions required for RAC1-dependent migration towards the surface of the fluid cavity, where PrE cells are trapped due to decreased tension. Concomitantly, PrE cells deposit an extracellular matrix gradient, presumably breaking the tissue-level symmetry and collectively guiding their own migration. Tissue size perturbations of mouse embryos and their comparison with monkey and human blastocysts further demonstrate that the fixed proportion of PrE/epiblast cells is optimal with respect to embryo size and tissue geometry and, despite variability, ensures patterning robustness during early mammalian development.

Tissue patterning in developing embryos depends on the coordination between cellular dynamics and fate specification. While specification of cell fate is typically accomplished through gene regulatory networks activated by secreted morphogens and signalling, cellular behaviours including division, migration and sorting drive morphogenesis to form spatially organized tissues. In developing tissues, equipotent cells start differentiating to form a heterogeneous mixture of cell populations. Through cell rearrangements and sorting, tissues refine patterns and generate sharp boundaries, such as in the neural tube[1], rhombomeres[2], somites[3,4] and glandular and sensory epithelia[5,6]. In these scenarios, local cell rearrangement and cell fate changes can achieve pattern precision.

However, how fate specification, cell dynamics and their inherent variability adapt to the growing tissue size and geometry to achieve robust patterning remains less understood. For example, the size of early mouse embryos varies up to fourfold among in utero embryos, and experimentally manipulated double- or half-size embryos develop to term[7,8], though how they achieve pattern precision remains unclear. While in many organisms, morphogen gradients extend across tissues and determine the orientation and length scale of tissue patterns, early mammalian embryos lack such gradients or other forms of pre-patterning[9–11].

Morphogenesis and patterning of the mouse embryo starts with the formation of a blastocyst, which comprises three distinct cell

[1]Hubrecht Institute, Royal Netherlands Academy of Arts and Sciences (KNAW), Utrecht, Netherlands. [2]Developmental Biology Unit, European Molecular Biology Laboratory, Heidelberg, Germany. [3]Collaboration for joint PhD degree between EMBL and Heidelberg University, Faculty of Biosciences, Heidelberg, Germany. [4]Cell Biology and Biophysics Unit, European Molecular Biology Laboratory, Heidelberg, Germany. [5]Institute for the Advanced Study of Human Biology (WPI-ASHBi), Kyoto University, Kyoto, Japan. [6]Department of Developmental Biology, Graduate School of Medicine, Kyoto University, Kyoto, Japan. [7]Research Center for Animal Life Science, Shiga University of Medical Science, Shiga, Japan. ✉e-mail: anna.erzberger@embl.de; t.hiiragi@hubrecht.eu

lineages and a fluid cavity. The trophectoderm forms the outermost layer of epithelial cells enclosing the pluripotent inner cell mass (ICM)[11]. ICM cells, initially equivalent and expressing lineage marker genes heterogeneously, progressively differentiate into the innermost embryonic epiblast (EPI) and the cavity-facing, extra-embryonic primitive endoderm (PrE) with the characteristic salt-and-pepper distribution of cell fates[9,12–14]. These cell fates are specified by a gene regulatory network involving lineage-specific transcription factors NANOG and GATA6 and fibroblast growth factor (FGF) signalling[15–19]. As multiple fluid-filled cavities emerge, expand, coalesce and collapse in the blastocyst, cell sorting within the heterogeneous ICM segregates PrE cells to a monolayer at the cavity surface enveloping the epiblast[14,20–25], followed by PrE maturing into a polarized epithelium to pattern the ICM[24,26]. Dynamic mechanisms such as directional cell movement, cell surface fluctuations, oriented divisions, apoptosis and positional induction were proposed to drive EPI:PrE segregation[14,22,27–29], although how these properties arise among ICM cells, coupling among these processes and their coordination with cell fate specification and blastocyst morphogenesis are poorly understood.

Thus far, cell fate specification and spatial segregation have been studied independently and an integrative view of ICM patterning is lacking. Specifically, it remains unclear whether EPI and PrE cells exhibit distinct movements in the ICM, if so, what drives them, and how robust patterning is ensured in mouse embryos despite inevitable spatiotemporal developmental variabilities, particularly in embryo size and geometry[13,14,30,31]. This is largely due to the technical challenge of analysing cellular dynamics in the presence of the expanding and collapsing fluid cavity. Here, we systematically analyse cellular dynamics, cell position, polarity and fate to gain mechanistic insights into EPI/PrE segregation and patterning within the mouse ICM. In particular, we investigate the role of the extracellular matrix (ECM) in guiding PrE cell migration and the mechanism that ensures patterning robustness in mammalian blastocysts.

## Results

### Distinct EPI and PrE cell movements underlie segregation in the ICM

As the expansion and collapse of the blastocyst cavity change embryo shape and make it challenging to track and analyse cellular dynamics, we isolated ICMs from whole blastocysts via immunosurgery[32] (Fig. 1a). This experimental system eliminates the abrupt change in overall embryo shape and effectively reduces the complexity of cellular dynamics from a three-dimensional and heterogenous geometry to a system near spherical symmetry where dynamics can be analysed in one radial dimension. In agreement with previous studies[27,29], we verified that the in vitro culture of isolated ICMs faithfully recapitulates the EPI/PrE sorting in the blastocyst in terms of cell number and timing (Fig. 1a,b). The total cell number in the ICM was unchanged after immunosurgery (Extended Data Fig. 1a) and those in the ICMs isolated at E3.5

and E4.5, or after 24-h culture from E3.5, were comparable with ICM cell numbers in E3.5 and E4.5 whole blastocysts, respectively (Fig. 1b). We live-imaged the isolated ICMs using a fluorescent reporter of PrE fate, *Pdgfra*[H2B-GFP] (refs. 14,33) combined with a ubiquitous *H2B-mCherry* reporter[34] (Fig. 1c and Supplementary Video 1) and quantitatively analysed the dynamics of cell sorting, using a custom, semi-automated nuclear detection and tracking pipeline[31] (Fig. 1d). To quantify ICM segregation, we define the sorting score as the extent of overlap between EPI and PrE spatial domains (Extended Data Fig. 1b), which describes both live and immunostained ICMs (Fig. 1e and Extended Data Fig. 1c).

With these tools established, we analysed the comprehensive cell-tracking dataset to examine whether cells exhibit preferential directionality of movement along the ICM radial axis (Fig. 1f–h). Notable differences are first that EPI cells initially on the ICM surface rapidly move inward in the early stages of sorting, whereas surface PrE cells do not show such directed movement, and second, that inside PrE cells show more outward movement than inside EPI cells (Fig. 1g,h). We first investigated cellular dynamics at the ICM–fluid interface.

### Apical domain decreases surface tension to position PrE cells at the fluid interface

To clearly visualize cell shape dynamics, we generated fluorescence-chimeric ICMs by tamoxifen-induced Cre-mediated recombination of the mTmG transgene[35,36] (Fig. 2a). Live-imaging showed that certain cells changed shape and flattened upon reaching the surface (Supplementary Video 2). To examine whether these cells are PrE, and whether EPI and PrE cells show distinct surface behaviour, we analysed cell shape in immunostained ICMs at the E3.5 stage. PrE cells located at the ICM surface were more likely to have stretched cell shapes (Fig. 2b), in contrast to the more rounded EPI cells (Extended Data Fig. 2a), indicating that it is indeed PrE cells that flatten when reaching the fluid interface (Extended Data Fig. 2b).

To characterize the underlying mechanics generating this PrE cell shape change and difference from EPI cells, we immunostained ICMs for actomyosin cytoskeletal elements. Among surface cells, the EPI cell cortex clearly showed higher accumulation of biphosphorylated myosin regulatory light chain (ppMRLC) and actin (Fig. 2c), suggesting higher actomyosin activity in EPI cells. Direct measurement of surface tension at the ICM–fluid interface by micropipette aspiration showed that cell–fluid interfacial tension negatively correlates with *Pdgfra*[H2B-GFP] intensity (Fig. 2d), indicating that EPI cells have higher interfacial tension than PrE cells at the ICM–fluid interface. Furthermore, immunostaining of aPKC isoforms showed their localization on the contact-free surface of PrE cells but not of EPI cells (Extended Data Fig. 2c,d). These findings suggest that the cell–fluid interfacial tension is reduced at the apical domain in PrE cells, similar to the role of apical polarity in trophectoderm cells of the 16-cell stage embryo[37], thereby enabling retention of PrE cells at the ICM surface.

---

**Fig. 1 | Differential cell movements between epiblast and primitive endoderm contribute to fate segregation in the ICM. a**, Schematic and immunostaining images of blastocysts and ICMs at E3.5 and E4.5 stages. **b**, Quantification of total cell number in the ICM from blastocysts and isolated ICMs at stage E3.5, blastocysts and isolated ICMs at stage E4.5 and isolated ICMs cultured in vitro for 24 h from stage E3.5 to E4.5. *n* = 33, 30, 40, 21 and 31 embryos for the different groups, respectively. Independent-samples *t*-test between E3.5 blastocysts and E3.5 ICMs; *P* = 0.106. One-way analysis of variance (ANOVA) between E4.5 blastocysts, E4.5 ICMs, and E3.5 ICMs + 24 h; *P* = 0.145. **c**, Time-lapse imaging of a representative ICM isolated from an E3.5 blastocyst expressing PrE-specific H2B-GFP (*Pdgfra*[H2B-GFP]) and ubiquitous H2B-mCherry (*R26-H2B-mCherry*). *n* = 8 datasets from three independent experiments. Time is indicated in h:min. *t* = 00:00, stage E3.5 + 3 h, following completion of immunosurgery. **d**, Schematic representation of single-cell tracking of EPI and PrE cells from isolated ICMs from **c**. Line plots indicating radial distances of all cells from one representative ICM until the E4.0 stage. The colour of the line indicates cell fate: PrE, green; EPI,

magenta. Shaded regions show spatial dispersion as mean ± s.d. of cell position along ICM radial axis. The geometric centroid of the ICM is considered as *d* = 0.0 and ICM outer surface is considered as *d* = 1.0 to normalize the cell position across samples. Time-series curves for individual cell positions were smoothed using a rolling average. **e**, Quantification of sorting score for isolated ICMs between stage E3.5 and E4.0. Data from *n* = 8 ICMs. For estimation of sorting score, see Methods. **f**, Plots for radial cell position from tracking of PrE (top) and EPI (bottom) cell movements in isolated ICMs. Time-series curves for individual cell positions were smoothed using a rolling average. Cell-tracking data from *n* = 158 PrE cells and *n* = 131 EPI cells from 8 ICMs. **g**, Schematic for analysis of PrE (top) and EPI (bottom) cell movements. Cell displacement along the radial axis is classified as inward or outward movement. **h**, Polar plots indicating preferential direction of cell movements among PrE and EPI. Measurements are binned according to radial cell position and time. The mean displacement of each interval is plotted, colour indicates magnitude and direction of movement. Scale bars, 20 μm. NS, not significant.

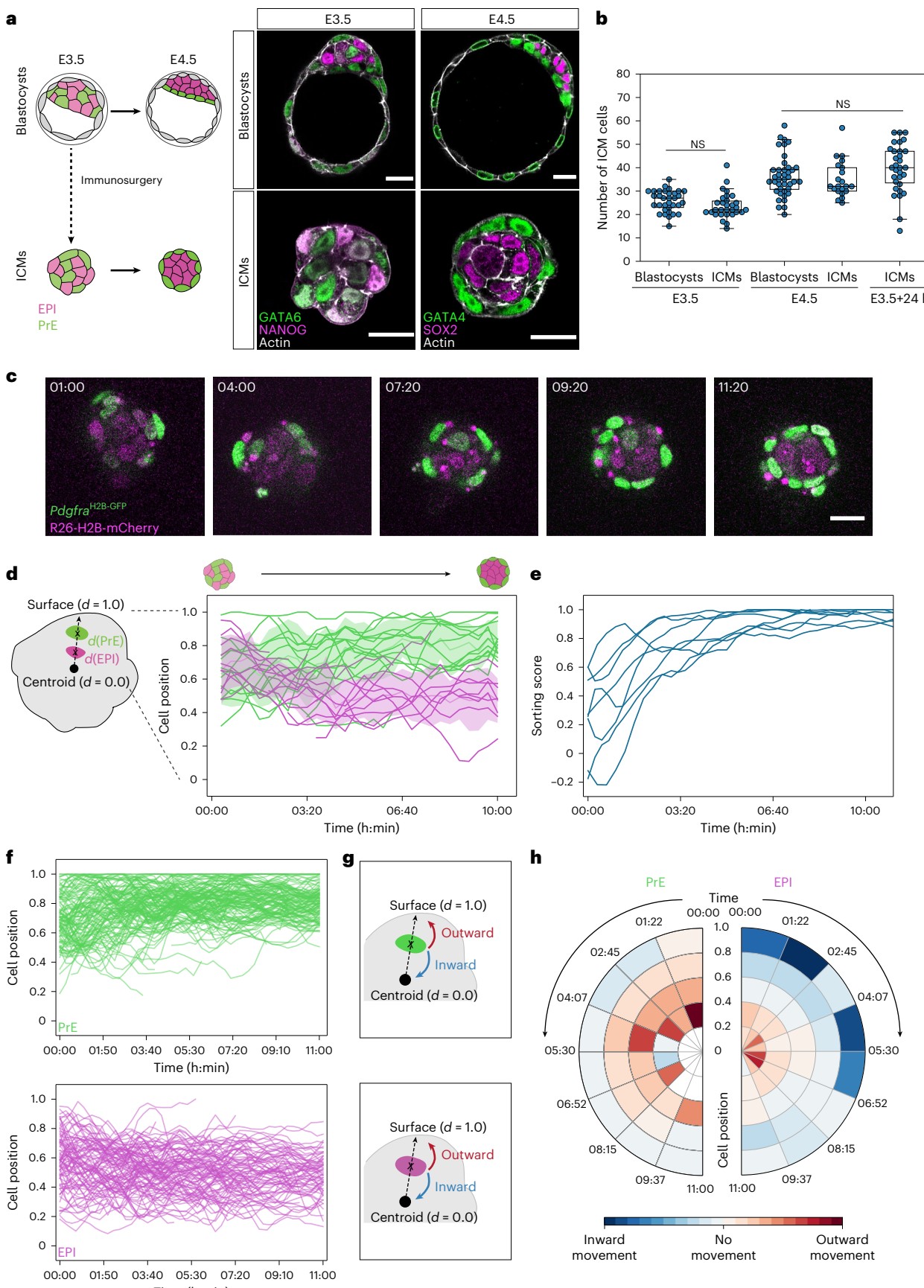

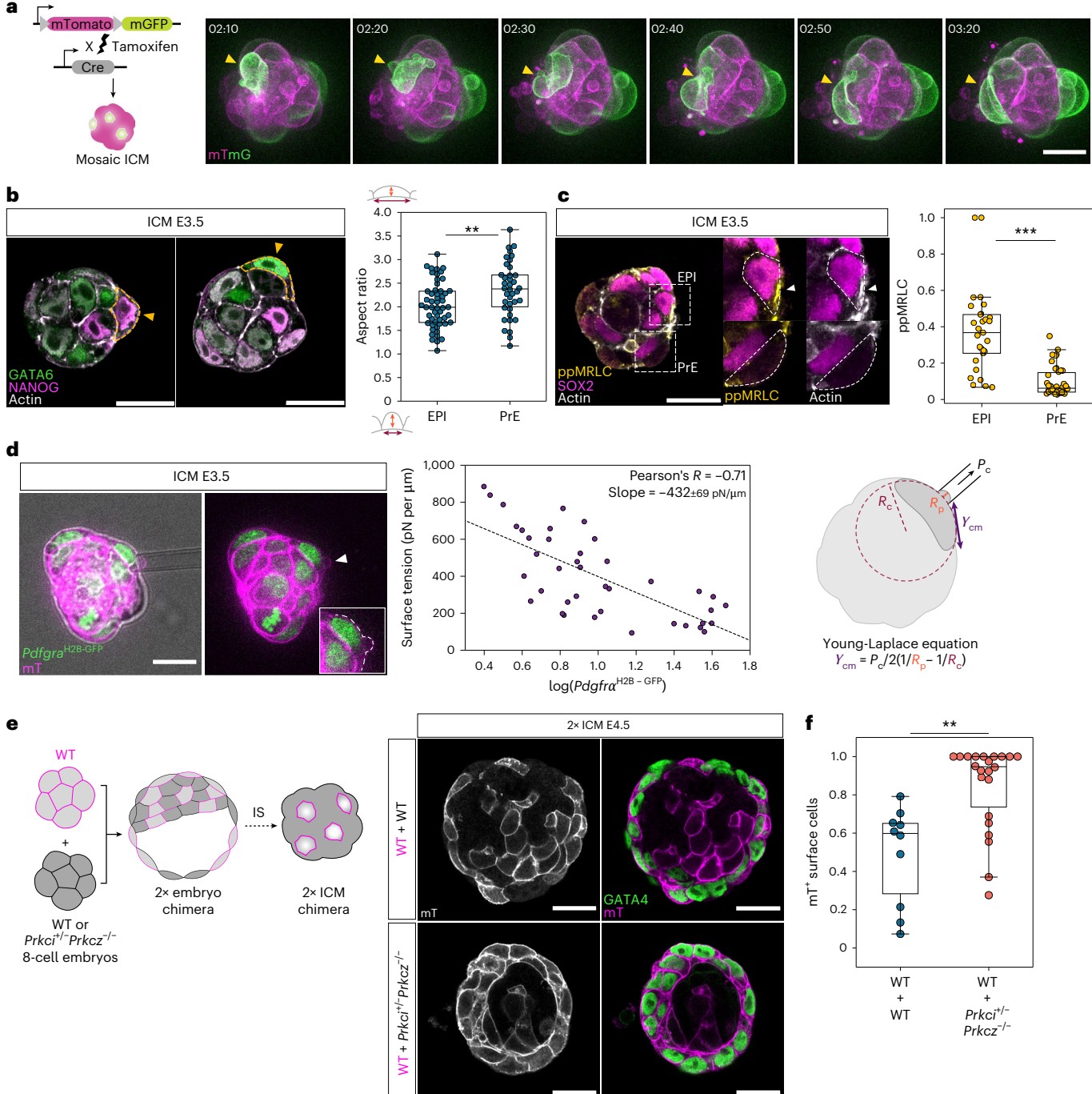

**Fig. 2 | Acquisition of the apical domain decreases surface tension and is sufficient for retaining PrE cells at the fluid interface. a**, Schematic representation and time-lapse images of a mosaic-labelled ICM isolated from an *R26-creER*;mTmG blastocyst. Time is indicated as h:min, *t* = 00:00 marks stage E3.5 + 3 h. Yellow arrowheads denote cell shape changes in a surface cell. **b**, Representative immunofluorescence images of E3.5 ICMs highlighting cell shape among EPI and PrE at the ICM–fluid interface. Analysis of surface cell aspect ratio from E3.5 isolated ICMs to compare EPI and PrE cell shape. *n* = 53 and 38 EPI and PrE cells from 16 isolated ICMs. Two-sided Mann–Whitney *U*-test, *P* = 6.44 × 10⁻³. **c**, Immunofluorescence image of an E3.5 isolated ICM showing the distribution of ppMRLC and Actin in EPI (top) and PrE (bottom) cells on the ICM surface, and quantification of normalized ppMRLC distribution at the outer cell cortex. *n* = 29 EPI and 42 PrE cells from 10 ICMs. Two-sided Mann–Whitney *U*-test, *P* = 2.18 × 10⁻⁹. **d**, Micropipette aspiration of E3.5 ICMs expressing *Pdgfra*^H2B-GFP (green) and membrane tdTomato (mT, magenta) and scatter-plot of measured surface tension of outer cells versus logarithm of *Pdgfra*^H2B-GFP fluorescence

intensity of the cell. White arrowhead marks the site of cell aspiration and the white dotted line indicates cell surface contour. *n* = 40 cells from 24 ICMs. Black dotted line denotes linear regression with slope −432 ± 69 pN μm⁻¹, Pearson's *R* = −0.71 and *P* = 2.8 × 10⁻⁷. Interfacial tension is calculated using the Young–Laplace equation where $\gamma_{cm}$ indicates cell–medium interfacial tension, $P_c$, aspiration pressure, $R_p$, radius of pipette and $R_c$, curvature radius of cell surface. **e**, Left, schematic for the experimental strategy using chimeric ICMs to test the functional role of apical polarity in cell positioning. Right, immunofluorescence images of 2× chimeric ICMs composed of cells from WT + WT combination (top) and WT + *Prkci*⁺/⁻*Prkcz*⁻/⁻ combination (bottom). IS, immunosurgery. **f**, Analysis of the surface retention of WT versus *Prkci*⁺/⁻*Prkcz*⁻/⁻ cells in chimeric ICMs from **e**. The plot indicates the proportion of all surface cells that are mT⁺ for WT + WT and WT + *Prkci*⁺/⁻*Prkcz*⁻/⁻ combinations. *n* = 10 and 22 ICMs for the two groups, respectively. Two-sided Mann–Whitney *U*-test, *P* = 1.08 × 10⁻³. Scale bars, 20 μm. **P* ≤ 0.05, ***P* ≤ 0.01, ****P* ≤ 0.001.

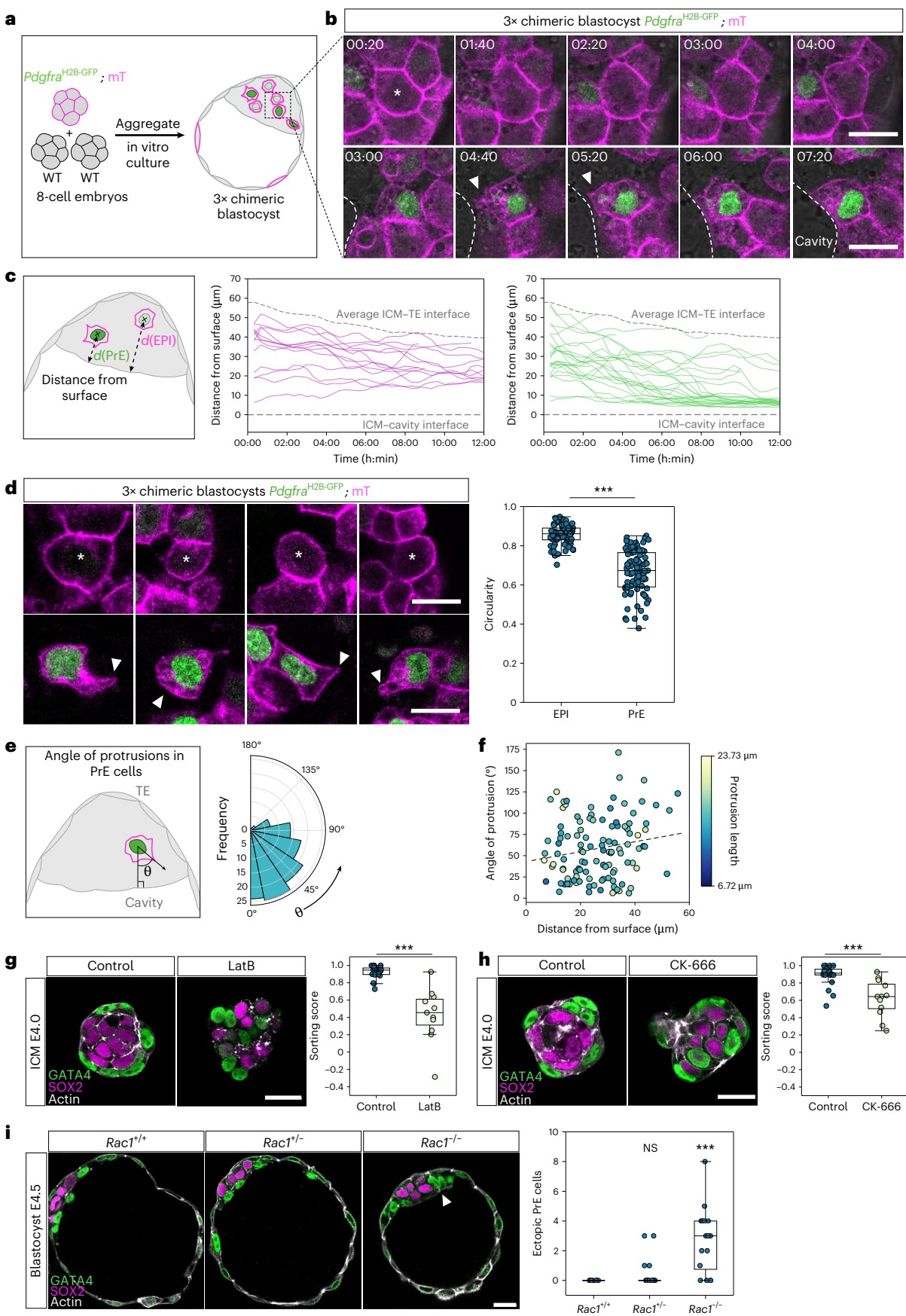

**Fig. 3 | Cell sorting involves active directed migration of PrE cells towards the surface via actin-mediated protrusions. a**, Experimental strategy to generate mosaic-labelled cells in large blastocysts to visualize EPI/PrE cell dynamics. **b**, Time-lapse images of a representative EPI cell (top) and PrE cell (bottom) expressing $Pdgfra$[H2B-GFP] and membrane tdTomato from mosaic-labelled blastocysts. White dotted lines denote cavity surface; white asterisk indicates EPI cell of interest; white arrowheads denote membrane protrusions in PrE cells. Time is indicated as h:min, $t = 00:00$ corresponds to start of live-imaging at stage E3.5 + 3 h. **c**, Distance of fluorescence-labelled EPI and PrE cells from the cavity surface. Cell position curves were smoothed using a rolling average. Grey dotted lines, average position of ICM–trophectoderm (TE) interface and the ICM–cavity interface. $n = 14$ EPI cells and 31 PrE cells from 13 embryos. **d**, Representative images (left) and circularity quantification (right) of EPI (top) and PrE (bottom) cell shapes in mosaic E3.75 blastocysts. White asterisks denote EPI cells; white arrowheads indicate PrE cell protrusions. Two-sided Mann–Whitney $U$-test, $P = 7.9 \times 10^{-22}$. $n = 68$ and 84 measurements from 14 embryos for EPI and PrE, respectively. **e**, Schematic and polar histogram of direction of PrE

cell protrusions in the ICM with respect to the cavity. $n = 113$ measurements from 12 embryos. Two-sided Mann–Whitney $U$-test compared with Extended Data Fig. 3c, $P = 1.31 \times 10^{-6}$. **f**, Scatter-plot of angle of protrusions in PrE cells versus distance of the cell from the cavity. Dotted line, linear regression. Pearson's $R = 0.179$, $P = 0.058$. $n = 113$ measurements from 12 embryos. **g**, Representative images of control and latrunculin B (LatB)-treated E4.0 isolated ICMs (left) and quantification of sorting score (right). $n = 20$ and 11 ICMs for control and LatB-treated ICMs, respectively. Mann–Whitney $U$-test, $P = 6.16 \times 10^{-6}$. **h**, Representative images of control and CK-666-treated E4.0 isolated ICMs and quantification of sorting score. $n = 20$ and 12 ICMs for control and CK-666-treated ICMs, respectively. Mann–Whitney $U$-test, $P = 3.69 \times 10^{-4}$. **i**, Immunofluorescence images (left) and quantification of number of ectopic PrE cells (right) of $Rac1^{+/+}$, $Rac1^{+/-}$ and $Rac1^{-/-}$ E4.5 blastocysts White arrowhead denotes ectopic PrE cell. $n = 9$, 17 and 16 blastocysts, respectively. Mann–Whitney $U$-test, $P = 0.133$ ($Rac1^{+/+}$ versus $Rac1^{+/-}$), $P = 0.001$ ($Rac1^{+/+}$ and $Rac1^{-/-}$). Scale bars, 20 μm. *$P \le 0.05$, **$P \le 0.01$, ***$P \le 0.001$.

To test experimentally the functional role of the apical polarization in cell positioning in the ICM, we generated chimeras between fluorescently labelled wild-type embryos and those lacking aPKC isoforms (Fig. 2e). If aPKC suppresses the cortical contractility in the apical domain and if differential contractility sorts surface EPI/PrE cells[37], aPKC knockout cells should be selectively positioned on the inside of the ICM. In control chimeras, where fluorescent and nonfluorescent wild-type embryos are combined, both fluorescent and nonfluorescent cells contributed to the PrE layer at the ICM–fluid interface. However, in aPKC knockout chimeras, most of the surface cells were derived from fluorescent wild-type embryos, whereas nonfluorescent aPKC knockout cells mostly accumulated inside (Fig. 2e,f). Together, these experiments demonstrate that apical polarization is sufficient for retaining PrE cells at the fluid interface due to the lower interfacial tension relative to EPI cells.

### Directed migration of PrE cells depends on actin dynamics and RAC1

Next, we investigated cellular dynamics inside the bulk of the ICM (Fig. 1h). To analyse differential cell dynamics and its underlying mechanisms, however, the small size and spherical geometry of the ICM system limit the interpretation of the analysis, in particular for cells around the centre of the ICM. Visualization of cell membrane is also necessary to fully characterize cellular dynamics. We thus generated large, mosaic blastocysts using fluorescent reporters marking cell fate with $Pdgfra$[H2B-GFP] and membrane with mTmG (Fig. 3a, Extended Data Fig. 3a and Supplementary Video 3). Live-imaging of these mosaic-labelled cells clearly showed distinct cell motility between EPI and PrE. PrE cells migrate towards the fluid-filled cavity with protrusions, whereas EPI cells remain within the ICM (Fig. 3b,c, Extended Data Fig. 3b and

Supplementary Videos 4 and 5), in agreement with the dynamics noted in the ICM culture (Fig. 1f,h). Of note, PrE cells exhibit a variety of cell shapes, whereas EPI cells remain largely spherical (Fig. 3d) and the protrusions of PrE cells are predominantly directed towards the blastocyst cavity (Fig. 3e, compare with Extended Data Fig. 3c), indicative of their directed migration. Notably, the directed migration and the length of PrE protrusions, 13.4 μm on average and 18.8 μm at 95th percentile (Extended Data Fig. 3d), are independent of the distance to the cavity interface (Fig. 3f).

To test whether PrE cells actively migrate towards the ICM–cavity interface, we first pharmacologically disrupted actin polymerization with latrunculin B. This effectively diminished the spatial segregation between EPI and PrE in ICMs (Fig. 3g) without compromising cell survival or proliferation (Extended Data Fig. 3e). Second, targeted inhibition of actin branching by blocking ARP2/3 activity with CK-666 resulted in failure of PrE cells to reach the ICM surface (Fig. 3h). Finally, pharmacological and genetic perturbation of RAC1, a small GTPase essential for active cell migration, led to the failed segregation of PrE cells to the ICM surface as a uniform layer (Fig. 3i, Extended Data Fig. 3f, and Supplementary Video 6), again without change in the ICM cell number (Extended Data Fig. 3g,h). Collectively, these data show that RAC1 activity and branched actin-mediated protrusions drive directed migration of PrE cells towards the ICM–cavity interface during EPI/PrE sorting.

### Apical polarity in PrE cells is required for directed migration and sorting

To identify what causes RAC1 activation and branched actin network in PrE cells, we examined our single-cell gene-expression database for genes differentially expressed between PrE and EPI cells at the

**Fig. 4 | Apical polarization in PrE cells is required for directed migration and sorting. a**, Immunofluorescence image of a 3× blastocyst at stage E3.75 showing laminin distribution around PrE cells. White dotted line denotes the ICM–cavity interface; white arrowhead indicates a GATA6-expressing PrE cell. **b**, Immunofluorescence image of a 3× blastocyst at stage E3.75 showing PKCλ+ζ distribution in PrE cells. White arrowhead denotes the leading edge of a PrE cell with PKCλ+ζ localization. **c**, Immunofluorescence image of an E3.75 ICM showing PKCλ+ζ localization in PrE and EPI cells. White dotted lines denote cell boundaries; the yellow line marks the segment from the cell inner edge (towards ICM centroid) to the cell outer edge (towards the ICM–fluid interface) along which fluorescence intensity is measured. **d**, Normalized fluorescence intensity of PKCλ+ζ in individual inside cells from E3.75 isolated ICMs. $n = 260$ cells from 32 ICMs. Each of the thin lines corresponds to measurement from one cell. Bold line and shaded region indicate mean ± s.d. of aPKC intensity for GATA6-high and GATA6-low cells. **e**, Left, schematic of polarization index. Right, boxplots for comparison of the polarization index in PrE (GATA6-high)

versus EPI cells (GATA6-low). GATA6 expression level is categorized as high or low by thresholding the bimodal distribution of GATA6 fluorescence intensity. $n = 136$ GATA6-high and 124 GATA6-low cells from 32 ICMs. One-way ANOVA, $P = 6.03 \times 10^{-20}$. **f**, Scatter-plot of polarization index of cells versus radial distance of the cell from the ICM centroid. Black dotted line indicates linear regression with Pearson's $R = 0.079$, $P = 0.205$. $n = 260$ cells from 32 ICMs. **g**, Immunofluorescence images of control and Gö6983-treated E4.0 isolated ICMs (left) and quantification of sorting score (right). $n = 16$ and 24 ICMs for control and Gö6983-treated ICMs, respectively. Two-sided independent-samples $t$-test, $P = 8.01 \times 10^{-4}$. **h**, Immunofluorescence images of representative WT, $Prkci^{+/+}Prkcz^{-/-}$ and $Prkci^{+/-}Prkcz^{-/-}$ E4.5 blastocysts (left) and quantification of number of ectopic PrE cells in E4.5 blastocysts from each group (right). $n = 25$, 17 and 14 blastocysts for WT, $Prkci^{+/+}Prkcz^{-/-}$ and $Prkci^{+/-}Prkcz^{-/-}$, respectively. Two-sided Mann–Whitney $U$-test, $P = 2.43 \times 10^{-4}$ and $6.36 \times 10^{-4}$. Scale bars, 20 μm. *$P \le 0.05$, **$P \le 0.01$, ***$P \le 0.001$.

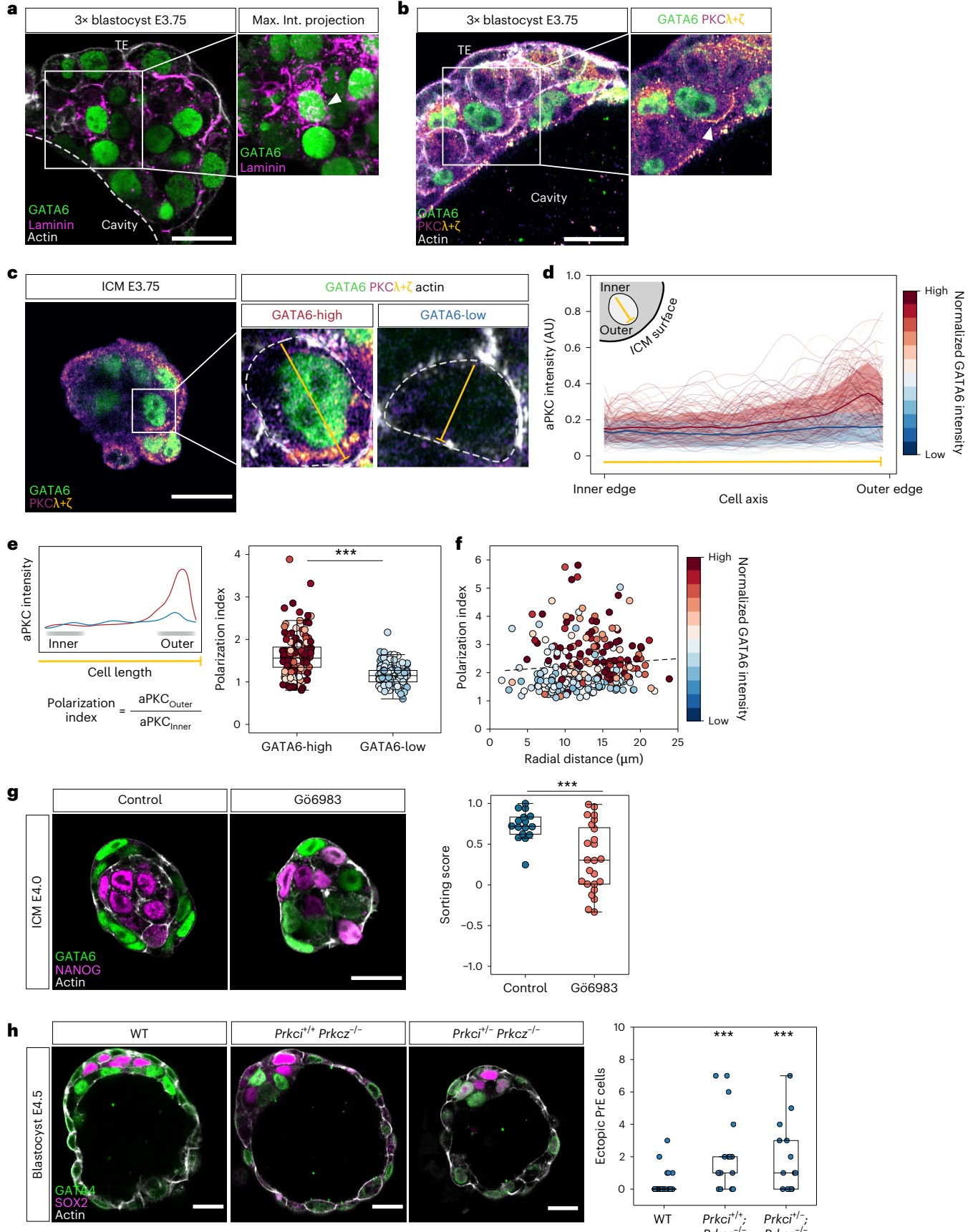

beginning of their lineage segregation at E3.5 (ref. [13]). In addition to *Fgfr2*, genes encoding ECM components such as the α1, β1 and γ1 subunits of laminin 1 (*Lama1*, *Lamb1* and *Lamc1*) and collagen IV (*Col4a1*, *Col4a2*), and factors involved in their synthesis such as *Serpinh1* and *P4ha2*, as well as PKCλ (*Prkci*), are specifically expressed in PrE cells[13] (Extended Data Fig. 4a,b). Immunostaining of the embryo confirmed dense accumulation of laminin and aPKC in PrE cells (Fig. 4a,b and Extended Data Fig. 4c). Notably, aPKC is localized near the leading edge of migrating PrE precursors in the E3.75 blastocyst (Fig. 4b). To quantitatively analyse the aPKC localization in PrE cells, we measured the accumulation of aPKC at the subcortical region in PrE and EPI cells, which showed that aPKC is differentially localized in PrE cells at the side facing towards the ICM surface (Fig. 4c,d). GATA6-expressing PrE cells are more polarized than EPI cells (Fig. 4e), independently of cell position within the ICM (Fig. 4f), suggesting that PrE cells acquire apical polarity in a cell-autonomous, fate-dependent manner. These findings led us to the hypothesis that the apical polarization in PrE cells may be functionally linked with their front-rear polarity for directed migration.

To test the functional role of aPKC in PrE cell migration, we first inhibited the activity of aPKC in the ICM with Gö6983, which disrupted sorting and patterning (Fig. 4g, Extended Data Fig. 4d and Supplementary Video 7) in agreement with a previous report[24]. Further analysis in mosaic-labelled blastocysts showed that Gö6983 disrupted the directed movement of PrE cells towards the blastocyst fluid cavity (Extended Data Fig. 4e). PrE cells do not extend protrusions towards the cavity upon inhibition of aPKC, and instead exhibit more rounded cell shape comparable with EPI cells (Extended Data Fig. 4f). Finally, combined genetic knockouts of aPKC isoforms, *Prkci*⁺/⁺*Prkcz*⁻/⁻ and *Prkci*⁺/⁻*Prkcz*⁻/⁻, resulted in smaller ICMs (Extended Data Fig. 4g) with failed segregation of PrE to the cavity surface (Fig. 4h and Supplementary Video 8), indicating that functional apical polarity is necessary for directed migration and sorting of PrE cells. Together, early in differentiation and within the ICM, PrE cells acquire the apical polarity that is required for directed migration and sorting to the ICM–cavity interface.

## ECM deposited by PrE cells builds a gradient and likely guides PrE cell migration

Thus far, our findings showed that acquisition of apical polarity is required and sufficient for PrE cell migration and surface retention, respectively. However, it is unclear what directs PrE cells within the ICM tissue to migrate towards the ICM surface, particularly towards the ICM–fluid interface in the blastocyst. PrE cells near the cavity may be trapped at the surface when protrusions reach the fluid interface, though this mechanism per se cannot explain the biased orientation of directed protrusions of PrE cells deeper than 20 µm from the cavity (Fig. 3e,f). We reasoned that this surface trapping of PrE cells near the cavity may break tissue-level symmetry with respect to the distribution of salt-and-pepper EPI and PrE cells. As PrE cells start expressing ECM components (Fig. 4a and Extended Data Fig. 4a), their secretion by these trapped PrE cells may cause a shift in the ECM distribution, with more ECM deposited near the fluid cavity. This shifted distribution of ECM may in turn guide subsequent PrE cell migration at the tissue scale towards the cavity surface.

Colocalization of active integrinβ1 with laminin in PrE cells indicates active cell–ECM adhesion, in agreement with the hypothesis that ECM may guide PrE cell migration (Extended Data Fig. 5a,b). Additionally, inner PrE cells extend protrusions towards laminin deposited around PrE cells that have reached the surface of the cavity (Extended Data Fig. 5c).

We investigated the interplay between PrE dynamics and ECM distribution in silico using computational simulations of a custom cellular Potts model (CPM)[38]. Our framework relies on Poissonian stochastic dynamics with explicit energy and time scales, instead of the traditional Metropolis scheme, to account for heterogeneities in material transport properties in systems containing different cell types and extracellular components (Fig. 5a). To simulate the sorting process, we chose cell tension parameters from the experimentally observed ranges of values and used a nonunique set of the remaining parameter values that recapitulate the exponential-like relaxation of the EPI/PrE sorting score observed in our live-imaging experiments with isolated ICMs[38]. Our implementation directly introduces ECM into the CPM framework. Specifically, we include ECM components that are actively produced by PrE cells deposited at their cell–cell interfaces, which then undergo diffusion and degradation. For simplicity, we assume that ECM surface properties are the same as those of PrE cells (Methods). Our simulations not only show the sorting of EPI:PrE, but also indicate a concomitant progressive change in the distribution of ECM components. While initially abundant within the bulk of the ICM before sorting, the ECM progressively accumulates near the periphery, and is highest near cells facing the surrounding medium, particularly at the PrE–EPI interface (Fig. 5b and Supplementary Video 9).

To test this prediction experimentally, we immunostained enlarged blastocysts and isolated ICMs against laminin to gain higher spatial resolution and quantified its distribution across the ICM. This revealed that the uniform distribution of laminin at E3.5 indeed changes into a gradient increasing towards the ICM surface at E3.75 when EPI/PrE-sorting takes place, before forming the basement membrane at the PrE–EPI boundary (Fig. 5c,d and Extended Data Fig. 5d,e). These findings support the model in which retention of PrE cells near the cavity

**Fig. 5 | Extracellular matrix deposited in the ICM guides PrE cells towards the cavity surface. a**, Schematic of a 3D Poissonian CPM[38]. The system state is given by a collection of voxels with one of the following three identities: 0, 1 and 2 for medium, PrE and EPI, respectively. Cell–cell and cell–medium interfaces have tensions $\gamma_{PrE:M} < \gamma_{PrE:PrE} < \gamma_{EPI:EPI} < \gamma_{PrE:EPI} < \gamma_{EPI:M}$, and the two cell types have different kinetic parameters $\alpha_{EPI} < \alpha_{PrE}$. ECM is secreted by PrE cells and is taken to have the same parameters as PrE cells for simplicity. See Methods for details. **b**, Predicted change in the distribution of ECM from 3D Poissonian CPM simulations. Data are mean ± s.e.m. **c**, Immunofluorescence images of ICMs in 3× blastocysts at stages E3.5, E3.75 and E4.5. **d**, Laminin fluorescence intensity from the ICM–trophectoderm interface to the ICM–cavity interface in maximum intensity projections of the blastocysts from **c**. Data are mean ± s.d. from $n = 5$, 6 and 5 embryos for the different stages, respectively. Lines of the same colour correspond to measurements from the same embryo at respective stages. **e**, Left, experimental strategy using coated microbeads to introduce ectopic laminin localization in the ICM. Middle, brightfield and immunofluorescence images of 2× E4.5 blastocysts with implanted beads coated with E-cadherin (CDH1) or E-cadherin + laminin (CDH1 + Lam). Yellow asterisks and dashed circles indicate microbead position in the ICM. Right, quantification of ectopic PrE cells localized at the coated beads in 2× E4.5 blastocysts. Two-sided Mann–Whitney *U*-test, $P = 0.0319$; $n = 4$ and 3 embryos with successfully integrated E-cadherin-coated beads and E-cadherin + laminin-coated beads, respectively. **f**, Left, experimental strategy to rescue the incorrectly patterned phenotype of *Lamc1*⁻/⁻ blastocysts. Middle, immunofluorescence images of late-stage blastocysts from *Lamc1*⁻/⁻ and chimeric blastocysts between comprising *Lamc1*⁻/⁻ + WT cells. White arrowheads indicate *Lamc1*⁻/⁻ cells successfully segregated to the PrE monolayer at the fluid interface. Right, quantification of ectopic PrE cells in *Lamc1*⁻/⁻ blastocysts and *Lamc1*⁻/⁻ + WT chimeric blastocysts; $n = 8$ and 5 embryos for the two groups, respectively. One-way ANOVA, $P = 8.88 \times 10^{-3}$. **g**, Schematic for EPI–PrE fate segregation. Until stage E3.5, there is negligible asymmetry in ICM composition. Around stage E3.75, PrE cells in the ICM acquire hallmarks of apical polarity and begin to express and secrete ECM components. Polarization of PrE cells lowers their tension and PrE cells at the cavity are trapped. Apolar EPI cells have higher surface tension and move inwards. Inner PrE cells extend cell protrusions that facilitate their migration towards the cavity for tissue pattern formation. **h**, Multiscale feedback model of tissue-level patterning between cell polarization, mechanics, cell migration and ECM deposition underlying blastocyst patterning. Scale bars, 20 µm. **$P ≤ 0.01$.

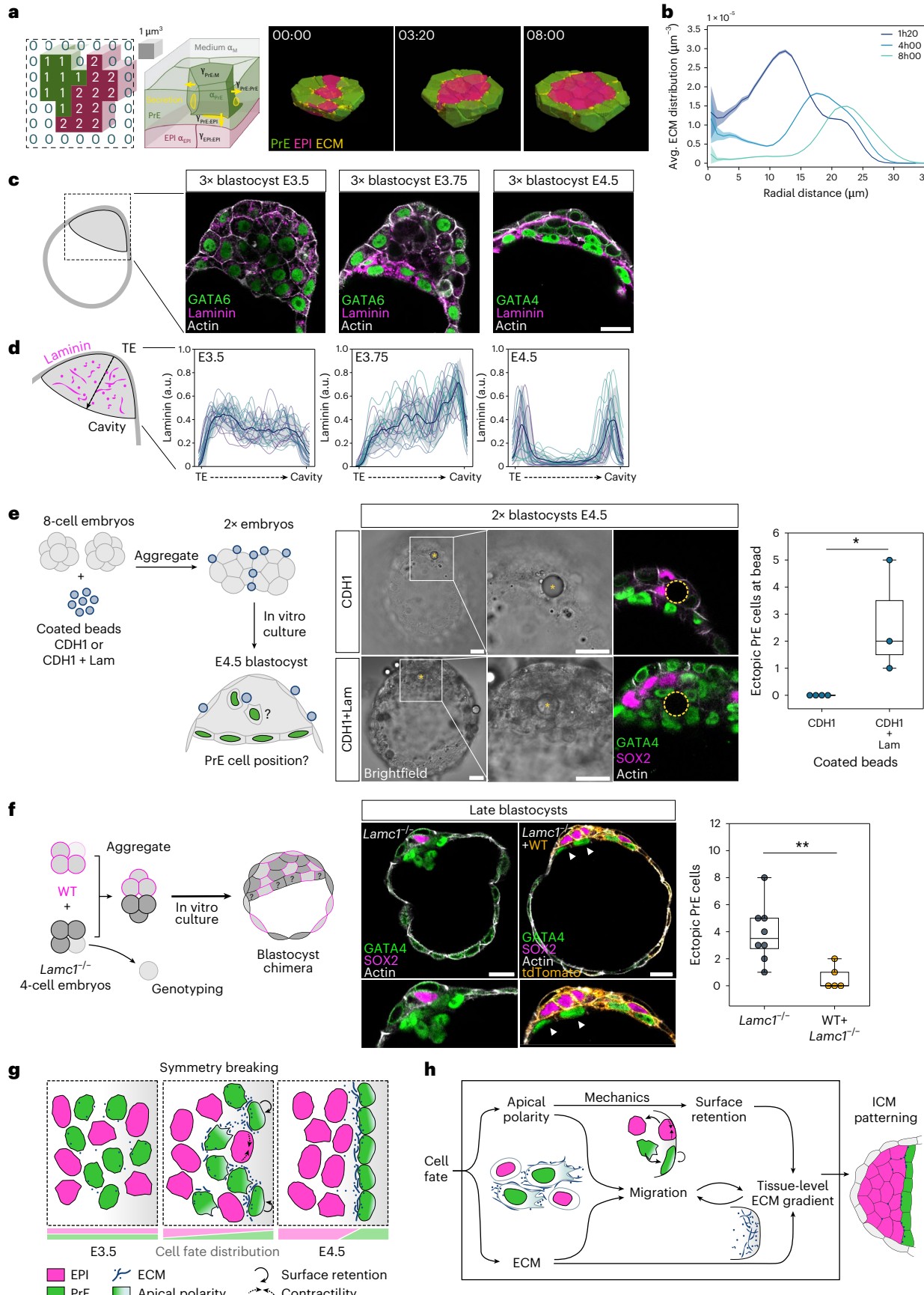

surface breaks symmetry in ECM distribution (Extended Data Fig. 5f) and builds a gradient across the ICM tissue, which may guide other PrE cells to migrate towards the cavity surface at subsequent stages.

To determine whether laminin distribution could guide PrE cell migration, we examined whether the asymmetry in laminin deposition surrounding PrE cells correlates with that in PrE cell protrusions indicative of cellular migratory activity, detected as regions of high local membrane curvature (Extended Data Fig. 5g). In contrast to PrE cells at stage E3.5, the distribution of cell membrane regions enriched for laminin and those with high curvature overlap, and are both oriented towards the fluid cavity at the onset of migratory activity at E3.75 (Extended Data Fig. 5h). These findings are consistent with the potential of laminin distribution to guide PrE cell migration.

The role for laminin in PrE cell migration is also in line with our earlier findings that integrinβ1 and laminin γ1 are required for proper PrE segregation to the ICM surface[39]. To test further the functional role of ECM in guiding PrE cell migration, we experimentally induced ectopic localization of ECM and examined whether it can attract PrE cell migration. To this end, we implanted laminin-coated poly(methyl methacrylate) (PMMA) microbeads into the ICM of blastocysts and examined the distribution of PrE cells at stage E4.5 (Fig. 5e and Extended Data Fig. 5i). Notably, PrE cells were attracted to the laminin-coated bead in addition to the ICM–fluid interface, without changing the EPI:PrE proportion (Extended Data Fig. 5j), in stark contrast to control beads coated with only E-cadherin, which did not disrupt EPI/PrE segregation (Fig. 5e and Supplementary Video 10). These results indicate that laminin deposition is functionally sufficient for guiding PrE cell migration.

Furthermore, to test the role of ECM, we generated chimeric embryos in which the ICM consists of both wild-type and $Lamc1^{-/-}$ cells (Fig. 5f and Supplementary Video 11). If laminin deposition guides PrE migration, laminin deposited by wild-type cells in $Lamc1^{-/-}$ blastocysts should be sufficient to rescue the disrupted PrE pattern in $Lamc1^{-/-}$ embryos. Immunofluorescence showed that while PrE cells in late-stage $Lamc1^{-/-}$ blastocysts tend to clump together in agreement with our earlier findings[39], those in the chimeric embryos successfully form a segregated monolayer at the fluid interface (Fig. 5f), supporting the functional role of laminin deposition in directing PrE cell migration.

Together, these data led us to a mechanistic model of EPI/PrE sorting that integrates cell fate, polarity, mechanics and tissue-scale positional information (Fig. 5g). First, within the ICM tissue, salt-and-pepper-distributed PrE cells acquire apical polarity that induces cell protrusive and migratory activity. Protrusions from PrE cells near the cavity reach the fluid interface and induce their retention at the surface, which shifts the balance of PrE cell distribution and thereby that of secreted ECM. The progressively increasing asymmetry in ECM distribution can guide other PrE cells to migrate towards the cavity, which in turn contribute to the emerging tissue-level ECM gradient, effectively enabling collective cell migration towards the

surface, which we term 'breadcrumb navigation' (Fig. 5g). This multiscale feedback model explains tissue-level symmetry breaking and dynamic pattern emergence within an initially equivalent population of cells (Fig. 5h).

## Fixed EPI:PrE cell proportion challenges precision in ICM patterning

While this feedback model may explain dynamic EPI–PrE cell segregation and pattern emergence, we sought to understand how this mechanism is linked with cell fate specification. Cell lineage, division pattern and gene-expression pattern are variable among embryos in pre-implantation mouse development, and in such systems, feedback of cell positional information to cell fate specification could ensure robust patterning[20,37,40,41]. In line with this model, earlier studies[14,22,27] proposed position-dependent PrE fate specification, in which cells on the cavity surface are induced to differentiate into PrE. However, this is incompatible with another notion that the proportion of EPI:PrE cells is fixed according to the gene regulatory network between GATA6, NANOG and FGF signalling activity[15,17–19,25,42,43].

First, we analysed the proportion of EPI:PrE cells in blastocysts and ICMs experimentally isolated from blastocysts and found it indeed constant (Fig. 6a) with PrE proportion 0.605 ± 0.078 (mean ± s.d., $n = 101$ embryos), in agreement with earlier studies[25,42]. Next, cell-lineage tracking with fate markers showed highly limited contribution of position-dependent fate-switching during EPI:PrE sorting; only 3 out of 181 cells differentiated to PrE by increasing the expression of $Pdgfra^{H2B-GFP}$ on the cavity surface, whereas 2 out of 93 cells differentiated to EPI by decreasing the $Pdgfra^{H2B-GFP}$ signal inside the ICM (Fig. 6b). These findings suggest that the fate and proportion of EPI:PrE cells are fixed in E3.5–4.5 blastocysts.

To distinguish the presence or absence of position-dependent cell-fate plasticity, we challenged the system by manipulating embryo size up to fourfold larger or smaller (Fig. 6c). Embryo size manipulations result in a linear scaling of ICM cell numbers, but a nonlinear scaling of the ICM base radius (Extended Data Fig. 6a,b). These major changes in ICM cell number and shape lead to corresponding differences in the ICM interface:volume ratio (Extended Data Fig. 6c), which, in the absence of position-dependent cell-fate plasticity, would result in failure to fit one layer of PrE cells on the ICM surface (Fig. 6c and Supplementary Video 12). In agreement with the notion of fixed EPI:PrE proportion, we found that, despite the wide range of variability in the number of ICM cells in larger or smaller embryos, the proportion of PrE remained constant at 0.599 ± 0.006 ($n = 153$; Fig. 6d). Remarkably, in larger embryos, we observed ectopic PrE cells within the ICM, and conversely, in smaller embryos, ectopic EPI cells at the ICM–fluid interface (Fig. 6c,e and Supplementary Video 12). These findings support the lack of cell fate plasticity at this stage and the lack of feedback from cell position to fate specification.

**Fig. 6 | The fixed proportion of EPI:PrE cells without cell fate-switching challenges precision in ICM patterning. a**, Proportion of cell fates in the ICM in embryos under different conditions during development. One-way ANOVA, $P = 0.085$. $n = 19, 29, 21$ and 32 embryos for the different groups, respectively. Mean PrE proportion of 0.605 ± 0.078. **b**, Limited contribution of position sensing and cell fate-switching in E3.75 ICMs to final patterning of the ICMs. Consecutive time-lapse images from isolated ICMs expressing $Pdgfra^{H2B-GFP}$ and the corresponding lineage tree for the ICM. White dotted line marks ICM boundary. $t = 00:00$ corresponds to start of live-imaging at stage E3.5 + 3 h, following completion of immunosurgery. Lineage tree of an isolated ICM from single-cell tracking in Fig. 1c,d. Yellow arrowhead, inside cell that increases $Pdgfra^{H2B-GFP}$ expression after moving to the surface and its lineage. Stacked bar plots indicating frequency of position sensing and fate-switching contributing to the final EPI/PrE cell fates. **c**, Schematic and immunofluorescence images of size-manipulated blastocysts at stage E4.5. White arrowheads indicate ectopic EPI cells in smaller blastocysts and ectopic PrE cells in larger blastocysts.

**d**, Number of PrE cells in the ICM in E4.5 size-manipulated blastocysts. Dotted line shows linear regression with Pearson's $R = 0.98$; $P = 1.18 \times 10^{-134}$. PrE proportion of 0.599 ± 0.006. $n = 26$ embryos for 2/8×, 29 embryos for 3/8×, 24 embryos for 4/8×, 29 embryos for 1×, 17 embryos for 2×, 18 embryos for 3× and 10 embryos for 4× size ratios. **e**, Quantification of ectopic EPI/PrE cells in size-manipulated E4.5 blastocysts. The number of ectopic cells is plotted as a function of total number of cells in the ICM. **f**, Left, schematic of chimera experiments to test feedback between cell fate and position in the ICM. Right, immunofluorescence images of 2× chimeric E4.5 blastocysts composed of cells from WT + WT combination (left column) and WT + $Myh9^{+/-}$ combination (right column). White arrowhead marks GATA4-negative cells on the ICM surface. **g**, Quantification of number of GATA4-negative cells on the cavity surface in WT + WT combination and WT + $Myh9^{+/-}$ combination of E4.5 chimeric blastocysts. $n = 40$ and 19 embryos for the two groups, respectively. Two-sided Mann–Whitney $U$-test, $P = 2.82 \times 10^{-5}$. Scale bars, 20 µm. *$P \leq 0.05$, **$P \leq 0.01$, ***$P \leq 0.001$.

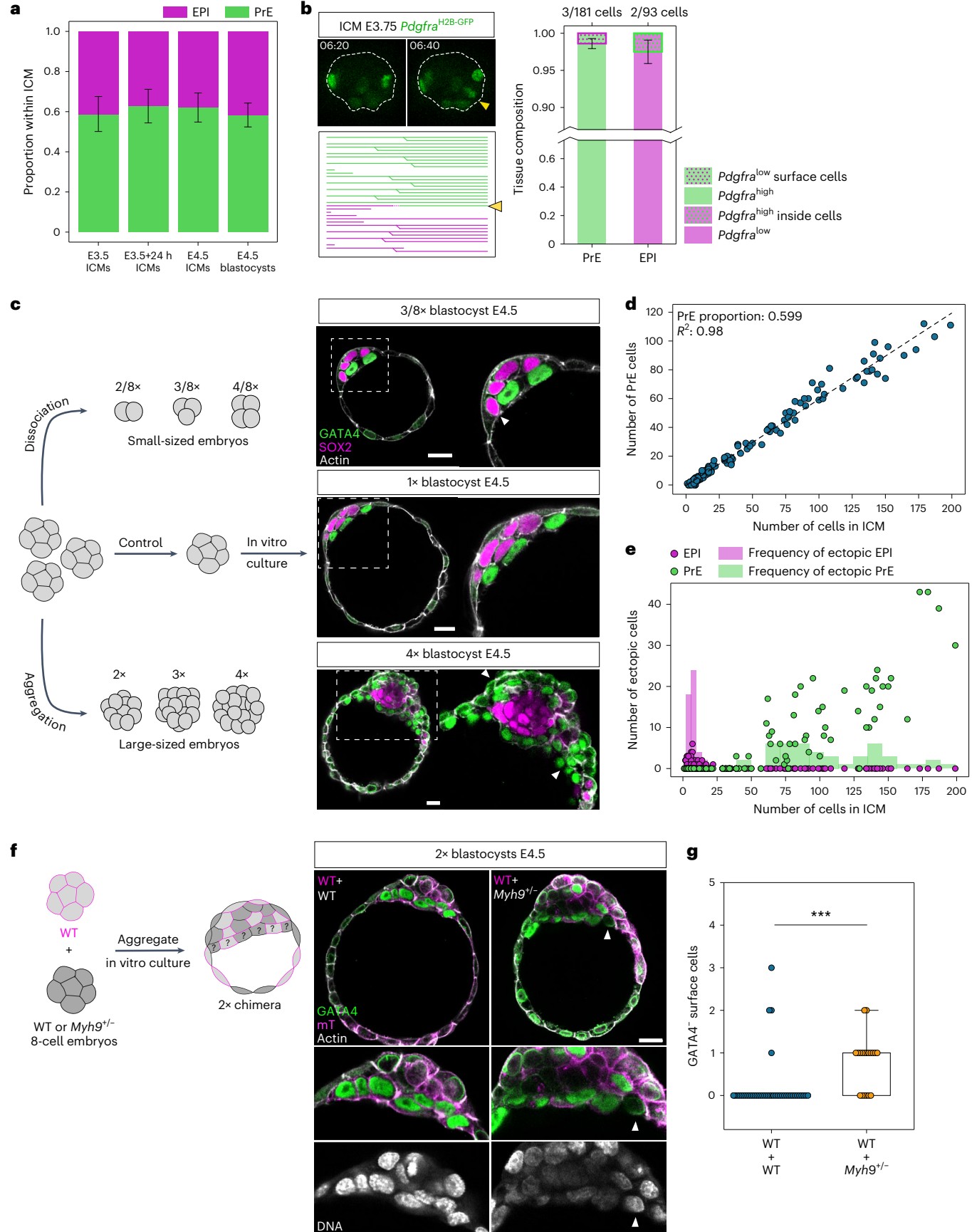

To unequivocally demonstrate the presence or absence of cell fate plasticity at this stage, we further challenged the system by generating chimeric blastocysts using embryos heterozygous for *Myh9*, which encodes the myosin heavy chain (*Myh9*[+/−]; Fig. 6f). The chimeras between wild-type and *Myh9*[+/−] embryos would force some EPI cells derived from *Myh9*[+/−] embryos to be located on the ICM surface as they have relatively lower cortical tension. Of note, *Myh9*[+/−] embryos form blastocysts with ICM cell number and EPI:PrE proportion comparable with wild-type (Extended Data Fig. 6d and Supplementary Video 13). Without position-dependent cell fate plasticity, these cells would not change fate to PrE despite being on the surface. Control chimeras form a precise ICM pattern at the E4.5 stage without ectopic cells (Fig. 6g). By contrast, chimeras with *Myh9*[+/−] embryos show ectopic cells on the ICM surface that do not change fate to PrE, thus experimentally supporting the model that EPI/PrE cell fate and their proportions are fixed without plasticity during ICM patterning in late blastocysts.

## ICM cell composition is optimal for embryo size and geometry across mammals

The fixed EPI:PrE proportion and lack of plasticity present a challenge for early mammalian embryos to achieve precision in blastocyst patterning, because cell numbers and embryo geometry are variable among pre-implantation embryos, and in general, surface area scales nonlinearly with volume. Therefore, we asked how robust patterning is ensured in the absence of cell fate plasticity. Not only the number of cells but also their shape varies in each embryo. Thus, the variability of cell shape defines the range of cell numbers with which an embryo with a given geometry and fixed EPI:PrE proportion can achieve precise patterning, covering the bulk of EPI cells with an intact PrE monolayer. To estimate this range and examine its distribution across embryos of various sizes, we characterized the in vivo geometry of the tissue and individual cells from immunostained blastocysts. Specifically, we approximated the ICM shape as a combination of two spherical caps, corresponding to the ICM–trophectoderm interface and the ICM–cavity interface, and thus obtained estimates of the ICM–cavity interface area $A_{Interface}$ using measurements of the cap heights and base radii from immunostaining images (Fig. 7a). Next, we measured the PrE cell apical areas at the cavity surface and determined the 10th and 90th percentiles of the cell area $q_{10\%} = 157 \ \mu m^2$ and $q_{90\%} = 376 \ \mu m^2$, respectively (Extended Data Fig. 7a). Given the fixed proportion $f = 0.6$ of PrE cells, we calculated the corresponding range of the total PrE area $A_{PrE} = (f n \ q_{10\%}, f n \ q_{90\%})$ for a given number of ICM cells, $n$. This range is bound by the marginal sizes of a hypothetical monolayer formed by the PrE cells in an embryo with a total ICM cell count $n$, because the variability in single-cell apical areas gives rise to an interval of possible values that a total PrE area may have (Fig. 7a). By comparing $A_{Interface}$ and $A_{PrE}$, we predict the presence of gaps or multilayered regions for different ICM sizes: if $A_{Interface}$ is larger than the maximal bound of $A_{PrE}$, we expect a gap

in the PrE monolayer with EPI cells exposed to the interface, whereas if $A_{Interface}$ is smaller than the minimal bound of $A_{PrE}$, superfluous PrE cells would be located inside the ICM, thus forming a PrE multilayer.

Our measurements showed that most normal-sized embryos have ICM–cavity interface areas within the range that PrE cells could cover, thereby enabling formation of a PrE monolayer (Fig. 7b; $n = 29$ embryos). Note that surface-to-volume scaling is nonlinear, implying that a fixed PrE fate proportion may produce a surface monolayer only within certain size limits for a given shape. For example, a spherical ICM with a hemispherical $A_{Interface}$ is compatible with PrE monolayer formation only within a particular range of embryo sizes (Fig. 7b, dotted line). Furthermore, by counting the frequency of embryos inside (outside) the region $A_{Interface}$, we estimated the probability of observing a monolayer (gap/multilayer) and found that gap formation is more likely to occur in smaller embryos and multilayers are more likely in larger embryos (Figs. 6c,e and 7b, inset). We then compared the probability of PrE monolayer formation across embryos of various sizes and found it highest in the normal-size embryo (Extended Data Fig. 7b; 82.5%). Further, the probability of monolayer formation was higher than that of multilayers or gaps for embryos across the fourfold size difference (from double to half size), indicating the robustness of precise ICM pattern formation against natural variability of embryo size (Fig. 7c). However, when the probability is calculated for the scenario where the ICM is composed of 40% or 80% PrE, the likelihood of forming gaps in the PrE layer is higher in 40% PrE and that of multilayer formation is higher in 80% PrE (Fig. 7c). Notably, the probability of monolayer formation without a gap or multilayer is highest with 60% PrE for the mouse embryo for a range from double to half size, suggesting that this fixed EPI:PrE proportion is optimal for patterning the mouse ICM given its size and geometry.

Other mammalian species have different embryo sizes and proportions of EPI:PrE in the ICM[44–46]. Our findings, which suggest an optimal proportion of EPI:PrE for a specific embryo size and geometry, therefore raise the question whether different mammalian species have distinct optimal proportions according to their respective sizes and geometries. We tested this prediction by first analysing monkey blastocysts. Monkey embryos are larger in size than mouse embryos (Fig. 7d,e and Supplementary Video 14) and the ICM has a higher proportion of PrE cells (0.702 PrE, $n = 15$; Fig. 7f). Notably, measurements of cell and ICM geometry show that the observed monkey blastocysts have ICM–fluid interfacial areas within the range that PrE areas could cover when the increased 70% proportion of PrE cells is taken into account (Fig. 7g). However, with a 60% PrE proportion (as in mouse embryos) the hypothetical area of PrE cells in monkey embryos would decrease substantially below the observed values (Fig. 7g), indicating that the 70% PrE proportion is optimal for monkey blastocyst size.

Next, we tested the prediction with human blastocysts using image datasets published recently[47]. Human blastocysts are larger, with their

**Fig. 7 | The fixed proportion of EPI:PrE cells is optimal for the specific embryo size and ICM geometry. a**, Schematic for estimation of ICM–cavity interface area $A_{Interface}$ and total PrE area $A_{PrE}$ in blastocysts. For details, see Methods. **b**, Scatter-plot of $A_{Interface}$ as a function of total ICM cell number for size-manipulated E4.5 mouse blastocysts. Monolayer formation is predicted between the minimal and maximal bounds of $A_{PrE}$ based on the fixed fate ratio. Black dotted line indicates the surface area of a hemispherical PrE as a function of the volume corresponding to the respective number of cells, illustrating how nonlinear surface-to-volume scaling permits monolayer formation only within a particular range of embryo sizes for a simplified shape. Inset, zoom-in for ×0.25 and ×0.375 size ratio. $n = 25, 26, 24, 29, 17, 18$ and $10$ embryos for 2/8×, 3/8×, 4/8×, 1×, 2×, 3× and 4× size ratios, respectively. **c**, Estimated probabilities of forming PrE gap, monolayer and multilayer with embryo size for ICM composition of 40%, 60% and 80% PrE, depicted as mean ± s.e. Sample numbers same as **b**. Grey shaded region denotes the natural variability in embryo size. **d**, Immunostaining images of E4.5 mouse blastocysts and monkey blastocysts 7–8 days after intracytoplasmic

sperm injection (ICSI). **e**, Boxplot of blastocyst volume in mouse and monkey blastocysts. $n = 29$ and $15$ embryos, respectively. **f**, EPI:PrE proportion for mouse and monkey blastocysts, plotted as mean ± s.d., $n = 29$ and $15$ embryos, respectively. Mean PrE proportion, $0.59 ± 0.09$ and $0.70 ± 0.05$ for mouse and monkey embryos, respectively. **g**, Scatter-plot of $A_{Interface}$ as a function of total ICM cell number for monkey blastocysts 7–8 days after ICSI. Monolayer formation is predicted between the maximal and minimal bounds of $A_{PrE}$, for 70% and 60% PrE proportion within the ICM. **h**, Boxplot of human blastocyst volume, and human EPI:PrE proportion plotted as mean ± s.d. Mean PrE proportion for human embryos, $0.55 ± 0.11$, $n = 15$ embryos. **i**, Shape of the ICM–fluid interface in mouse, monkey and human blastocysts. Black dotted line indicates the ICM base diameter. Elevation of the ICM–fluid interface $h$ is measured from the ICM base diameter. $n = 29, 15$ and $15$ embryos, respectively. **j**, Scatter-plot of $A_{Interface}$ as a function of total cell number in the ICM for human blastocysts at stages late D6/ D7. $n = 15$ embryos. Scale bars, 20 μm.

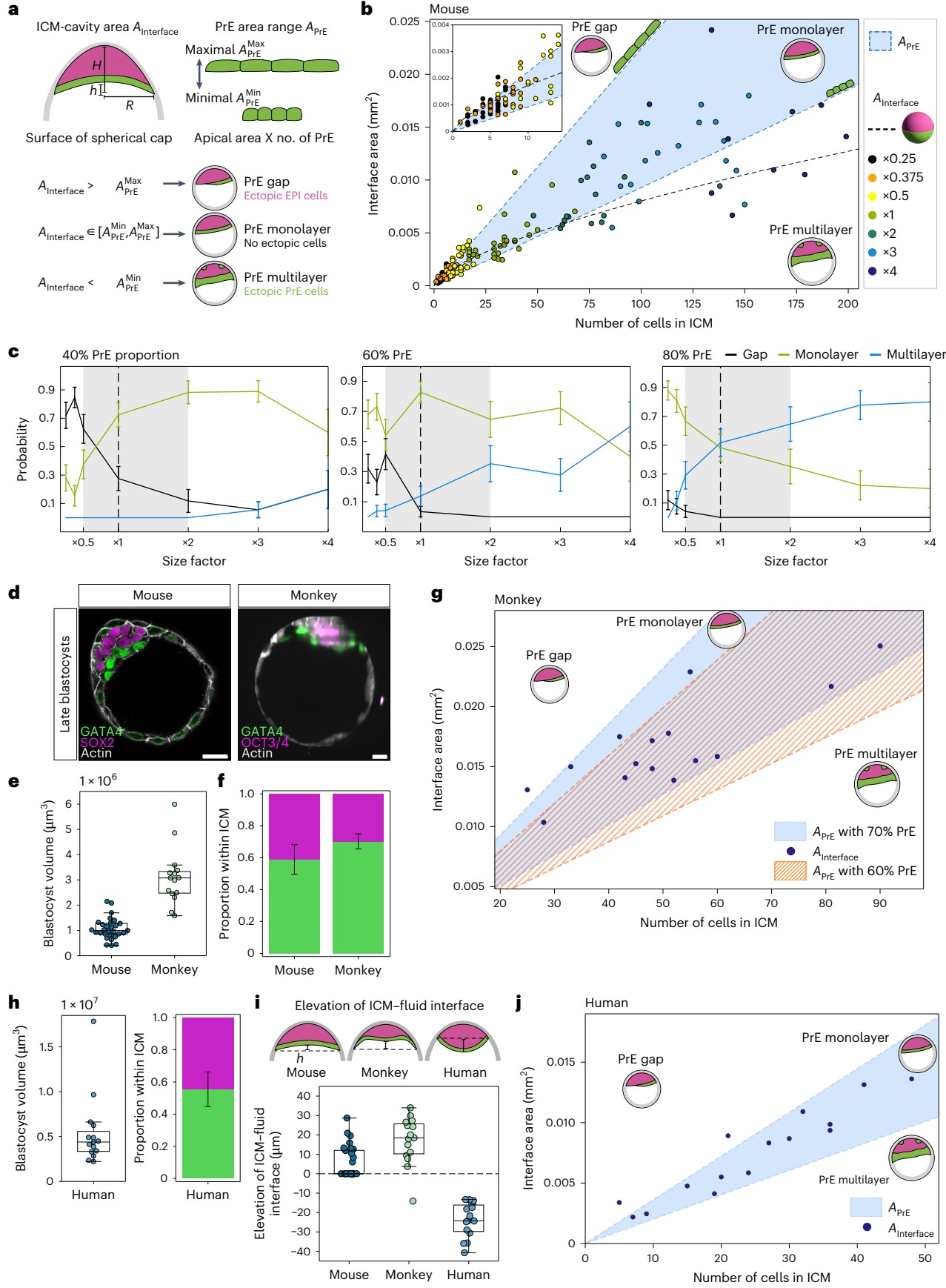

ICMs composed of a lower proportion of PrE cells (0.554 PrE, $n = 15$; Fig. 7h). The ICM–fluid interface in human blastocysts has a convex shape (Fig. 7i), allowing us to test whether the fate proportion is also optimized for a different tissue geometry. Notably, the ICM–fluid interfacial area in a majority of the human blastocysts is within the range that PrE cells could cover (Fig. 7j). Together, these data strongly suggest that an optimal proportion of the ICM cell fates is species-specific and that this optimal EPI:PrE ratio adapts to embryo size and tissue geometry.

## Discussion

Overall, this study uncovers how mammalian blastocysts of different sizes and shapes maintain robust fate patterning. We find that tissue-level symmetry within the ICM is first broken by retention of PrE cells at the fluid interface by differential surface tension. This builds a tissue-wide ECM gradient deposited by PrE cells, which potentially guides active migration of PrE cells driven by their acquisition of apical polarity. Despite the fixed proportion of EPI:PrE cells, patterning is robust against naturally variable sizes of the embryo because this proportion is species-specific and optimal for embryo size and geometry.

Although cell fate specification[12,13,15–19,28,43,48,49] and sorting in the ICM[14,22,24,27,29] have been studied, these processes were not investigated in combination to measure fate-specific dynamics underlying cell sorting. PrE-specific cell surface fluctuations and differential cell fluidity were recently shown to be sufficient for sorting cell aggregates[29], though how these properties arise only in PrE cells, and how this could pattern the blastocyst ICM with a specific in vivo geometry remained elusive. Here, we report for the first time, that PrE cells undergo RAC1-dependent active migration. Furthermore, we find autonomous polarization of PrE cells within the ICM, which drives the formation of actin-based protrusions for cell migration, corroborating a functional link between polarity and cell sorting, in agreement with previous findings[24,26]. While apical polarization has thus far been detected in PrE cells only after their sorting to the fluid interface[24,26,50], here we characterize the asymmetric cortical localization of aPKC in PrE cells in the salt-and-pepper ICM. This apical polarization is atypical for two reasons: first, the apical domain usually forms only at the cell–fluid interface[41,51,52], but here its formation within the salt-and-pepper cell aggregates was necessary for directed active migration. Second, epithelialization is typically associated with stabilization during collective cell migration[53,54], whereas here the apical domain is linked with protrusion and mesenchyme-like motility, which may be due to the immaturity of the apical polarity within the ICM tissue.

We propose that PrE cell migration within the ICM could arise as a collective behaviour, in which directional guidance is conceivably provided by a tissue-level gradient of ECM that is progressively deposited by the cells. This 'breadcrumb navigation' mechanism would not require aligning interactions between neighbouring cells as in other collective migration scenarios. Cell–ECM interactions govern various aspects of tissue patterning[55] and cell migration; for example, cells in vivo enhance their migratory capacity by secreting laminin[56], and cell–matrix interactions enable cell sensing of a stiffness gradient for durotaxis[57]. Moreover, cell–ECM signalling through integrins could regulate cell fate specification[39,58,59]. Within the ICM, the ECM accumulates more towards the cavity interface as EPI/PrE sorting progresses. We propose that this gradient is formed by PrE cells themselves, as the retention of PrE cells located close enough to the fluid interface sorted by differential surface contractility breaks tissue-scale symmetry, deposition of ECM by these cells could bias its distribution towards the cavity surface. Alternatively, or in addition, the deposited ECM could also reinforce PrE cell fate via integrin signalling. This mechanism can self-organize directed collective cell migration for a certain range in space and time.

Cell sorting at the fluid interface driven by relative differences in surface tension is reminiscent of the mechanism sorting the inside and outside cells in 16-cell embryos[37]. Enrichment of aPKC at the apical cortex in blastomeres antagonizes myosin phosphorylation, which can explain decreased interfacial tension in PrE cells and their retention at the cavity interface. This mechanism couples cell fate and position, ensuring robust patterning, and notably, is conserved across two consecutive lineage segregation events in pre-implantation mouse development.

Dynamic mechanisms generating patterns within initially equivalent cell populations described here may be widespread among undifferentiated or stem cell populations, wherein stochastically variable gene-expression is evident before lineage segregation[3,60–63]. In such systems, gene regulatory networks or signalling from the niche may drive formation of distinct cell types at a certain ratio, first in a salt-and-pepper pattern, which is subsequently sorted to form a pattern within a specific geometrical context. This may present a challenge to developing or homeostatic systems, as they need to accommodate a certain proportion of cell types into varying geometries and environments. Our findings in this study have implications based on the relationship between the ratio of cell types and the geometric properties of the tissue: PrE cells must form a monolayer on the surface of the ICM of varying size and shape, while keeping a fixed PrE:EPI ratio within the ICM. Thus, patterning precision is not always compatible with scaling in tissues. We propose that in such a case, robustness in tissue patterning may be ensured by selecting and coupling optimal parameter sets in space and time; in the case of blastocyst patterning, cell fate proportions, cellular dynamics, duration of sorting, tissue size and geometry may co-adapt on evolutionary timescales to be robust against a certain degree of variability. Further investigations into mechanisms that enable coordination of these parameter changes will be valuable to gain insights into robustness of embryo development and evolution.

## Online content

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

## Methods

### Mouse work

Mouse-related animal work was performed in the Laboratory Animal Resources (LAR) facility at the European Molecular Biology Laboratory (EMBL) with permission from the Institutional Animal Care and Use Committee overseeing the operation (no. TH11 00 11) and at the Animal Facility at the Hubrecht Institute. LAR facilities operate according to the Federation for Laboratory Animal Science Associations guidelines and recommendations. At the Hubrecht animal facility, mice were housed according to institutional guidelines and procedures were performed in compliance with Standards for Care and Use of Laboratory Animals with approval from the Hubrecht Institute ethical review board. Animal experiments were approved by the Animal Experimentation Committee of the Royal Netherlands Academy of Arts and Sciences. All experimental mice were maintained in specific-pathogen-free conditions, with ambient temperature 22.5–23 °C and humidity between 50–60%, ad libitum food and water, on a 12-h light–dark cycle and used from 8 weeks of age.

### Monkey work

Monkey animal work was performed with female cynomolgus monkeys (*Macaca fascicularis*), of ages ranging between 6 to 11 years. The animals were maintained on a 12-h light–dark cycle. Each animal was fed 20 g kg$^{-1}$ body weight of commercial pellet monkey chow (CMK-1, CLEA Japan) in the morning, supplemented with 20–50 g of sweet potato in the afternoon. Water was provided ad libitum. Animals were housed with temperature and humidity maintained at 25 ± 2 °C and 50 ± 5%, respectively. The animal experiments were appropriately performed by following the Animal Research: Reporting in Vivo Experiments guidelines developed by the National Centre for the Replacement, Refinement & Reduction of Animals in Research and also by following 'The Act on Welfare and Management of Animals' from Ministry of the Environment, 'Fundamental Guidelines for Proper Conduct of Animal Experiment and Related Activities in Academic Research Institutions' under the jurisdiction of the Ministry of Education, Culture, Sports, Science and Technology, and 'Guidelines for Proper Conduct of Animal Experiments' from Science Council of Japan. All animal experimental procedures were approved by the Animal Care and Use Committee of Shiga University of Medical Science (approval no. 2021-10-4).

### Mouse lines and genotyping

The following mouse lines were used in this study: C57BL/6×C3H F$_1$ hybrid as wild-type, mTmG[36], *Pdgfra*[H2B-GFP] (ref. [33]), *Prkci*[tm1.1Kido], *Prkcz*[tm1.1Cda] (ref. [64]), *R26-creER*[35], *R26-H2B-mCherry*[34], *Myh9*[tm5Rsad] (ref. [65]), *Rac1*[flox/flox] (refs. [66,67]), *Lamc1*[tmStrl(floxed)] (ref. [68]) and *Zp3-cre*[69]. To generate *Myh9*[+/−] mice, *Myh9*[flox/flox] females were crossed with *Zp3-cre*[tg/+] males. *Prkci*[+/−]*Prkcz*[−/−] mice were generated by mating *Prkci*[flox/flox] *Prkcz*[−/−] females with *Prkcz*[−/−]*Zp3-cre*[tg/+] males. *Rac1*[flox/flox] females were crossed with *Zp3-cre*[tg/+] males to generate *Rac1*[+/−] mice. Standard tail genotyping procedures were used to genotype transgenic mice (for primers and PCR product sizes, see Supplementary Table 1). *Prkci*[+/−]*Prkcz*[−/−] embryos were generated by crossing *Prkci*[+/−]*Prkcz*[−/−] females with *Prkcz*[−/−] males. *Rac1*[+/−], *Rac1*[−/−] and *Rac1*[+/+] embryos were obtained by mating *Rac1*[+/−] females with *Rac1*[+/−] males. *Myh9*[+/−] embryos were obtained by mating *Myh9*[+/−] females with wild-type males. To obtain *Lamc1*[+/−] mice, *Lamc1*[tmStrl(floxed)]*Zp3-cre*[tg/+] females were crossed with wild-type males. To obtain zygotic *Lamc1*[−/−] embryos, *Lamc1*[+/−] females were crossed with *Lamc1*[+/−] males.

### Single-embryo genotyping

Transgenic mutant embryos were genotyped retrospectively after imaging. Single embryos were transferred into individual PCR tubes containing 10 µl lysis buffer composed of PCR buffer (Fermentas, EP0402) supplemented with 0.2 mg ml$^{-1}$ proteinase K (Sigma, P8811), followed by incubation at 55 °C for 1 h and then 96 °C for 10 min. Then,

3–4 µl of the resulting lysate containing gDNA was mixed with the relevant primers (Supplementary Table 1) for PCR genotyping.

PCR products were mixed with 6× loading dye (Life Technologies, R0611) and were separated by electrophoresis in 1–1.2% (w/v) agarose gel (Lonza, 50004) supplemented with 0.03 µl ml$^{-1}$ DNA staining dye (Serva, 39804.01) in TAE buffer. DNA fragments were visualized under ultraviolet light on a video-based gel documentation system (Intas, GEL Stick 'Touch') and fragment lengths were measured against a standardized DNA ladder (Life Technologies, SM0323 and SM0313).

### Intracytoplasmic sperm injection into monkey oocytes

Monkey oocyte collection was performed as described previously[70]. In brief, 2 weeks after the subcutaneous injection of 0.9 mg gonadotropin-releasing hormone antagonist (Leuplin for Injection kit, Takeda Chemical Industries), a micro-infusion pump (iPRECIO SMP-200, ALZET Osmotic Pumps) with 15 IU kg$^{-1}$ human follicle-stimulating hormone (hFSH; Gonal-f; Merck Biopharma) was embedded subcutaneously under anaesthesia and injected 7 µl h$^{-1}$ for 10 days. After the hFSH treatment, 400 IU kg$^{-1}$ human chorionic gonadotropin (hCG; Gonatropin, Asuka Pharmaceutical) was injected intramuscularly. Forty hours after the hCG treatment, oocytes were collected by follicular aspiration using a laparoscope (Machida Endoscope, LA-6500). Cumulus-oocyte complexes (COCs) were recovered in alpha modification of Eagle's medium (MP Biomedicals), containing 10% serum substitute supplement (Irvine Scientific). The COCs were stripped off cumulus cells with 0.5 mg ml$^{-1}$ hyaluronidase (Sigma Chemical). ICSI was carried out on metaphase II (MII)-stage oocytes in mTALP containing HEPES with a micromanipulator. Fresh sperm were collected by electric stimulation of the penis with no anaesthesia. Following ICSI, embryos were cultured in CMRL 1066 Medium (Thermo Fisher Scientific) supplemented with 20% FBS at 38 °C in 5% CO$_2$ and 5% O$_2$.

### Embryo recovery and in vitro culture

To obtain pre-implantation embryos at different stages, female mice were super-ovulated by intraperitoneal injection of 7.5 IU pregnant mare's serum gonadotropin (Intervet, Intergonan) followed by 7.5 IU of human chorionic gonadotropin (hCG; Intervet, Ovogest 1500) 48 h later, and mated with males. Embryos were collected either 68 h post-hCG injection for uncompacted 8-cell stage or 96 h post-hCG injection for the 64-cell blastocyst stage, considered as the E3.5 stage. Recovery of embryos was performed under a stereomicroscope (Zeiss, StreREO Discovery.V8) equipped with a thermoplate (Tokai Hit) heated to 37 °C. Oviducts and uterine horns were dissected and submerged in global embryo culture medium containing HEPES (LifeGlobal, LGGH-050) in 1.5-ml Eppendorf tubes. They were then laid on a 35-mm Petri dish (Falcon, 351008) under a stereomicroscope and embryos were flushed using a flushing needle attached to a 1-ml syringe filled with global medium and HEPES. Embryos were washed, transferred to 10-µl drops of global medium (LifeGlobal, LGGG-050) covered with mineral oil (Sigma, M8410) on a Petri dish and cultured in a CO$_2$ incubator (Thermo Scientific, Heracell 240i) at 37 °C with 5% CO$_2$.

### Immunosurgery

The zona pellucida was removed from blastocysts with pronase (0.5% w/v proteinase K, Sigma P8811, in global medium containing HEPES supplemented with 0.5% PVP-40, Sigma, P0930) treatment for 2–3 min at 37 °C. Blastocysts were washed in 10-µl droplets of global medium (LifeGlobal, LGGG-050). To isolate the ICM, blastocysts were incubated in serum containing anti-mouse antibody (Cedarlane, CL2301, lot no. 049M4847V) diluted 1:3 with global medium for 30 min at 37 °C. Following 2–3 brief washes in Global medium with HEPES, embryos were incubated in guinea pig complement (Sigma, 1639, lot no. SLBX9353) diluted with global medium in a 1:3 ratio for 30 min at 37 °C. Lysed outer cells and remaining debris were removed by gentle pipetting with a narrow glass capillary (Brand, 708744) to isolate the ICM. The isolated ICMs were cultured in 10-µl drops of global medium in a Petri

dish (Falcon, 351008) covered with mineral oil (Sigma, M8410) and incubated at 37 °C with 5% $CO_2$ for up to 24 h.

## Embryo size manipulation

Embryos were recovered at the 8-cell stage and the zona pellucida was removed as described above. For generating small-sized embryos, uncompacted 8-cell stage morulae were dissociated into the desired fraction of blastomeres by incubation in KSOM without $Ca^{2+}$ and $Mg^{2+}$ (ref. 41) for 5 min at 37 °C, followed by gentle pipetting through a narrow, glass capillary (Brand, 708744). For generating large-sized embryos, the desired number of embryos were aggregated at the uncompacted 8-cell stage in a single microdroplet of global medium under mineral oil (Sigma, M8410). Embryo aggregation was encouraged by placing embryos in contact in micro-indented wells in 35-mm Petri dishes (Falcon, 351008), ensuring that the embryos adhered to each other without drifting apart. Size-manipulated embryos were cultured until E4.5, when a clear blastocyst cavity was discernible. Embryos that failed to aggregate were discarded from further analysis.

## Generation of chimeric embryos

Chimeric embryos were made using genetic mutant embryos from $Prkci^{+/-}Prkcz^{-/-}$, $Lamc^{-/-}$ and $Myh9^{+/-}$, and mTmG embryos to distinguish the knockout cells against the wild-type background. To make chimeras of $Prkci^{+/-}Prkcz^{-/-}$ with mTmG and $Myh9^{+/-}$ with mTmG, uncompacted 8-cell stage embryos from each were aggregated together into micro-indented wells in 35-mm Petri dishes (Falcon, 351008) to make 2× chimeras. Corresponding chimeras of B6C3F1 embryos with mTmG were used as controls. The microwells were made in 10 µl global medium droplets covered with mineral oil (Sigma, M8410) for 48-h in vitro culture until the E4.5 stage. The chimeric embryos or isolated ICMs were fixed at E4.5 and immunostained for cell fate markers and RFP/tdTomato. The embryos/ICMs were imaged and then genotyped retrospectively using the appropriate primers. For the rescue experiments with chimeras of $Lamc1$ embryos with mTmG, 4-cell stage embryos from each were first dissociated into blastomeres, and three blastomeres each of $Lamc1^{-/-}$ and mTmG were aggregated together to make normal-sized chimeric embryos. The remaining blastomere from the $Lamc1$ embryo was used for single-embryo genotyping to distinguish the genotypes of the chimeras. For comparison with the chimeras, $Lamc1^{-/-}$ blastocysts were used as controls.

## Generation of large embryos with mosaic-labelled cells

Wild-type and fluorescent embryos expressing reporters $Pdgfra^{H2B-GFP}$; mTmG were recovered at the uncompacted right-cell stage and used for making chimeras. For aggregation, each fluorescent embryo was combined with two wild-type embryos in micro-indented wells in 35-mm Petri dishes (Falcon, 351008). The microwells were made in 10 µl global medium droplets covered with mineral oil (Sigma, M8410) for 24-h in vitro culture until the E3.5 blastocyst stage. Embryos that formed successful aggregates and showed a singular, expanded blastocyst cavity were chosen and screened for mosaic labelling of cells and used for further live-imaging and analysis.

## Generation of mosaic-labelled ICMs

Embryos were recovered from a cross between mTmG and $R26$-creER mouse lines at the uncompacted 8-cell stage. After zona pellucida removal, embryos were incubated in 10 µM 4-hydroxytamoxifen in global medium for 10 min at 37 °C for tamoxifen-induced Cre-$loxP$ recombination. The embryos were washed five or six times in global medium and cultured for 24 h. At the E3.5 stage, the embryos were screened for sparse conversion of mT to mG under an inverted Zeiss Observer Z1 microscope with a CSU-X1M 5000 spinning disc unit and selected for further experimental procedures. Immunosurgery was performed at the E3.5 stage on the selected embryos and the isolated ICMs were used for live-imaging.

## Micropipette aspiration

Micropipette aspiration was performed as described previously[71] to measure surface tension of ICM cells. In brief, a microforged micropipette coupled to a microfluidic pump (Fluigent, MFCS) was used to measure the surface tension of ICM cells. Micropipettes were prepared from glass capillaries (Warner Instruments, GC100T-15) using a micropipette puller (Sutter Instrument, P-1000) and a microforge (Narishige, MF-900). A fire-polished micropipette with diameter ~7–8 µm was mounted on an inverted Zeiss Observer Z1 microscope with a CSU-X1M 5000 spinning disc unit, and its movement was controlled by micromanipulators (Narishige, MON202-D). Samples were maintained at 37 °C with 5% $CO_2$. A stepwise increasing pressure was applied on ICM surface cells using the microfluidic pump and Dikeria software (LabVIEW), until a deformation with the same radius as that of the micropipette ($R_p$) was reached. The equilibrium aspiration pressure ($P_c$) was measured, images were acquired in this configuration and then the pressure was released. Care was taken to avoid aspirating cell nuclei. At steady state, the surface tension $\gamma$ of the cells is calculated based on Young–Laplace's law: $\gamma = P_c/2(1/R_p - 1/R_c)$, in which $P_c$ is the net pressure used to deform the cell of radius $R_c$. Image analysis and measurement of the pipette radius $R_p$ and $R_c$ was conducted in Fiji and calculation of surface tension was conducted in Python v.3.9.

## Microbeads experiments

Protein-A-coated PMMA microbeads (Microparticles, PMMA-Protein-A-S4040) 12 µm in diameter, were used for bead implantation in the blastocyst ICM. To coat microbeads with CDH1 (E-cadherin) and laminin, recombinant mouse CDH1-Fc chimera protein (Sigma, E2153) was reconstituted at 100 mg ml$^{-1}$ in sterile PBS, and laminin (Sigma, L2020) was reconstituted at 1 mg ml$^{-1}$ in PBS. The microbeads were washed with cold PBS and incubated in 4 µg ml$^{-1}$ CDH1-Fc solution for 4 h at 4 °C with 1,400 rpm mixing (Thermomixer, Eppendorf), washed with PBS and incubated in 50 µg ml$^{-1}$ laminin solution overnight at 4 °C with 1,400 rpm shaking. The microbeads were resuspended in global medium (LifeGlobal, LGGG-050) before aggregating with uncompacted 8-cell stage embryos. For aggregation, two embryos were put together with 8–10 coated microbeads in a Petri dish with indented microwells, and cultured until E4.5. To check whether the beads were incorporated, blastocysts were screened under a stereomicroscope and those with successfully implanted beads were fixed with 4% paraformaldehyde (PFA) and immunostained. Embryos where the bead was implanted outside the ICM or at the PrE layer in the ICM were excluded. An aliquot of the coated PMMA microbeads was fixed with 4% PFA, washed and immunostained using an antibody against E-cadherin to validate protein coating on the surface of the microbeads. Laminin coating was first validated with immunostaining of PMMA microbeads without Protein-A (Microparticles, PMMA-R-12.4) as the Fc region of immunostaining antibodies binds Protein-A and then identical coating conditions were used for laminin coating of Protein-A PMMA microbeads.

## Pharmacological inhibition

Latrunculin B (Sigma, 428020) was reconstituted in dimethylsulfoxide (DMSO) at a stock concentration of 100 mM and a final concentration of 1 µM. CK-666 (Sigma, 182515) was resuspended in DMSO at 25 mM and a working concentration of 2 µM was used. NSC23766 (Sigma, SML0952) was resuspended in DMSO at a stock concentration of 10 mM and working concentrations of 50 µM, 100 µM and 200 µM. Gö6983 (Sigma, 365251) was resuspended in DMSO at 10 mM and a final concentration of 5 µM. For working concentrations of the inhibitors, respective stock concentrations were diluted in global medium. Embryos or isolated ICMs were incubated with the appropriate working concentrations of latrunculin B, CK-666, Gö6983 or NSC23766 and corresponding controls in µ-Slide chambered coverslips (Ibidi, 81506) either for live-imaging or in vitro culture, before fixation in 4% PFA (see 'Immunofluorescence staining' section).

## Immunofluorescence staining

Mouse embryos or isolated ICMs were fixed in 4% PFA (Sigma, P6148) at room temperature for 15 min. Fixed embryos were washed three times for 5 min each in wash buffer DPBS-Tween containing 2% BSA (Sigma, A3311) and permeabilized at room temperature for 20 min in permeabilization buffer 0.5% Triton-X in DPBS (Sigma, T8787). After permeabilization, samples were washed, followed by incubation in blocking buffer DPBS-Tween20 (Sigma, P7949) containing 5% BSA, either overnight at 4 °C or for 2 h at room temperature. Blocked samples were then incubated with the desired primary antibodies overnight at 4 °C, washed and incubated in fluorophore-conjugated secondary antibodies and dyes at room temperature for 2 h. Stained samples were washed and incubated in 4,6-diamidino-2-phenylindole (DAPI) solution (Invitrogen, D3571; diluted 1:1,000 in DPBS) for 10 min at room temperature. Samples were then transferred into individual droplets of DPBS covered with mineral oil on a 35-mm glass-bottom dish (MatTek, P35G-1.5-20-C) for imaging. Primary antibodies against GATA6 (R&D Systems, AF1700), GATA4 (R&D Systems, BAF2606), SOX2 (Cell Signaling, 23064), biphosphorylated myosin regulatory light chain (ppMRLC) (Cell Signaling, 3674), E-cadherin (Sigma, U3254) and laminin (Novus Biologicals, NB300-14422) were diluted at 1:200. Primary antibodies against NANOG (ReproCell, RCAB002P-F), PKCλ (Santa Cruz Biotechnology, sc-17837), PKCζ (Santa Cruz Biotechnology, sc-17781), Integrinβ1 (Millipore, MAB1997), active integrinβ1 (9EG7, BD Bioscience, 553715) and RFP/tdTomato (Rockland, 600-401-379 and Chromotek, 5f8) were diluted 1:100. Secondary antibodies donkey anti-goat IgG Alexa Fluor 488 (Invitrogen, A11055), donkey anti-rabbit IgG Alexa Fluor 546 (Invitrogen, A10040), donkey anti-mouse IgG Alexa Fluor 555 (Invitrogen, A31570), donkey anti-rabbit IgG Alexa Fluor 647 (Invitrogen, A31573), donkey anti-mouse Cy5 (Jackson ImmunoResearch, 715-175-150) and donkey anti-rat Cy5 (Jackson ImmunoResearch, 712-175-153) were used at 1:200 dilution. Immunofluorescence samples were imaged on a Zeiss LSM 880 microscope with AiryScan Fast mode. A ×40 water-immersion Zeiss C-Apochromat 1.2 NA objective was used and raw AiryScan images were acquired and processed using ZEN black software (Zeiss).

Monkey embryos that successfully developed to blastocysts were fixed between day 7–8 post-ICSI in 4% PFA (Wako 166-23251) in DPBS for 15 min at room temperature, permeabilized in DPBS with 0.5% Triton-X-100 (Nacalai, 12967-32) for 30 min at room temperature, and blocked overnight at 4 °C in DPBS with 3% BSA (Sigma, A9647) and 0.05% Triton-X-100. Embryos were then transferred into primary antibody solution in blocking buffer and incubated overnight at 4 °C. Primary antibodies against Alexa Fluor 647-conjugated OCT3/4 (Santa Cruz, sc-5279 AF647) and GATA4 (Cell Signaling, 36966S) were diluted 1:200. Embryos were then washed four times for 5 min in blocking buffer and transferred into secondary antibody solution in blocking buffer for 2 h at room temperature. Secondary antibody conjugated with Alexa Fluor Plus 488 against rabbit IgG (Thermo Fisher Scientific, A32790) was diluted 1:200. Alexa Fluor Plus 555 Phalloidin (Thermo Fisher Scientific, A30106) and DAPI (Thermo Fisher Scientific, D3571) were added in the secondary antibody solution diluted 1:400. Embryos were mounted in 1-μl drops of DPBS for imaging. Imaging of immunostained monkey embryos was performed with LSM 980 (Zeiss) with Airyscan 2 Multiplex CO-8Y mode. An LD LCI Plan-Apochromat ×25/0.8 water-immersion objective (Zeiss) was used.

## Time-lapse imaging

Embryos or ICMs were placed into global medium drops covered with mineral oil on a glass-bottom imaging dish (MatTek, P50G-1.5-14-F). For drug treatment experiments, embryos were placed in 60 μl global medium supplemented with inhibitor in 15-well glass-bottom dishes (Ibidi, 81501). Time-lapse imaging of live, fluorescent samples was performed on an inverted Zeiss Observer Z1 microscope with a CSU-X1M 5000 spinning disc unit. Excitation was achieved using 488 nm and 561 nm laser lines through a 63/1.2C Apo W DIC III water-immersion objective. Emission was collected through 525/50 nm, 605/40 nm, band pass filters onto an EMCCD Evolve 512 camera. Images were acquired every 20 min for up to 12 h. The microscope was equipped with a humidified incubation chamber to keep the sample at 37 °C and supply the atmosphere with 5% $CO_2$.

## Confocal live-imaging

For counting cell numbers before and after immunosurgery, zona-removed E3.5 embryos were incubated in global medium with 5 μg ml$^{-1}$ Hoechst 33342 (Invitrogen, H21492) for 10 min at 37 °C. The embryos were washed and mounted in global medium drops covered with mineral oil on 35-mm glass-bottom dishes (MatTek, P35G-1.5-20-C). A full confocal z-stack was obtained for each blastocyst on the LSM 880 confocal microscope, with samples maintained in the humidified incubation chamber at 37 °C and 5% $CO_2$. Images were acquired with Airyscan Fast mode. Next, immunosurgery was performed, and confocal z-stacks of the isolated ICMs were acquired after immunosurgery with identical imaging conditions.

Large-sized blastocysts with mosaic-labelled cells were transferred to individual 10 μl global medium drops covered with mineral oil on 35-mm glass-bottom dishes (MatTek, P35G-1.5-20-C). The imaging dish was mounted on a Zeiss LSM 880 microscope. Embryos were maintained in a humidified chamber at 37 °C and atmosphere was supplemented with 5% $CO_2$. Confocal z-stacks were obtained at 20 min intervals for up to 24 h.

## Nuclear detection and tracking in isolated ICMs

Nuclear detection and tracking of cell centres in isolated ICMs was performed with a semi-automatic analysis pipeline developed previously[31]. In brief, centres of all nuclei were detected from time-lapse images using a difference of Gaussians (DoG) algorithm[72] using the nuclear fluorescence signal. Using the high-performance computing cluster at EMBL, the best parameters for the DoG algorithm were found by a grid-search procedure, which explored thousands of different configuration parameters simultaneously. The best output was manually curated and used for cell tracking with a nearest-neighbour algorithm[31]. Manual curation from the E3.5 to E4.0 stage of the ICM was performed and validated using the software Mov-IT by one operator. Cell fate was assigned based on the $Pdgfra^{H2B-GFP}$ fluorescence intensity. All the cells in the ICM were inspected and cells that could not be traced with confidence (<1% cells) were excluded from the lineage trees.

## Evaluation of cell dynamics in isolated ICMs

Directionality of cell movements from the four-dimensional live-imaging datasets of isolated ICMs was analysed using cell-tracking information. Cell positions in 3D were obtained for each time point using the DoG and nearest-neighbour algorithm as mentioned previously. 3D cell tracking was converted into one-dimensional radial cell positions. For this purpose, first, the geometric centroid of the ICM was calculated for each time point as the average of the $x$, $y$, $z$ coordinates of the ICM cells. Next, radial distances of cells were calculated from the centroid at each time point using the Euclidean distance formula. Cell displacements were calculated for both EPI and PrE cells as the difference in radial position between two consecutive time points. Positive displacements along the radial axis were considered as outward movement and negative displacements as inward movement. Displacements were binned according to radial cell position and time.

## Simulations of Poissonian cellular Potts models

A CPM of the ICM system was constructed over a grid of voxels with resolution 1 μm³. As described in a companion paper[38], the time evolution of the system was implemented as a Poissonian process, which explicitly introduces the physical time into cellular Potts simulations through state-transition rates determined by the cellular Potts Hamiltonian

and novel kinetic parameters controlling the diffusive mobility of cells. Computer simulations were carried out with a discrete time step of 0.1 min. To prevent cell fragmentation, we adopted the approach described previously[73].

The Hamiltonian of our CPM reads $E = \Sigma_{ij} J_{ij}/2 + (\kappa/2) \Sigma_c (V_c - \bar{V}_c)^2$, in which the first term sums over all pairs of voxels $i$ and $j$, and the second term runs over individual cells $c$. The symmetric coefficients $J_{ij} = J_{ji}$ vanish when the voxels $i$ and $j$ are not within each other's Moore neighbourhood[73]. These coefficients assume the values listed in Supplementary Table 2 depending on the type of the voxels $i$ and $j$ – PrE, EPI or medium. As previously described[38], the constants $J_{medium:EPI}$ and $J_{medium:PrE}$ are chosen to correspond to the maximum and minimum surface tensions observed experimentally between medium and EPI cells and between medium and PrE cells, respectively. Smaller surface tension differences led to lower sorting scores than those observed in the experiments, further supporting the presence of additional sorting mechanisms, as we discuss in the main text.

The total area of cells is not constrained in our simulations, in agreement with the actomyosin cortex being the main determinant of cellular shape on the relevant timescales[74,75]. The Poissonian transition rates are determined by kinetic action rates parameters $\alpha$, which control frequency of updates for one of the three voxel types (medium, EPI and PrE) as described previously[38].

The cells' preferred volume grows linearly as $\bar{V}_c(t) = \bar{V}_c(0) + g\,t$ with a constant rate $g$ until a target division volume is reached. The cell is then divided into two daughter cells along the plane perpendicular to the longest axis of the cell's gyration tensor. The target division volume is sampled from an empirical distribution constructed from experimental volumes of mitotic cells: 2,406.63, 2,428.09, 2,455.23, 2,517.92, 2,994.28, 3,002.65, 3,110.31, 3,116.73, 3,294.93 and 4,133.67 $\mu m^3$.

To model the ECM component, we introduced an additional type of medium, which was actively produced by PrE cells at cell–cell contacts. Specifically, when the target neighbourhood of a voxel occupied by another cell contains a PrE cell, we add to the target values the one which represents the ECM. The parameters of action rate and surface tension are chosen identical to that of PrE cells: $\alpha_{ECM} = \alpha_{PrE}$, $J_{medium:ECM} = J_{medium:PrE}$, $J_{EPI:ECM} = J_{EPI:PrE}$, $J_{PrE:ECM} = J_{PrE:PrE}$. The production of ECM by PrE cells is implemented through active exponents[38] $\phi$(EPI, ECM) = $4 \times 10^5\,k_B T$ and $\phi$(PrE, ECM) = $4 \times 10^5\,k_B T$ to promote the transition probability of EPI and PrE voxels respectively into ECM, when the latter is among the target values. Each ECM voxel bears an additional energy cost $\varepsilon$ to enhance its degradation and the ECM regions are allowed to fragment and vanish (local domain connectivity is not enforced)[38,73], such that ECM voxels diffuse and disappear over time. Supplementary Table 2 summarizes all the numerical parameters of our simulations. For further details, see ref. 38.

## Estimation of first-passage probabilities
To characterize directionality of cell motion from tracking data in mosaic-labelled blastocysts, we measured distance from their respective geometric centres to the lumen surface over time. From these measurements, we calculated first-passage probabilities following the approach described previously[76]. In brief, we consider the total number of events in which a cell moves from an initial position $X_0$ by 3 $\mu m$ either away from or toward the cavity. The fraction of cases where the motion was directed toward the lumen is reported in the graph as a function of $X_0$. To exclude the effect of correlations when extracting the first-passage from a single-cell trajectory, we ensured that samples of $X_0$ were separated by at least 3 min.

## Image analysis
**Cell counting.** Cell counting from immunofluorescence images of both blastocysts and isolated ICMs was carried out using Imaris v.9.7.2 (Bitplane). The Spots module was used to detect all nuclei from DAPI signal. Estimated nucleus diameter was set to 7 $\mu m$, the automated spot

detection algorithm was used to detect all cells, followed by manual validation. Cells were classified as either EPI or PrE based on the fluorescence intensities of transcription factors NANOG and GATA6 until stage E4.0, and transcription factors SOX2 and GATA4 beyond stage E4.0, respectively.

## Estimation of sorting scores
For live-imaging datasets, EPI and PrE cell positions were obtained in 3D from the nuclear detection and tracking as described previously. For immunofluorescence images, cell positions of EPI and PrE were obtained in 3D after nuclear detection using the Spots module on immunofluorescence images in Imaris. The geometric centroid of isolated ICMs was calculated as an average of the $x$, $y$ and $z$ coordinates of all cells in the ICM. The radial distance of ICM cells from the centroid was calculated as the Euclidean distance between the 3D coordinates of cells and the ICM centroid. Next, the sorting score was calculated as the average of the sign of the pair-wise difference between EPI and PrE radial distances.

## Unbiased classification of cell fates
Unbiased classification of cell fates into EPI and PrE from immunofluorescence images of isolated ICMs between stages E3.5–E4.0 was carried out using $k$-means clustering in Python v.3.9 with the scikit-learn v.1.0.1. Imaging datasets were generated using identical immunostaining and imaging conditions. Fluorescence intensities of DAPI and transcription factors GATA6 and NANOG were obtained after spot detection in Imaris. To compensate for fluorescence intensity decay with image depth, fluorescence intensities were log-transformed and a linear regression was fitted to the log values as a function of $z$-depth. Next, a $k$-means clustering algorithm was performed on the $z$-corrected log intensity values of GATA6 and NANOG to classify cell fates.

## Cell tracking of mosaic-labelled cells
Cell tracking of mosaic-labelled cells was performed in Fiji. The cell centre was determined by fitting an ellipse to the membrane signal of the cell of interest in the equatorial plane of the cell at each time point. Distance of the cell from the cavity interface was measured by dropping a normal to the cavity surface from the centre of the cell at each time point, and measuring the length of this line segment. Cavity interface was visualized from brightfield.

## Analysis of PrE cell protrusions
Length of PrE cell protrusions was measured in Fiji, where a line segment was drawn from the centre of the nucleus to the tip of the protrusion. Protrusions were defined as membrane deformations that are first extended and then retracted in subsequent time points. Angle of the protrusion with respect to the cavity was measured in Fiji as the angle between two line segments, one from the centre of the cell to the tip of the protrusion, and the other from the centre of the cell, normal to the cavity surface.

## Analysis of membrane curvature and laminin localization
Membrane curvature and laminin intensity along the cell membrane was measured in Fiji. The outline of PrE cells was manually traced along the actin immunofluorescence signal in the cell equatorial plane using the segmented polyline tool. Laminin intensity was measured along this polyline. To calculate curvature for each pixel along the cell contour, two neighbouring points were chosen at a distance of 10 pixels each, creating a segment of 20 pixels in total. Local membrane curvature was calculated for this segment as the inverse of the radius of the circle that fits the three points, spanning a length of 2–3 $\mu m$. Curvature was considered to be zero if the three pixels were collinear.

To evaluate the correlation between cell shape asymmetry and laminin localization, the cell boundary was binned into six equal length intervals oriented along the cavity–TE axis. The average membrane

curvature and average laminin intensity were calculated for each bin and the bins with the highest average curvature and highest average laminin intensity were plotted for each cell in polar plots.

## Quantification of ectopic cells in blastocysts

The number of ectopic cells was determined using Imaris 3D visualization. Ectopic EPI cells in E4.5 blastocysts or ICMs were defined as SOX2$^+$:GATA4$^-$ cells located at the ICM–fluid interface. Ectopic PrE cells in E4.5 blastocysts or ICMs were defined as SOX2$^-$:GATA4$^+$ cells entirely lacking a contact-free surface. For quantification of ectopic cells in size-manipulated mouse blastocysts, small-sized blastocysts without an ICM were excluded from the analysis.

## Evaluation of cell shape

Cell shape analysis to estimate aspect ratio and circularity of cells was performed using Fiji. With the freehand selection tool, EPI and PrE cell outlines were traced manually in the equatorial plane of the cell. The cell shape descriptors such as aspect ratio and circularity were measured in Fiji.

## Analysis of embryo chimeras

In chimeric ICMs made from wild-type (mTmG) and *Prkci*$^{+/-}$*Prkcz*$^{-/-}$ embryos, cells within the ICM were first detected using the Spots module in Imaris. Based on fluorescence signal, mT$^+$ and mT$^-$ cells were classified as wild-type and aPKC knockout, respectively. ICM cells were also classified as either 'surface' or 'inner' depending on whether they were in contact with the fluid medium. The proportion of mT$^+$ surface cells out of all surface cells was then computed for both chimeric combinations.

In chimeric blastocysts made from wild-type (mTmG) and *Lamc1*$^{-/-}$ embryos, cells within the ICM were detected using the Spots module in Imaris. Based on membrane fluorescence signal, mT$^+$ and mT$^-$ cells were classified as wild-type and *Lamc1*$^{-/-}$, respectively. The number of ectopic *Lamc1*$^{-/-}$ PrE cells were counted as mT$^-$ cells that are not part of the PrE monolayer. For the control group, ectopic PrE cells were counted in *Lamc1*$^{-/-}$ blastocysts published previously[39].

In chimeric blastocysts made from wild-type (mTmG) and *Myh9*$^{+/-}$ embryos, cells within the ICM were first detected using the Spots module in Imaris. Next, based on fluorescence signal, mT$^+$ and mT$^-$ cells were classified as wild-type and *Myh9*$^{+/-}$, respectively. Using Imaris 3D visualization, GATA4$^-$:mT$^-$ cells at the ICM–fluid interface were identified as ectopic EPI cells originating from the *Myh9*$^{+/-}$ population.

## Fluorescence intensity measurements

ppMRLC intensity analysis was performed in Fiji. A freehand line was manually traced along the outer cell surface of E3.5 ICM cells and mean fluorescence intensity of ppMRLC was measured along this curve in an equatorial section of the ICM. Next, the ppMRLC signal was normalized by nuclear DAPI intensity of the cell.

Quantification of aPKC fluorescence intensity was performed using Fiji. A line segment was manually drawn for E3.5 ICM cells from the inner edge to the outer edge of the cell along the radial direction. The inner edge was defined as the region of the cell towards the ICM centroid, whereas the outer edge of the cell was defined as the region of the cell facing the ICM–fluid interface. Fluorescence intensity of aPKC was measured along this line segment, with segment width set to ten pixels. The fluorescence intensity was binned into a fixed number of intervals to normalize cell length. Polarization index was calculated as the ratio between mean aPKC fluorescence intensity over one-quarter distance from outer edge and mean aPKC intensity over one-quarter distance from the inner edge.

Laminin intensity analysis was performed in Fiji. For quantification of laminin distribution around individual cells in ICMs, laminin intensity was measured along the cell boundary of EPI and PrE precursors in the equatorial plane of the ICM. To quantify the laminin gradient

in 3× blastocysts, a maximum intensity projection of a section of the ICM was obtained and the average laminin fluorescence intensity was measured along a line segment of width ten pixels, drawn from the ICM–TE interface to the ICM–cavity interface. The measurements were binned into a fixed number of intervals to normalize for ICM length. The intensity measurements were normalized by the maximum intensity and smoothed using a rolling average. To quantify the laminin gradient in 3× isolated ICMs, a line segment of width ten pixels was traced along the radial axis in the equatorial plane of the ICMs. Average intensity was measured from the centre to the outer surface of the ICM, normalized by the maximum intensity and smoothed using a rolling average.

Colocalization analysis of integrinβ1 and laminin was performed in Fiji using the Colocalization plugin. Cells of interest were cropped out of whole blastocyst images and the integrinβ1 and laminin signals were thresholded to remove background signal. Thresholded images were used as input for the plugin to calculate Mander's coefficients and Pearson's correlation coefficient for signal overlap.

## Estimation of blastocyst volume

For estimating total blastocyst volume of mouse, monkey and human embryos, the major and minor axis of the blastocyst was measured in Fiji using the 'Fit ellipse' tool in the equatorial plane of the blastocyst. The mean of the major and minor axis was considered as the average diameter $d$, and the volume was estimated as the volume of a sphere with radius $r = d/2$.

## Estimation of ICM–fluid surface area and ICM volume

For size-manipulated mouse embryos, monkey embryos and human embryos, physical dimensions of the ICM were measured in Fiji from immunofluorescence images of blastocysts. In size-manipulated mouse blastocysts, small-sized blastocysts without an ICM and those without an ICM–cavity fluid interface were excluded from the analysis. First, a cross-section of the ICM was generated in two dimensions by reslicing 3D confocal images. The ICM base diameter was measured as the length of the line segment joining the extreme tips of the ICM where the furthest PrE cells are in contact with the trophectoderm, with ICM radius $R$ as half the length of the diameter. The major height $H$ of the ICM was measured as the largest perpendicular distance of the ICM–polar trophectoderm interface from the ICM base diameter. The minor height $h$, or specifically, elevation of the ICM–fluid interface from the ICM base diameter, was measured as the largest perpendicular distance of the ICM–fluid interface from the ICM diameter.

For mouse embryos, the volume $V$ of the ICM was measured as the difference between the volumes of two spherical caps, one with base radius $R$ and height $H$, the other with base radius $R$ and height $h$. The ICM–fluid interface area ($A_{Interface}$) was measured as the surface area of the spherical cap with base radius $R$ and height $h$. The mathematical equations used for calculating volume and surface area of a spherical cap with base radius $r$ and height $h$ are as follows:

$$\text{Volume} = \frac{1}{6}\pi h(3r^2 + h^2)$$

$$\text{Surface area} = \pi(r^2 + h^2)$$

To obtain the interface area of a hemispherical PrE in a spherical ICM as a function of cell number $N$, we assumed each cell to contribute an average cell volume $c$ so that the ICM volume is given by $V = cN$. Using the area–volume relationship of a hemisphere, we express the interface area in terms of cell number as

$$A = 3^{2/3}(\pi/2)^{1/3}(cN)^{2/3},$$

and estimate the cell volume $c$ by fitting the volume measurements across size-manipulated embryos with the relation $V = cN$. The best-fit value of $c$ is 1,907 ± 53 μm$^3$.

$A_{PrE}$ was calculated as the product of PrE cell apical area and the number of PrE cells in the ICM. Variability in PrE cell stretching gives rise to an interval of maximal and minimal $A_{PrE}$.

For analysis of ICM parameters in human blastocysts, we used a previously published image dataset from ref. 47, and did not acquire data from new human embryos in this study.

## Estimation of PrE cell apical areas

PrE cell apical areas from mouse, monkey and human embryos were measured using immunofluorescence images of blastocysts in Napari. In brief, freehand lines were manually traced with the Labels tool along the cell–fluid interface in individual PrE cells from confocal $z$-stacks based on the membrane or actin signal. With the label-interpolator plugin, a surface contour was constructed for individual cells from the traced lines. The number of pixels in the surface contour were measured to estimate the cell apical area.

## Estimation of PrE monolayer, gap and multilayer probabilities

The interval of typical values for the PrE area $A_{PrE}$ were estimated using the 10th and 90th percentiles of the PrE cells apical surface area $A_{10\%}$ and $A_{90\%}$, respectively. For a given total number $n$ of EPI and PrE cells in the ICM, the minimal $A_{PrE}$ was estimated as $fnA_{10\%}$ and maximal $A_{PrE}$ was estimated as $fnA_{90\%}$, in which $f$ is the fraction of PrE cells. The coefficient values $fA$ for minimal and maximal PrE area with zero offset plotted in the graphs are summarized in Supplementary Table 3 below. The fraction of PrE cells $f$ for mouse, monkey, and human embryos was calculated by a linear fit between number of PrE cells in the ICM versus total number of cells in the ICM: $0.595 \pm 0.005$ in mouse embryos, $0.694 \pm 0.011$ in monkey embryos and $0.569 \pm 0.024$ in human embryos. The probability of forming a PrE monolayer was determined by the relative frequency of embryos whose ICM–cavity interface area $A_{Interface}$ was within the area enclosed between minimal $A_{PrE}$ and maximal $A_{PrE}$. Likewise, the probability of a gap (or multilayer) was estimated as the relative frequency of embryos with $A_{Interface}$ > maximal $A_{PrE}$ (or $A_{Interface}$ < minimal $A_{PrE}$).

## Statistics and reproducibility

All statistical analysis and data visualization was performed in Python v.3.9 and the SciPy package v.1.7.1 used for statistical testing. Normality of the distribution for datasets was tested by a Shapiro–Wilks test. If the dataset followed a normal distribution, an independent-samples $t$-test was used for comparisons between two groups or one-way ANOVA was used for testing more than two groups, followed by a Tukey's post hoc test. Otherwise, a nonparametric Kruskal–Wallis test or Mann–Whitney $U$-test was used for comparisons between different groups. For testing correlation, a linear regression was used and the Pearson correlation coefficient was estimated. No statistical methods were used to predetermine sample size and no randomization method was used. The investigators were not blinded during experiments. All experiments were performed with at least three biological replicates. In box-and-whisker plots, the box spans from the first to the third quartile with a line at the median and the whiskers extend until the farthest point within 1.5 × interquartile range. Error bars indicate mean ± s.d. unless mentioned otherwise. The number of embryos analysed for different experimental conditions are indicated as $n$ values unless mentioned otherwise. Sample sizes, statistical tests, correlation coefficients and $P$ values are indicated in text, figures and figure legends.

## Reporting summary

Further information on research design is available in the Nature Portfolio Reporting Summary linked to this article.

## Data availability

All data supporting the findings of this study are available within the manuscript. Image data can be obtained from the corresponding author upon request. Previously published images of human embryonic material that were reanalysed here were obtained from https://doi.org/10.1242/dev.201522 (ref. 47). Source data are provided with this paper.

## Code availability

The source code to implement the Poissonian 3D CPM described in this study is available online at https://git.embl.de/rbelouso/dycpm/.

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

## Acknowledgements

We are grateful to the members of the Hiiragi laboratory and N. Petridou for discussions and comments on the manuscript; C. J. Chan and E. Kim for help with adapting the culture of isolated ICMs, D. Fabrèges and A. Stokkermans for assistance with image analysis; R. Bloehs, S. Friese, L. Pérez and S. Hozeifi for their technical support. We thank F. Graner for his help in the analysis of cellular dynamics and feedback on the manuscript. We thank E. Corujo-Simon and J. Nichols for sharing the human blastocyst image dataset. We thank members of the Tsukiyama group for the animal care with monkeys, in particular H. Tsuchiya and M. Nakaya. We thank the EMBL animal facility (LAR) and Animal Care at the Hubrecht Institute for their support. T.I. was supported by the JSPS Overseas Research Fellowship. The Hiiragi laboratory was supported by the EMBL, and currently by the Hubrecht Institute, the European Research Council (Advanced Grant 'SelforganisingEmbryo' grant agreement 742732 and Advanced Grant 'COORDINATION' grant agreement 101055287), Stichting

LSH-TKI (LSHM21020) and Japan Society for the Promotion of Science KAKENHI grant nos. JP21H05038 and JP22H05166. The Erzberger laboratory is supported by the EMBL.

## Author contributions

Conceptualization was by A.E. and T.H. Methodology was by P.M., R.B., A.E. and T.H. Software programming was by R.B. Validation was by P.M. and R.B. Formal analysis was by P.M. Investigation was by P.M., T.I. and C.I. Resources were by T.T. and T.H. Data curation was by P.M. and T.I. Writing (original draft) was by P.M., R.B., A.E. and T.H. Writing (review and editing) was by P.M., R.B., T.I., A.E. and T.H. Visualization was by P.M., R.B. and A.E. Supervision was by T.T., A.E. and T.H. Project administration and funding acquisition were by T.H.

## Competing interests

The authors declare no competing interests.

## Additional information

**Extended data** is available for this paper at https://doi.org/10.1038/s41556-025-01618-9.

**Correspondence and requests for materials** should be addressed to Anna Erzberger or Takashi Hiiragi.

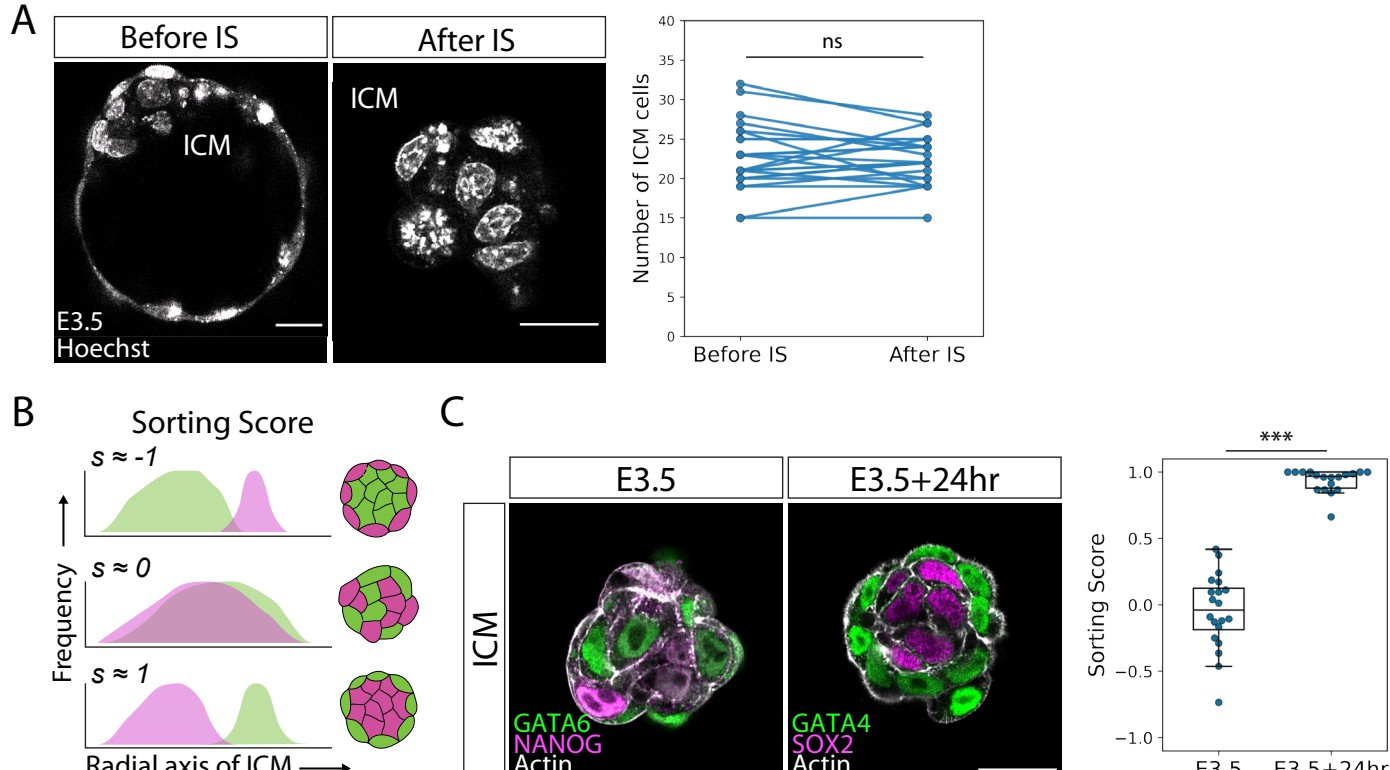

**Extended Data Fig. 1 | Differential cell movements between epiblast and primitive endoderm contribute to fate segregation in the ICM. a.** Representative images of an E3.5 blastocyst and corresponding isolated ICM after immunosurgery with quantification of total cell numbers in the ICM before and after immunosurgery. Paired samples $t$-test, two-sided, $p = 0.644$ with n = 24 embryos. **b.** Schematic representation for quantification of the sorting score in isolated ICMs. Sorting score values range from s = -1 to s = 1, with s = -1 indicating EPI enveloping PrE, s = 0 indicating salt-and-pepper distribution of cell types, and s = 1 indicating PrE enveloping EPI. **c.** Representative immunofluorescence images of ICMs isolated at stage E3.5 (left), compared to ICMs isolated at stage E3.5 followed by 24-hour *in vitro* culture (right), and quantification of the sorting score in these experimental groups. Two-sided Mann–Whitney U-test, $p = 1.45e^{-07}$ and n = 20, 18 ICMs for the two groups, respectively. Scale bar, 20 μm. *ns, non-significant,* \*\*\**p ≤ 0.001.*

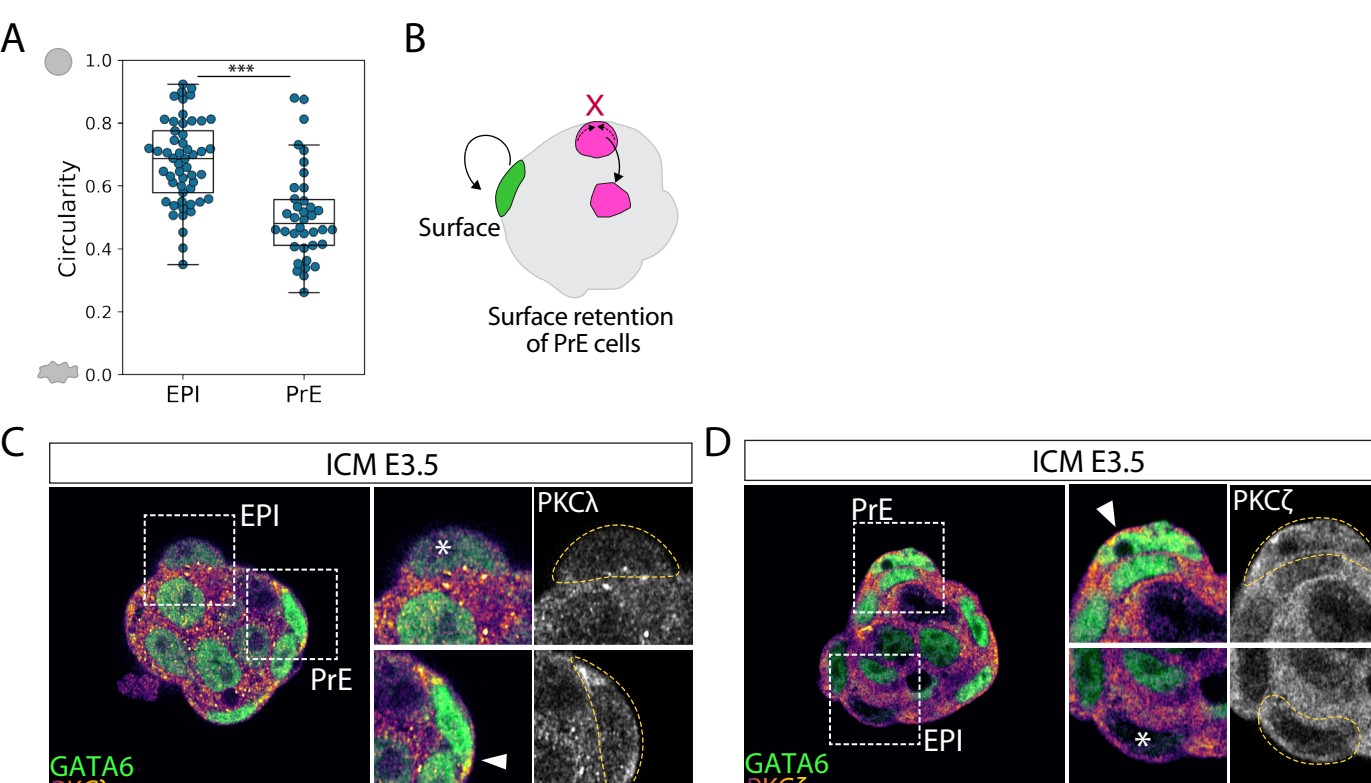

**Extended Data Fig. 2 | Acquisition of the apical domain decreases surface tension and is sufficient for retaining PrE cells at the fluid interface. a**. Analysis of surface cell circularity in E3.5 isolated ICMs to compare EPI and PrE cell shape. n = 53, 38 cells from 16 ICMs. Mann–Whitney U-test, p = 5.23e$^{-07}$. **b**. Schematic diagram for the retention hypothesis, where PrE cells are retained at the fluid interface, whereas EPI cells are not. **c**. Representative immunofluorescence image of an isolated ICM at stage E3.5 showing distribution of PKCλ, and higher magnification images of PKCλ localization in EPI (GATA6-low, top) and PrE (GATA6-high, bottom) cells on the ICM surface. White arrowhead, PKCλ

localization at the apical cortex in surface PrE cells. White asterisk, an EPI cell at the ICM surface. Yellow dotted line, cell boundary. **d**. Representative immunofluorescence image of an isolated ICM at stage E3.5 showing distribution of PKCζ, and higher magnification images of PKCζ localization in PrE (GATA6-high, top) and EPI (GATA6-low, bottom) cells on the ICM surface. White arrowhead, PKCζ localization at the apical cortex in surface PrE cells. White asterisk, an EPI cell at the ICM surface. Yellow dotted line, cell boundary. Scale bars, 20 μm; scale bars for zoomed-in panels, 10 μm.

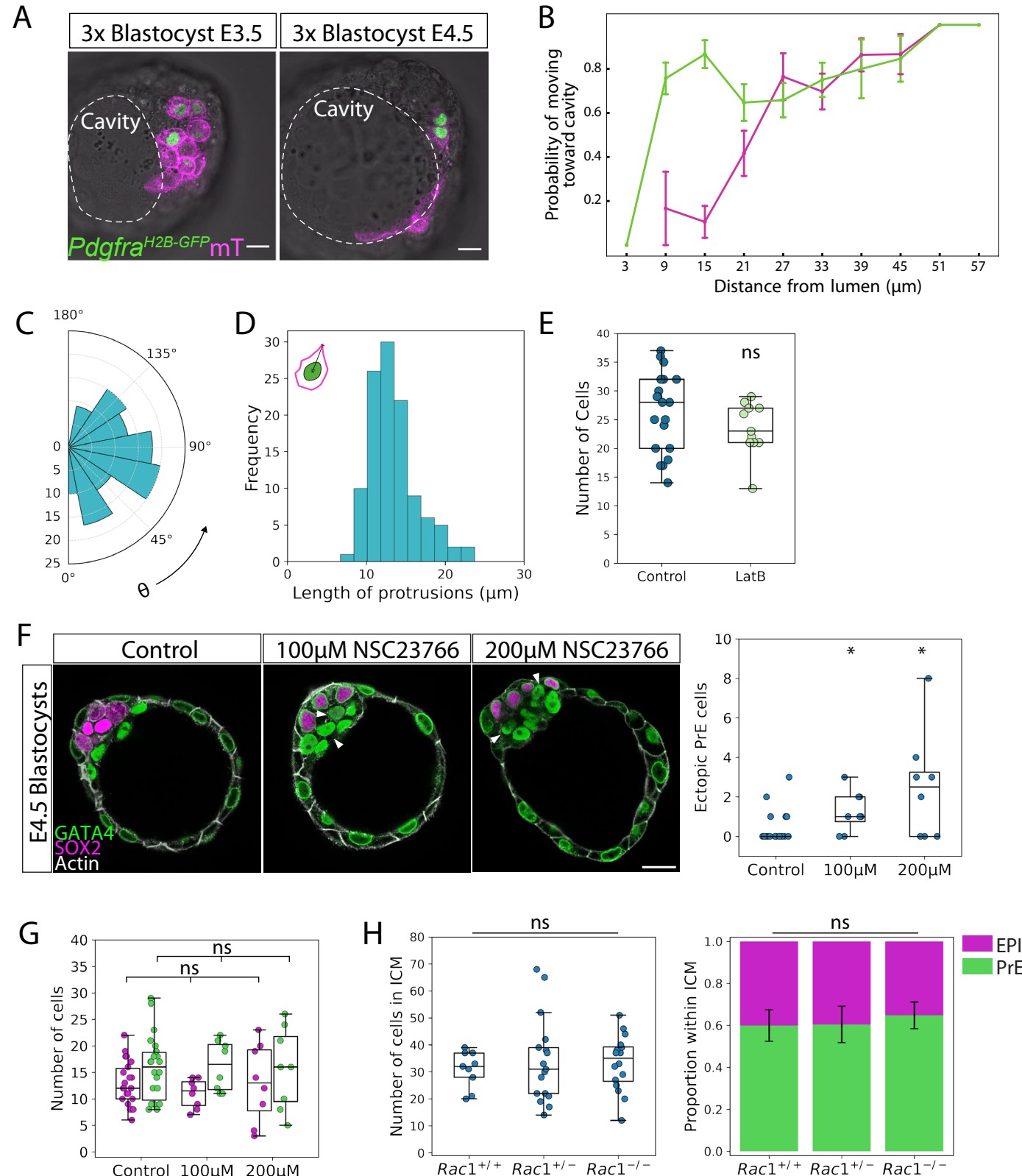

**Extended Data Fig. 3 | See next page for caption.**

**Extended Data Fig. 3 | Cell sorting involves active directed migration of PrE cells towards the surface via actin-mediated protrusions. a**. Representative time-lapse images of mosaic-labelled blastocysts at E3.5 and E4.5 generated to visualize EPI and PrE cell dynamics. **b**. Probability of EPI and PrE cells to move towards the cavity surface as a function of cellular distance from the lumen, plotted as mean±SD. Probabilities are estimated from single-cell tracking in mosaic-labelled blastocysts from Fig. 2b, c n = 14 EPI cells, 31 PrE cells from 13 embryos. For details, see Methods. **c**. Polar histogram for protrusion angles sampled from a uniform distribution between 0-180 degrees. **d**. Distribution of protrusion length in PrE cells as measured from centre of the nucleus to tip of the longest protrusion. Mean protrusion length = 13.38 ± 3.12 μm. n = 113 measurements, 43 cells from 12 embryos. **e**. Quantification of total number of ICM cells in control and latrunculin B-treated ICMs. n = 20,11 ICMs for the two groups, respectively. Two-sided Mann–Whitney U-test, $p$ = 0.207.

**f**. Representative images of control E4.5 blastocysts and 100 μM and 200 μM NSC23766-treated E4.5 blastocysts and quantification of number of ectopic PrE cells in each condition. White arrowheads, ectopic PrE cells. n = 22, 8, 8 blastocysts for the treatment groups, respectively. Two-sided Mann–Whitney U-test, $p$ = 0.011 for comparison between control and 100 μM group, $p$ = 0.012 for comparison between control and 200 μM group. **g**. Quantification of total number of EPI and PrE cells in control and NSC23766-treated E4.5 blastocysts. n = 22, 8, 8 blastocysts respectively. Kruskal–Wallis test, $p$ = 0.62 for EPI cell numbers, $p$ = 0.903 for PrE cell numbers across the treatment groups. **h**. Quantification of total number of ICM cells in $Rac1^{+/+}$, $Rac1^{+/-}$, and $Rac1^{-/-}$ E4.5 blastocysts. n = 9, 17, 16 blastocysts respectively. One-way ANOVA, $p$ = 0.826. Quantification of EPI/PrE cell fate proportion within the ICM in $Rac1^{+/+}$, $Rac1^{+/-}$, and $Rac1^{-/-}$ E4.5 blastocysts, plotted as mean±SD. One-way ANOVA, p = 0.216. Scale bar 20 μm. *ns, non-significant, *$p$ ≤ 0.05.

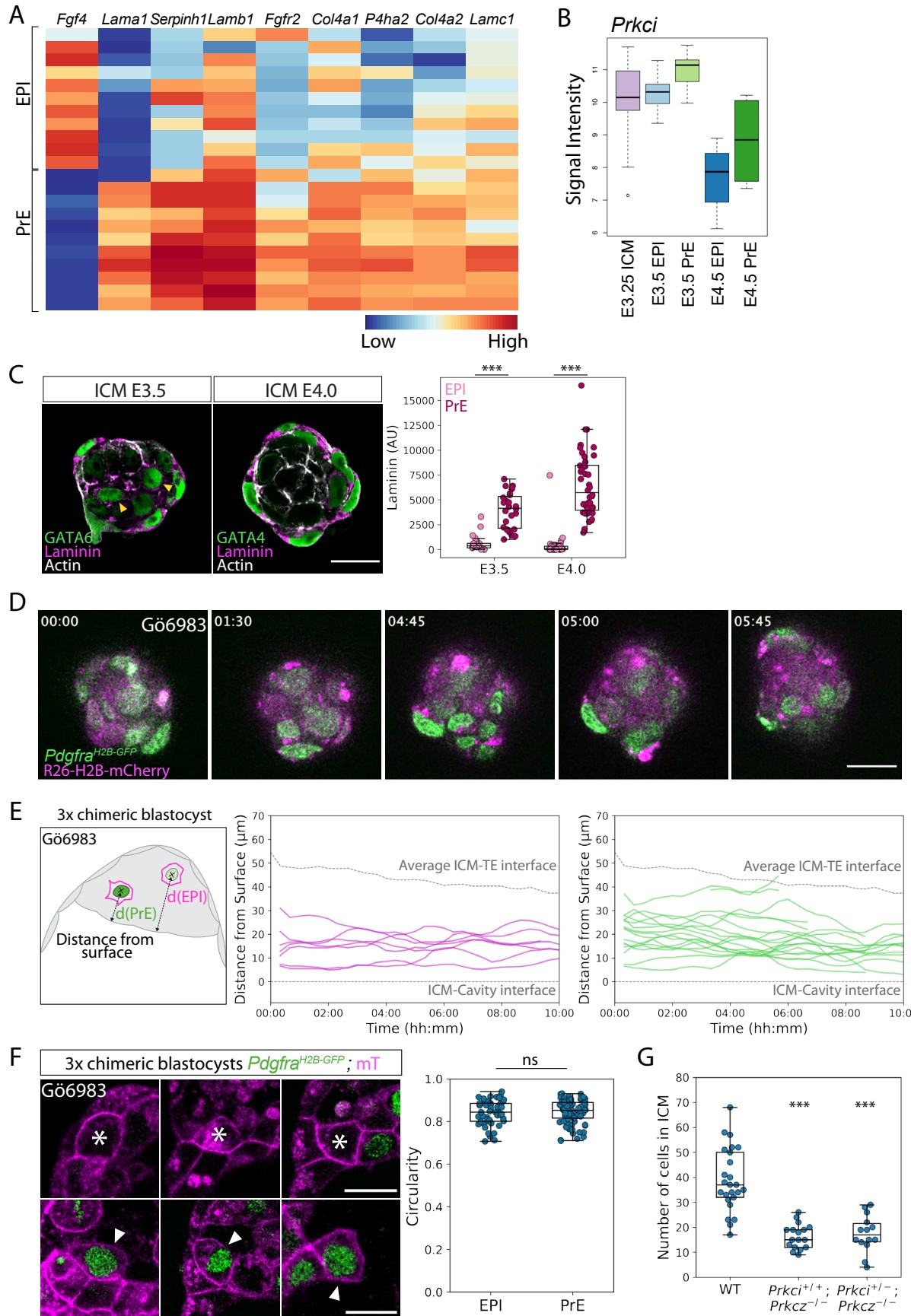

Extended Data Fig. 4 | See next page for caption.

**Extended Data Fig. 4 | Apical polarization in PrE cells is required for directed migration and sorting. a**. Heatmap indicating gene expression of extracellular matrix-related components in EPI and PrE cells from E3.5 blastocysts analysed by single-cell qPCR (22 cells from 3 embryos at E3.5) from ref. 13. **b**. Boxplot indicating differential expression of *Prkci* among EPI and PrE cells in the ICM cells at different stages from ref. 13, n = 36, 11, 11, 4, 4 cells from 6, 3, 3, 1, 1 embryos for the different groups, respectively. **c**. Immunofluorescence images of E3.5 and E4.0 isolated ICMs. Quantification of laminin deposition around EPI and PrE cells in E3.5 and E4.0 isolated ICMs. n = 18, 27, 37, 42 cells for the different groups, analysed from 7,7 embryos at stages E3.5 and E4.0 respectively. Two-sided Mann–Whitney U-test, $p = 2.54e^{-07}$ for stage E3.5, $p = 1.46e^{-13}$ for stage E4.0. **d**. Time-lapse imaging of ICMs isolated from E3.5 blastocysts expressing $Pdgfra^{H2B-GFP}$ and *R26-H2B-mCherry*, treated with aPKC inhibitor Gö6983. White arrowhead marks a PrE cell unable to maintain its surface position. Time is indicated as hh:mm, t = 00:00 corresponds to start of live-imaging at stage E3.5 + 3 hours, following completion of immunosurgery. **e**. Single-cell tracking and analysis of distance of fluorescence-labelled EPI and PrE cells from the cavity surface in chimeric blastocysts treated with the aPKC inhibitor Gö6983. Individual cell position curves were smoothed using a rolling average. Grey dotted lines mark the average position of ICM–TE interface, and ICM– cavity interface set at d = 0. n = 7 EPI cells, 19 PrE cells from 7 embryos. **f**. Representative images of EPI (top) and PrE (bottom) cell shape and its quantification in mosaic-labelled E3.75 blastocysts upon inhibition of aPKC with Gö6983. White asterisks, EPI cells of interest. White arrowheads, PrE cells. Two-sided Mann–Whitney U-test, $p = 0.6$, n = 48, 80 measurements for EPI and PrE cells from 7 blastocysts, respectively. **g**. Box and scatter-plot for total ICM cell number in WT, $Prkci^{+/+};Prkcz^{-/-}$, and $Prkci^{+/-};Prkcz^{-/-}$ blastocysts at E4.5 stage. n = 25,17 and 14 blastocysts for the experimental groups, respectively. Two-sided independent-samples t-test, $p = 2.23e^{-08}$ for $Prkci^{+/+};Prkcz^{-/-}$, and $p = 1.25e^{-06}$ for $Prkci^{+/-};Prkcz^{-/-}$ blastocysts. Scale bars, 20 μm. *ns, non-significant,* ***$p \le 0.001$.

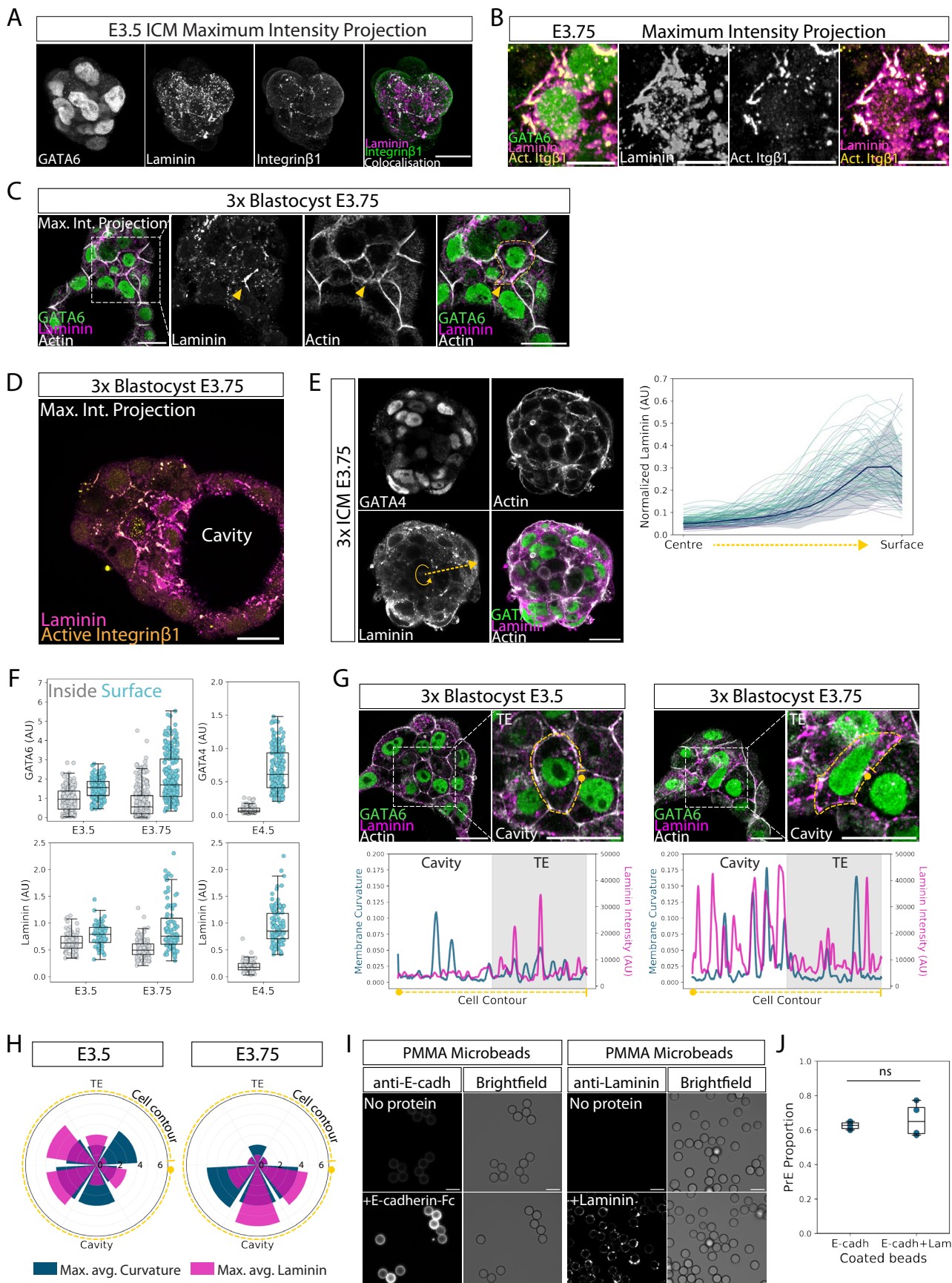

**Extended Data Fig. 5 | See next page for caption.**

**Extended Data Fig. 5 | Extracellular matrix deposited in the ICM guides PrE cells towards the cavity surface. a**. Immunofluorescence maximum intensity projection of an isolated ICM at stage E3.5. **b**. Maximum intensity projections of laminin and active integrinβ1 (9EG7) in PrE cells in the ICM in 3× blastocysts at stage E3.75. Manders' coefficients for colocalised fractions are: Laminin overlapping Itgβ1 = 0.185, and Itgβ1 overlapping Laminin = 0.891. **c**. Immunofluorescence images of an inner PrE cell at the onset of migratory activity. Yellow arrowhead, site of protrusion; yellow dotted line, cell boundary. **d**. Maximum intensity projection of a 3× blastocyst at stage E3.75 to visualize ECM distribution in the ICM. **e**. Immunofluorescence images and radial laminin fluorescence intensity in 3× isolated ICMs at E3.75 stage, plotted as mean±SD. Yellow dotted arrow, line segments along which fluorescence intensity was measured. Intensity profiles were smoothed using a rolling average, and lines of the same colour correspond to measurements from the same embryo. Data from n = 8 embryos. **f**. Quantification of nuclear GATA6/GATA4 fluorescence intensities and laminin accumulation around cells located at the ICM surface or inside the ICM from 3× blastocysts at stages E3.5, E3.75 and E4.5. GATA6/GATA4 measurements from 366, 493, and 350 cells from n = 9,13,10 3× blastocysts for

stages E3.5, E3.75 and E4.5 respectively. Laminin measurements from 140, 216, 199 cells from n = 9,13,10 3× blastocysts for stages E3.5, E3.75 and E4.5 respectively. **g**. Immunofluorescence images of non-migratory and migratory PrE cells in 3× blastocysts at stages E3.5 and E3.75, respectively. Quantification of local membrane curvature and laminin distribution along the PrE cell contour. Yellow dotted lines, cell contour traced starting at the yellow circle clockwise until the flat arrowhead. Line profiles were smoothed using a rolling average. n = 19, 17 cells from 15, 14 embryos at E3.5 and E3.75 stages, respectively. **h**. Orientation of PrE cell membrane regions with highest average curvature and highest average laminin distribution along the cell contour from (g) at stages E3.5 and E3.75, respectively. The cell contour was binned into 6 length intervals, and the curvature and laminin intensity were averaged for each bin. n = 19, 17 cells from 15, 14 embryos at E3.5 and E3.75 stages, respectively. **i**. Immunofluorescence and brightfield images of control and coated PMMA microbeads incubated with E-cadherin-Fc chimeric protein and laminin protein. **j**. PrE cell proportion in the ICM of blastocysts with implanted beads at stage E4.5. Data from n = 5,4 embryos for the two groups respectively, two-sided independent-samples t-test, $p = 0.46$. Scale bars, 20 μm. *ns, non-significant,* \*\*\**p* ≤ 0.001.

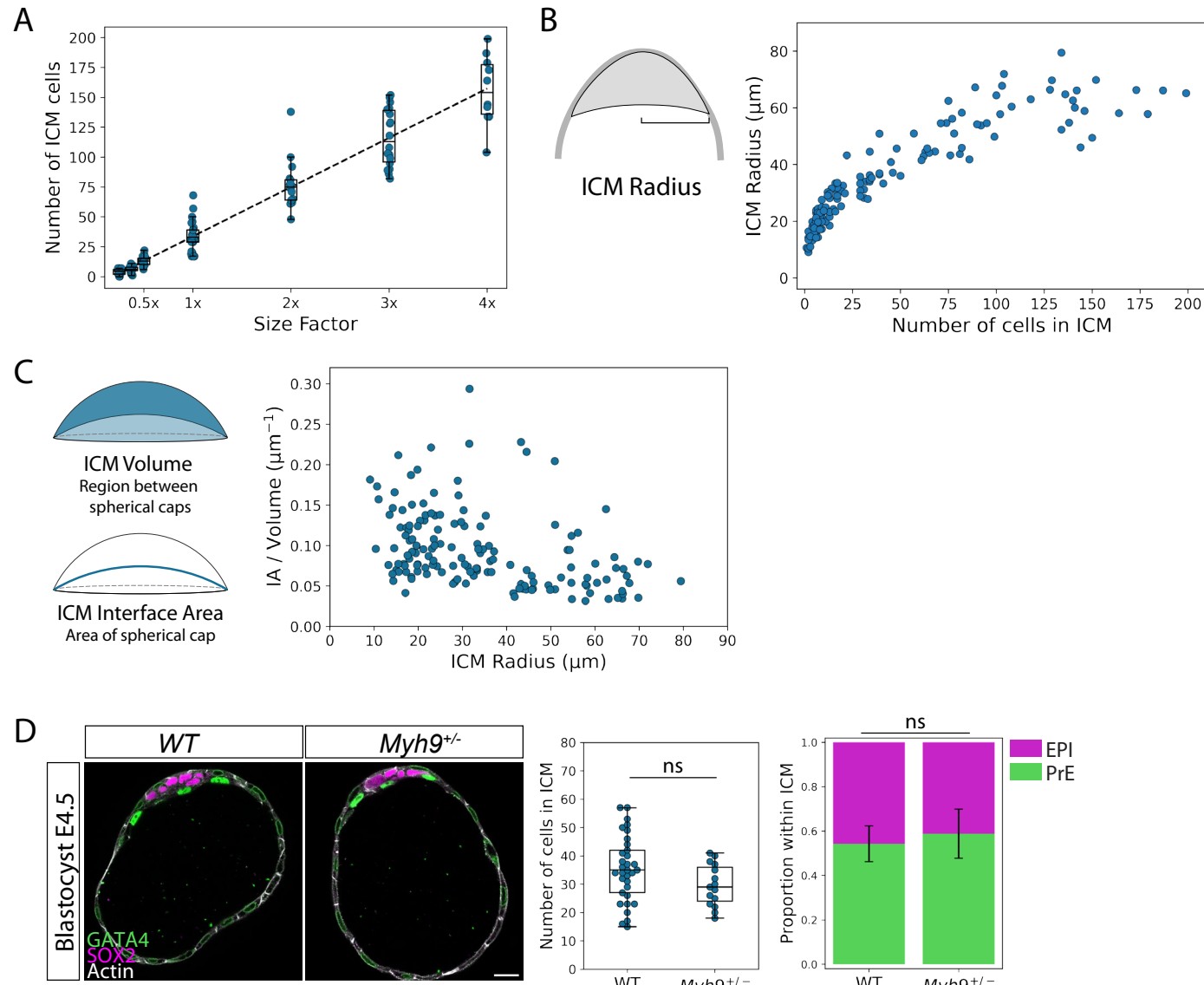

**Extended Data Fig. 6 | The fixed proportion of EPI/PrE cells without cell fate-switching challenges precision in ICM patterning. a**. Number of cells in the ICM scales linearly with embryo size factor in size-manipulated E4.5 blastocysts. Dotted line, linear regression with Pearson's R = 0.961, $p = 1.22e^{-88}$. n = 30 embryos for 2/8×, 29 for 3/8×, 24 for 4/8×, 29 for 1×, 17 for 2×, 18 for 3×, and 10 embryos for 4× size ratios. **b**. Scatter-plot depicting how ICM radius increases non-linearly with total cell number in the ICM in size-manipulated E4.5 blastocysts. n = 25 embryos for 2/8×, 26 for 3/8×, 24 for 4/8×, 29 for 1×, 17 for 2×, 18 for 3×, and 10 embryos for 4× size ratios. **c**. Scatter-plot depicting how ratio of ICM interface area to volume decreases with increasing ICM radius. n = 25 embryos for 2/8×,

26 for 3/8×, 24 for 4/8×, 29 for 1×, 17 for 2×, 18 for 3×, and 10 embryos for 4× size ratios. **d**. Representative immunofluorescence images of WT and *Myh9*[+/−] blastocysts at stage E4.5. Box and scatter-plot for quantification of total ICM cell number in WT and *Myh9*[+/−] blastocysts at stage E4.5. n = 33, 15 embryos for WT and *Myh9*[+/−], respectively. Two-sided independent-samples t-test, *p* = 0.086. Stacked bar plot indicating EPI/PrE cell fate proportion within the ICM in WT and *Myh9*[+/−] blastocysts at stage E4.5, plotted as mean ± SD. n = 33, 15 embryos for WT and *Myh9*[+/−], respectively. Two-sided independent-samples t-test, *p* = 0.121. Scale bars, 20 μm. *ns, non-significant*.

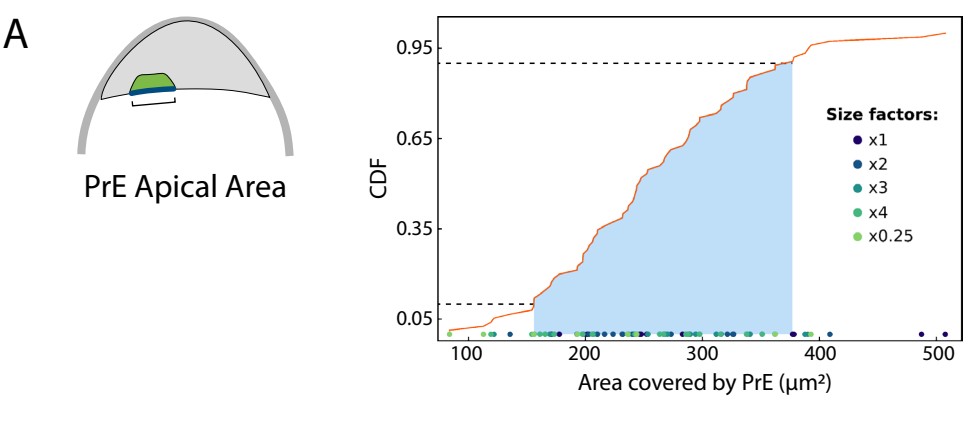

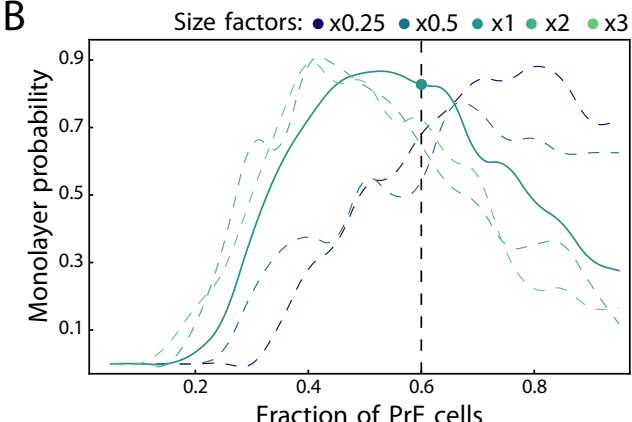

**Extended Data Fig. 7 | The fixed proportion of EPI/PrE cells is optimal for the specific embryo size and ICM geometry. a**. Schematic diagram for PrE cell apical area and distribution of individual PrE apical areas measured in 3D across size-manipulated E4.5 blastocysts. Blue shaded region corresponds to the region between the 10th and 90th percentile of 75 apical area measurements from 20 blastocysts with $q_{10\%}$=157 µm² and $q_{90\%}$=376 µm². **b**. Probability of monolayer PrE formation for size-manipulated embryos at stage E4.5 with respect to PrE proportion within the ICM. Black dotted line indicates the fixed 60% PrE proportion in the ICM, for which the highest monolayer probability is in 1× blastocysts. Probability derived from n = 25, 26, 24, 29, 17, 18, 10 embryos for 2/8×, 3/8×, 4/8×, 1×, 2×, 3×, 4× size ratios, respectively.

# Reporting Summary

## Statistics

For all statistical analyses, confirm that the following items are present in the figure legend, table legend, main text, or Methods section.

| n/a | Confirmed | |
|---|---|---|
| ☐ | ☒ | The exact sample size (*n*) for each experimental group/condition, given as a discrete number and unit of measurement |
| ☐ | ☒ | A statement on whether measurements were taken from distinct samples or whether the same sample was measured repeatedly |
| ☐ | ☒ | The statistical test(s) used AND whether they are one- or two-sided<br>*Only common tests should be described solely by name; describe more complex techniques in the Methods section.* |
| ☐ | ☒ | A description of all covariates tested |
| ☐ | ☒ | A description of any assumptions or corrections, such as tests of normality and adjustment for multiple comparisons |
| ☐ | ☒ | A full description of the statistical parameters including central tendency (e.g. means) or other basic estimates (e.g. regression coefficient) AND variation (e.g. standard deviation) or associated estimates of uncertainty (e.g. confidence intervals) |
| ☐ | ☒ | For null hypothesis testing, the test statistic (e.g. *F*, *t*, *r*) with confidence intervals, effect sizes, degrees of freedom and *P* value noted<br>*Give P values as exact values whenever suitable.* |
| ☒ | ☐ | For Bayesian analysis, information on the choice of priors and Markov chain Monte Carlo settings |
| ☒ | ☐ | For hierarchical and complex designs, identification of the appropriate level for tests and full reporting of outcomes |
| ☒ | ☐ | Estimates of effect sizes (e.g. Cohen's *d*, Pearson's *r*), indicating how they were calculated |

*Our web collection on statistics for biologists contains articles on many of the points above.*

## Software and code

Policy information about availability of computer code

| Data collection | Zen (Zeiss, 2012), Dikeria (Biro and Maitre, 2015) |
|---|---|
| Data analysis | FIJI v2.14.0 (RRID: SCR_002285), Imaris v9.7.2 (RRID: SCR_007370), Napari v0.4.17, Mov-IT (Faure et al., 2016), Nuclear Detection and Tracking (Fabreges et al., 2024), Python v3.9 (RRID:SCR_008394), SciPy v1.7.1 (RRID:SCR_008058), Scikit-learn (RRID:SCR_019053), Wolfram Mathematica (RRID:SCR_014448) |

For manuscripts utilizing custom algorithms or software that are central to the research but not yet described in published literature, software must be made available to editors and reviewers. We strongly encourage code deposition in a community repository (e.g. GitHub). See the Nature Portfolio guidelines for submitting code & software for further information.

## Data

Policy information about availability of data

All manuscripts must include a data availability statement. This statement should provide the following information, where applicable:
- Accession codes, unique identifiers, or web links for publicly available datasets
- A description of any restrictions on data availability
- For clinical datasets or third party data, please ensure that the statement adheres to our policy

Source data are provided with this manuscript. All other data supporting the findings of this study are available within the manuscript. Image data can be obtained from the corresponding author upon request. Previously published images of human embryonic material that were re-analysed here were obtained from

doi.org/10.1242/dev.201522.

# Research involving human participants, their data, or biological material

Policy information about studies with human participants or human data. See also policy information about sex, gender (identity/presentation), and sexual orientation and race, ethnicity and racism.

| | |
|---|---|
| Reporting on sex and gender | No human research participants were used. |
| Reporting on race, ethnicity, or other socially relevant groupings | Not Applicable |
| Population characteristics | Not Applicable |
| Recruitment | Not Applicable |
| Ethics oversight | Not Applicable |

Note that full information on the approval of the study protocol must also be provided in the manuscript.

# Field-specific reporting

Please select the one below that is the best fit for your research. If you are not sure, read the appropriate sections before making your selection.

☒ Life sciences  ☐ Behavioural & social sciences  ☐ Ecological, evolutionary & environmental sciences

For a reference copy of the document with all sections, see nature.com/documents/nr-reporting-summary-flat.pdf

# Life sciences study design

All studies must disclose on these points even when the disclosure is negative.

| | |
|---|---|
| Sample size | Sample size was based on prior literature using similar experimental paradigms:<br><br>Ryan, A. Q., Chan, C. J., Graner, F., & Hiiragi, T. (2019). Lumen expansion facilitates epiblast-primitive endoderm fate specification during mouse blastocyst formation. Developmental cell, 51(6), 684-697.<br><br>Chan, C. J., Costanzo, M., Ruiz-Herrero, T., Mönke, G., Petrie, R. J., Bergert, M., ... & Hiiragi, T. (2019). Hydraulic control of mammalian embryo size and cell fate. Nature, 571(7763), 112-116. |
| Data exclusions | 1. In the cell tracking analysis in isolated ICMs, all cells in the ICM were inspected, cells that could not be tracked with confidence (<1% cells) were excluded from the lineage trees.<br>2. In the embryo aggregation experiments, embryos that failed to aggregate and those that did not form a singular blastocyst cavity were excluded from analysis.<br>3. For quantification of ectopic ICM cells in size-manipulated mouse blastocysts, small-sized blastocysts lacking an ICM were excluded from the analysis.<br>4. In size-manipulated mouse blastocysts, small-sized blastocysts lacking an ICM and those lacking an ICM-cavity fluid interface were excluded from the analysis to measure ICM geometrical dimensions.<br>5. In the implanted bead experiments, embryos where the bead was not successfully implanted in the ICM were excluded. |
| Replication | All data were generated in triplicates, and replication attempts were successful. |
| Randomization | Mice of the desired genotype were pooled together and randomly allocated for experiments. This is not relevant for monkey blastocysts as we do not have multiple experimental groups, all monkey embryos obtained from intracytoplasmic sperm injection of oocytes were fixed at late blastocyst stage. |
| Blinding | For experiments with knock-out mutant embryos, the researchers were blinded to the genotype of the samples during immunofluorescence imaging and data analysis as single-embryo genotyping by PCR was performed retrospectively. |

# Reporting for specific materials, systems and methods

We require information from authors about some types of materials, experimental systems and methods used in many studies. Here, indicate whether each material, system or method listed is relevant to your study. If you are not sure if a list item applies to your research, read the appropriate section before selecting a response.

## Materials & experimental systems

| n/a | Involved in the study |
|-----|----------------------|
| ☐ | ☒ Antibodies |
| ☒ | ☐ Eukaryotic cell lines |
| ☒ | ☐ Palaeontology and archaeology |
| ☐ | ☒ Animals and other organisms |
| ☒ | ☐ Clinical data |
| ☒ | ☐ Dual use research of concern |
| ☒ | ☐ Plants |

## Methods

| n/a | Involved in the study |
|-----|----------------------|
| ☒ | ☐ ChIP-seq |
| ☒ | ☐ Flow cytometry |
| ☒ | ☐ MRI-based neuroimaging |

# Antibodies

| | |
|---|---|
| Antibodies used | Primary antibodies used in this study were:<br>GATA6 (R&D systems, AF1700),<br>GATA4 for mouse embryos (R&D systems, BAF2606),<br>GATA4 for monkey embryos (Cell Signalling, 36966S),<br>SOX2 (Cell Signaling, 23064),<br>bi-phosphorylated myosin regulatory light chain (ppMRLC) (Cell Signaling, 3674),<br>Laminin (Novus Biologicals, NB300-14422),<br>NANOG (ReproCell, RCAB002P-F),<br>PKC-lambda (Santa Cruz Biotechnology, sc-17837),<br>PKC-zeta (Santa Cruz Biotechnology, sc-17781),<br>Integrinb1 clone MB1.2 (Millipore, MAB1997),<br>active Integrinb1 (9EG7, BD Bioscience, 553715),<br>E-cadherin (Sigma, U3254),<br>RFP/tdTomato (Rockland, 600-401-379)<br>RFP/tdTomato (Chromotek, 5f8)<br>Alexa Fluor 647-conjugated Oct3/4 (Santa Cruz, sc-5279 AF647).<br><br>Secondary antibodies used in this study were:<br>donkey anti-goat IgG Alexa Fluor 488 (Invitrogen, A11055),<br>donkey anti-rabbit IgG Alexa Fluor Plus 488 (Thermo Fisher Scientific, A32790),<br>donkey anti-rabbit IgG Alexa Fluor 546 (Invitrogen, A10040),<br>donkey anti-mouse IgG Alexa Fluor 555 (Invitrogen, A31570),<br>donkey anti-rabbit IgG Alexa Fluor 647 (Invitrogen, A31573),<br>donkey anti-mouse Cy5 (Jackson ImmunoResearch, 715-175-150),<br>donkey anti-rat Cy5 (Jackson ImmunoResearch, 712-175-153),<br>donkey anti-rabbit IgG Alexa Fluor Plus 488 (Thermo Fisher Scientific, A32790). |
| Validation | GATA6 (R&D systems, AF1700), was validated by the manufacturer using WB and ELISA.<br>GATA4 (R&D systems, BAF2606), was validated by the manufacturer using WB.<br>GATA4 (Cell Signalling, 36966S) was validated by the manufactures using WB and IF.<br>Alexa Fluor 647-conjugated Oct3/4 (Santa Cruz, sc-5279 AF647) was validated by the manufacturer using WB.<br>SOX2 (Cell Signaling, 23064), was validated by the manufacturer using WB, IP, IF and ChIP.<br>bi-phosphorylated myosin regulatory light chain (ppMRLC) (Cell Signaling, 3674), was validated by the manufacturer using WB.<br>Laminin (Novus Biologicals, NB300-14422), was validated by the manufacturer using WB, IF.<br>NANOG (ReproCell, RCAB002P-F), was validated by the manufacturer using IF.<br>PKC-lambda (Santa Cruz Biotechnology, sc-17837), was validated by the manufacturer using WB.<br>PKC-zeta (Santa Cruz Biotechnology, sc-17781), was validated by the manufacturer using WB, IF.<br>Integrinb1 (Millipore, MAB1997), was validated by the manufacturer using WB.<br>active Integrinb1 (9EG7, BD Bioscience, 553715) was validated by the manufacturer using IHC.<br>E-cadherin (Sigma, U3254) was validated by the manufacturer using IHC and IP.<br>RFP/tdTomato (Rockland, 600-401-379) was validated by the manufacturer using IHC, IF and WB<br>RFP/tdTomato (Chromotek, 5f8) was validated by the manufacturer using IF and ELISA |

# Animals and other research organisms

Policy information about studies involving animals; ARRIVE guidelines recommended for reporting animal research, and Sex and Gender in Research

| | |
|---|---|
| Laboratory animals | Mouse, species: Mus musculus, strain: C57BL/6xC3H F1 hybrids, age: between 8-30 weeks of age. Details of genetically-modified mice (GM) are provided in the Methods section of the manuscript.<br>Monkey, species: Macaca fascicularis (monkeys do not have strain information as in inbred mouse strains), ages ranging between 6 to 11 years. |
| Wild animals | The study did not involve wild animals. |
| Reporting on sex | Sex does not influence the findings reported in the study. |

| Field-collected samples | The study did not involve field-collected samples. |
| --- | --- |
| Ethics oversight | All mouse-related animal work  performed at the Laboratory Animal Resources (LAR) Facility at European Molecular Biology Laboratory (EMBL) was done with permission from the Institutional Animal Care and Use Committee (IACUC) overseeing the operation (IACUC number TH11 00 11). LAR facilities operate according to Federation for Laboratory Animal Science Associations (FELASA) guidelines and recommendations.<br><br>At the Hubrecht Institute animal facility, mice were housed according to institutional guidelines, and procedures were performed in compliance with Standards for Care and Use of Laboratory Animals with approval from the Hubrecht Institute ethical review board. Animal experiments were approved by the Animal Experimentation Committee (DEC) of the Royal Netherlands Academy of Arts and Sciences.<br><br>Monkey animal work was appropriately performed by following the Animal Research:Reporting in Vivo Experiments (ARRIVE) guidelines developed by the National Centre for the Replacement, Refinement & Reduction of Animals in Research (NC3Rs), and also by following "The Act on Welfare and Management of Animals" from Ministry of the Environment, "Fundamental Guidelines for Proper Conduct of Animal Experiment and Related Activities in Academic Research Institutions" under the jurisdiction of the Ministry of Education, Culture, Sports, Science and Technology, and "Guidelines for Proper Conduct of Animal Experiments" from Science Council of Japan. All animal experimental procedures were approved by the Animal Care and Use Committee of Shiga University of Medical Science (approval number: 2021-10-4) |

Note that full information on the approval of the study protocol must also be provided in the manuscript.

