## [Peer Review File · Nature Cell Biology]

Coupling of cell shape, matrix, and tissue dynamics ensures embryonic patterning robustness

Corresponding Author: Professor Takashi Hiiragi

Version 0:

Decision Letter:

*Please delete the link to your author homepage if you wish to forward this email to co-authors.

Dear Dr Hiiragi,

I apologize for the delay. Your manuscript, "Optimality of cell, matrix, and tissue dynamics coupling ensures patterning robustness", has now been seen by 3 referees, who are experts in mouse embryonic development (referee 1); biomechanics and modelling (referee 2); and mechanics and cell fate (referee 3). As you will see from their comments (attached below) they find this work of potential interest, but have raised substantial concerns, which in our view would need to be addressed with considerable revisions before we can consider publication in Nature Cell Biology.

Nature Cell Biology editors discuss the referee reports in detail within the editorial team, including the chief editor, to identify key referee points that should be addressed with priority, and requests that are overruled as being beyond the scope of the current study. To guide the scope of the revisions, I have listed these points below. We are committed to providing a fair and constructive peer-review process, so please feel free to contact me if you would like to discuss any of the referee comments further.

I should stress that the referees' concerns point to potential pleiotropic effects and unclear mechanistic links which would need to be addressed with experiments and data, and reconsideration of the study for this journal and re-engagement of referees would depend on strength of these revisions.

In particular, it would be essential to:

- A) Experimentally assess the contribution of ECM on embryonic sorting and migration (all Reviewers)
- B) Further assess the effects on cell fate specification, taking into account interspecies considerations and scaling (Reviewer #2). Please note that although Reviewer #2 suggests removing cell fate specification analysis from your manuscript, we consider this essential for consideration at Nature Cell Biology and would ask for further analysis.
- C) Experimentally assess the potential role(s) of mechanics and material properties of the different tissues (Reviewers #2 and #3)
- D) All other referee concerns pertaining to strengthening existing data, providing controls, methodological details, clarifications and textual changes, as well as appropriate citation and discussion of relevant literature should also be addressed.
- E) Finally please pay close attention to our guidelines on statistical and methodological reporting (listed below) as failure to do so may delay the reconsideration of the revised manuscript. In particular please provide:

We would be happy to consider a revised manuscript that would satisfactorily address these points, unless a similar paper is published elsewhere, or is accepted for publication in Nature Cell Biology in the meantime.

- ensure that it conforms to our format instructions and publication policies (see below and www.nature.com/nature/authors/).

- provide a point-by-point rebuttal to the full referee reports verbatim, as provided at the end of this letter.

- provide the completed Editorial Policy Checklist (found here <https://www.nature.com/authors/policies/Policy.pdf>), and Reporting

Summary (found here https://www.nature.com/authors/policies/ReportingSummary.pdf). This is essential for reconsideration of the manuscript and these documents will be available to editors and referees in the event of peer review. For more information see http://www.nature.com/authors/policies/availability.html or contact me.

Nature Cell Biology is committed to improving transparency in authorship. As part of our efforts in this direction, we are now requesting that all authors identified as 'corresponding author' on published papers create and link their Open Researcher and Contributor Identifier (ORCID) with their account on the Manuscript Tracking System (MTS), prior to acceptance. ORCID helps the scientific community achieve unambiguous attribution of all scholarly contributions. You can create and link your ORCID from the home page of the MTS by clicking on 'Modify my Springer Nature account'. For more information please visit please visit www.springernature.com/orcid.

Link Redacted

We would like to receive a revised submission within six months. We would be happy to consider a revision even after this timeframe, however if the resubmission deadline is missed and the paper is eventually published, the submission date will be the date when the revised manuscript was received.

We hope that you will find our referees' comments, and editorial guidance helpful. Please do not hesitate to contact me if there is anything you would like to discuss.

Best wishes,

Daryl

Daryl Jason Verzosa David, PhD

Senior Editor, Nature Cell Biology
Nature Portfolio

Heidelberger Platz 3, 14197 Berlin, Germany
Email: daryl.david@nature.com
ORCID: <https://orcid.org/0000-0002-9253-4805>

Reviewers' Comments:

Reviewer #1:

Remarks to the Author:

In this manuscript the authors analyse the mechanisms that help ensure the precision of patterning during mammalian pre-implantation development. For this they study the dynamics of segregation of the epiblast (EPI) and primitive endoderm (PrE). They find that PrE and EPI cells sort by a combination of differential membrane tension, that pushes EPI cells away from the surface, and active migration, that directs. Furthermore, they also observe that as PrE cells move to the surface of the embryo they establish a gradient of ECM. Finally, they provide evidence that the proportion of EPI to PrE cells is species and dependant on the size and geometry of the embryo.

This is a very interesting study that sheds important light on the mechanics of pre-implantation development. The dissection of the different contributing factors is elegant and provides a convincing model. There is a point that would however benefit from clarification. The authors argue that the graded ECM deposition guides PrE migration. As the data currently stands, the differences in ECM localization could also be interpreted as being a consequence of PrE sorting, but not contributing to their migration. If the authors would like to make this statement, they would need to provide some functional data.

Reviewer #2:

Remarks to the Author:

A. Summary of the key results

This paper focuses on the mechanisms by which internal embryonic epiblast (EPI) cells and primitive endoderm (PrE) cells organize themselves in mammalian blastocysts: an initially well-mixed population of the two cell types in the inner cell mass (ICM) sorts itself out so that the PrE cells form a monolayer in contact with a fluid cavity and the EPI cells cluster together between the PrE cells and the overlying trophectoderm. There are two primary sets of key results. The first focuses on the cellular / biophysical mechanisms by which the sorting takes place. These include key roles for long actin-based protrusions on PrE cells, polarization and reduced apical surface tension of PrE cells (that makes it more favorable for them to position themselves on the ICM surface), and a positive feedback mechanism in which secretion of extracellular matrix (ECM) from PrE cells gradually forms a gradient of ECM density that provides cues for the innermost PrE cells to direct their motility toward the ICM surface. The second set of key results focuses on cell fate specification within the ICM. One observation is that very few cells change from PrE to EPI or vice-versa based on their position within the ICM, i.e., there is little to no feedback from positional information to cell-fate specification. Another is that different species (mouse, monkey, human)

with different-sized blastocysts seem to have evolved toward different PrE and EPI cell-fractions in the ICM that are optimal for forming monolayers of PrE cells over EPI cells – given that species' specific blastocyst geometry. The cell-fractions are stable within a species even when blastocyst size varies, so there does not appear to be feedback between ICM size and cell-fractions. The robustness of the system with respect to making PrE surface monolayers for a wide range of surface-area-to-volume ratios is instead made possible by variations in the average surface area of PrE cells.

B. Originality and significance: if not novel, please include reference

The questions of how self-organization is accomplished in early blastocysts is clearly significant. There have of course been decades of studies on a variety of cell-sorting phenomenon, but the authors undertake a particularly detailed and thorough examination of the cellular / biophysical mechanisms here. The finding of a positive feedback between current cell position, ECM secretion, ECM density gradients, and directed motility of PrE cells is novel, as is the finding of an early role for PrE cell polarization even before PrE cells reach the ICM surface. The second set of key findings related to cell-fate specification within the ICM are less significant. In many ways, they are negative results showing that there is not really any influence on cell-fate specification from position within the ICM or size of the ICM. That does bring up an interesting question of whether the system is then 'brittle' in the face of intraspecies ICM- and blastocyst-size variation, but the flexibility of PrE cells to adopt a wide range of more- or less-flattened geometries, and thus surface areas, provides robustness. That is a useful observation, but it makes the initial question a bit less interesting.

C. Data & methodology: validity of approach, quality of data, quality of presentation

The experimental approaches are sound and well described, as are the detailed and quantitative analyses of the image sets. The figures are well constructed, and the writing is mostly clear.

There is one issue that decreases the quality of presentation: lots of relevant information is distributed to the figure captions and the Extended Data. This presentation makes the reader do a lot of work to put together the complete story line. Such decisions are often driven by length constraints, and I would suggest, as I do below, that the authors consider focusing this paper on the cellular and biophysical mechanisms of PrE/EPI cell sorting, which would free up space to more clearly present these key findings.

Some other minor points:

Fig. 4E: The schematic describing how polarization index is calculated is not at all clear. What are the two ends of the arrow pointing to?

Section on ECM deposition (lines 238-275). Since many of the earlier experiments were conducted on isolated ICMs, are radial ECM gradients observable in these isolated ECMs?

D. Appropriate use of statistics and treatment of uncertainties

Lines 120-124 and Fig. 2B: The authors conclude that "PrE cells located at the ICM surface were more stretched, in significant contrast to the more rounded EPI cells." The scatter plots of cell aspect ratio in Fig. 2B do show that the distribution of aspect ratios is slightly skewed to larger values (flatter cells) for PrE cells; however, the authors evaluate significance here with a Mann-Whitney U test, so they should be a little more exact in describing the findings. The null hypothesis of this test is just that comparing a pair of randomly chosen surface PrE and EPI cells will with equal probability find the PrE or EPI cell being flatter (having larger aspect ratio). Rejecting the null hypothesis means that the distribution of PrE cell aspect ratios is different from that of EPI cells. Both cell types have lots of rounded cells, but PrE-cell distribution has a longer tail of flatter cells.

Fig. 2D: This appears to be a fairly weak correlation of measured surface tension with Pdgfra(H2B-GFP) intensity. There should be some quantification offered for this correlation, e.g., a Pearson correlation coefficient or a slope +/- standard error for the best fit line.

Line 289-291: When the PrE cell-fraction is noted in the main text, it is reported with a very tight constraint as 0.606 +/- 0.08. Is the 0.08 a standard deviation or standard error of the mean? The graphical representation in Fig. 6A seems to show much broader error bars.

Extended Data Fig 6B: The data are fit to a linear model, but the data for ICM radius versus size factor appear to be non-linear, with the slope flattening at larger size factors. In accord with expected dimensional scaling (see below), and given that size factor is based on a comparison of cell numbers or ICM volume, one would expect the ICM radius to scale as (size factor)^{1/3} or (size factor)^{1/2} depending on the blastocyst shape. The author's conclusion that "ICM radius scales linearly with embryo size factor in size-manipulated E4.5 blastocysts" is not well supported by this data. The p-value is very low, but that p-value only tells you that the best-fit slope is significantly different from zero, not that a linear model is the most appropriate.

E. Conclusions: robustness, validity, reliability

There is one section of the paper where I have strong doubts about the validity of the conclusions. In "Differential surface tension can sort EPI and PrE cells, but only for normal-size ICM," the first half of the conclusion is well-supported, but the second half is not. The authors measure the sorting dynamics for normal-size ICMs and tune a cellular Potts model to match those dynamics. They then run that model for larger ICMs and conclude that surface tension alone is not sufficient to sort the larger ICMs within the developmental/physiological window available; however, the physiological window is ~12 hrs, sorting of normal-size embryos took 2.5 hrs and the authors state that "larger embryos took 4-5 hours longer and achieved slightly lower sorting scores." Assuming the extrapolation of the model is valid, then the timing seems to work out to allow surface-tension-based sorting within 12 hrs, albeit perhaps to slightly less complete sorting. All of this assumes that the model and its non-unique fitted parameters can be extrapolated appropriately. I just do not find this argument convincing.

That said, the argument above is only used to justify looking for additional mechanisms beyond surface-tension-based sorting. The authors do show that there are additional mechanisms (ECM-gradient-based directed cell migration), so perhaps the model-based justification to look for additional mechanisms just isn't needed in this paper.

The author's conclusions regarding the importance and optimality of PrE/EPI cell proportions seem fairly weak, especially when those species-specific proportions are held up as being key to robustness with respect to natural size variations. It appears that the key to robustness is not the fixed proportions (which actually hinder robustness), but the flexibility in PrE cell shape and how the same number of PrE cells can cover a wide range of ICM surface area by changes in how flat or rounded these cells are.

F. Suggested improvements: experiments, data for possible revision

Given that there are two distinct sets of key findings and limited space to describe them all, the authors should give careful consideration to focusing this paper on the first set related to cellular and biophysical mechanisms of PrE/EPI cell sorting. The second set of findings on cell-fate specification are not as strong and detract from the paper's ability to fully and clearly explore the first set of findings.

If the parts about cell-fate specification remain, the analysis and discussion of how PrE-monolayer formation is maintained in different sized embryos really should have some discussion of dimensional scaling. The plots in Fig 7B, G and J all compare interface area to number of cells in ICM. Since the latter will be proportional to the volume of the ICM (assuming average cell volume doesn't change), then this is a question of surface-area-to-volume ratio, where one would naturally expect something like a $1/r$ scaling. With the current plots, one would thus expect surface area to scale as (volume)^{2/3} power. In accord with that expectation, there is some flattening of the data trendlines at larger numbers of cells; the trend in Fig. 7B is certainly not linear. It would be very useful to plot the results on a log-log scale to see if the expected power laws hold or not. The scaling may hold within a species, but not among species. It is worth noting that the blastocysts get larger from mouse to monkey to human, but the PrE cell fraction does not follow the same trend (rising from 0.606 to 0.702 from mouse to monkey, but then falling to 0.554 for human). Scaling analysis and arguments could add substantial clarity to this part of the paper.

It would also help to label panels B, G and J of Fig. 7 with the species from which the data was obtained.

Lines 397-399: "Despite the fixed proportion of EPI/PrE cells, patterning is robust against naturally variable sizes of the embryo because this proportion is species-specific and optimal for embryo size and geometry." This sentence overstates the importance of having a species-specific optimal EPI/PrE proportion for robustness to natural variations in embryo size. The robustness is not really from the fixed proportion, but from the adjustability of PrE cell shape (flattening) at the ICM surface.

The surface tension values used in the cellular Potts model (Table S2) are surprisingly large compared to kT , which suggests that cell movements under this model would be quite deterministic. I can see the value in placing the exact parameter values in a supplemental table, but it would be helpful for the main text to include some mention of these parameters, including their order of magnitude relative to kT and the relative ordering of $J_{EPI:EPI}$, $J_{EPI:PrE}$ and $J_{PrE:PrE}$, which would influence the sorting dynamics.

G. References: appropriate credit to previous work?

The references appear to give appropriate credit to previous work.

H. Clarity and context: lucidity of abstract/summary, appropriateness of abstract, introduction and conclusions.

The paper's Introduction does not contain a clear thesis statement about the paper's important findings. Instead, it says that "In this study, we systematically analyse cellular dynamics, position, fate, and polarity using reduced systems and blastocyst manipulation to gain mechanistic insights into ICM patterning robustness during mouse pre-implantation development." This statement makes the paper seem more descriptive than it actually is once you get into the meat of the results. The abstract does a much better job summarizing the key findings.

Reviewer #3:

Remarks to the Author:

In this paper, Hiragi and colleagues investigated the segregation of PrE cells to the outside of the ICM in mouse embryos, supplemented with additional analysis in monkey and human blastocysts to ensure robustness of the results. The authors perform a number of analyses, mostly with genetic/small molecule perturbations followed by imaging and advanced quantitative analysis, and some physical modeling, to make several claims about PrE segregation. The main claims are that the segregation is due to Rac1/aPKC – dependent directed migration towards the outside of the ICM, and that they are stuck there due to a decreased tension. The authors also show a gradient of laminin secreted by the PrE cells and claim that the PrE cells are using this as a pathway to crawl out of the ICM using the aforementioned mechanisms. The authors claim this is the first paper to show a coupling between the acquisition of PrE fate and spatial segregation, though this is not really the case. Indeed, there are very few claims in this paper that are genuinely novel and it's difficult to see a major conceptual advance. What is true is that this paper does a better job than its predecessors to quantify dynamics within embryos, and though that is to be commended, I am not sure that fact is enough to carry this paper for consideration for Nature Cell Biology. Following are a few comments on the claims.

1. (Fig. 1) The ex vivo segregation of blastocysts has been shown numerous times, including cited papers such as Wigger et al, and the Plusa et al paper led a number of papers that have shown high-fidelity sorting of PrE from the ICM.
2. (Fig. 2) Differential surface tension as a means of either sorting cells or keeping them on the outside of an aggregate has been shown numerous times across many different organisms, and AFM has been performed on EPI and PrE to show differences in surface mechanics previously, and active myosin stains have been shown before too. The current paper uses a better technique to show differences in tension using micropipette aspiration but overall differential surface tension is not a new concept. The correlation shown in Fig. 2D is also unconvincing.
3. (Fig. 3) The protrusion analysis has been shown at least in part by Meilhac et al (also cited in the paper) and migration has been proposed before in that paper and others. Again this paper does a better job of quantifying it, but it's not a new concept. And the shape analysis was performed in Yanagida et al – indeed this paper doesn't preclude the possibility this is enhanced fluidity of PrE as claimed in that paper. Much of what is shown in this paper could suggest a tension gradient through the whole ICM, which together with a change in cell mechanics (fluidity or otherwise) would explain why the extensions of the cell would go outward. Indeed, one might expect due to geometry that there is a stress gradient in an aggregate, particularly if the cells on the inside are more adhesive. The effects of Latrunculin are way too messy, and Rac1 affects many things (including surface tension itself, and it negatively regulates RhoA activity, among a great many things) to make this convincing.
4. (Fig. 4) The potentially most novel aspect of this paper is connection to aPKC and polarity, but the analysis isn't that thorough. Each of the perturbations can affect many things, and of course they do see smaller ICMs with the genetic perturbations. Overall, this isn't a convincing contribution to what's known, particularly in light of the work of Clare Chazaud and lab on the importance of polarity in PrE. Moreover, aPKC and polarity does not explain changes in mechanics in itself. What mechanical changes are associated? The authors may claim it's to facilitate migration but this reviewer is not particularly convinced by that data.

5. (Fig. 5) The differences in integrins and ECM have been noted by these same authors previously in a very nice paper (E.J.Y. Kim et al, 2022). However, ECM and integrin sensing of it will also affect cell fate, and there is very little evidence of the fact that the ECM is forming the sort of scaffolding that would act as a 'highway' for cells, instead of acting to promote a different kind of cell-cell adhesion or signalling hub for PrE. Indeed, ECM-integrin sensing is very difficult to prove as a mechanism for active migration in jammed tissue, particularly since it plays such a profound role in mitogenic signalling such as FGF signalling. Unfortunately, this part is not convincing and I do not see how it could become convincing, because even disrupting the ECM would cause many effects.
6. The remainder of the paper about size and robustness is potentially interesting, but doesn't hold up in light of the rest of the paper.

I do believe this paper has something to offer it and represents an incremental advance in what is known on this difficult and important topic. I simply do not see the conceptual advance the authors appear to be claiming. I am also not convinced that doing a significant number of experiments to address the above concerns would overcome the sense that the paper is not adding a great deal to what is already known.

Methods should be written concisely, but should contain all elements necessary to allow interpretation and replication of the results. As a guideline, Methods sections typically do not exceed 3,000 words. The Methods should be divided into subsections listing reagents and techniques. When citing previous methods, accurate references should be provided and any alterations should be noted. Information must be provided about: antibody dilutions, company names, catalogue numbers and clone numbers for monoclonal antibodies; sequences of RNAi and cDNA probes/primers or company names and catalogue numbers if reagents are commercial; cell line names, sources and information on cell line identity and authentication. Animal studies and experiments involving human subjects must be

reported in detail, identifying the committees approving the protocols. For studies involving human subjects/samples, a statement must be included confirming that informed consent was obtained. Statistical analyses and information on the reproducibility of experimental results should be provided in a section titled "Statistics and Reproducibility".

All Nature Cell Biology manuscripts submitted on or after March 21 2016 must include a Data availability statement at the end of the Methods section. For Springer Nature policies on data availability see <http://www.nature.com/authors/policies/availability.html>; for more information on this particular policy see <http://www.nature.com/authors/policies/data/data-availability-statements-data-citations.pdf>. The Data availability statement should include:

- Accession codes for primary datasets (generated during the study under consideration and designated as "primary accessions") and secondary datasets (published datasets reanalysed during the study under consideration, designated as "referenced accessions"). For primary accessions data should be made public to coincide with publication of the manuscript. A list of data types for which submission to community-endorsed public repositories is mandated (including sequence, structure, microarray, deep sequencing data) can be found here <http://www.nature.com/authors/policies/availability.html#data>.
- Unique identifiers (accession codes, DOIs or other unique persistent identifier) and hyperlinks for datasets deposited in an approved repository, but for which data deposition is not mandated (see here for details <http://www.nature.com/sdata/data-policies/repositories>).
- At a minimum, please include a statement confirming that all relevant data are available from the authors, and/or are included with the manuscript (e.g. as source data or supplementary information), listing which data are included (e.g. by figure panels and data types) and mentioning any restrictions on availability.
- If a dataset has a Digital Object Identifier (DOI) as its unique identifier, we strongly encourage including this in the Reference list and citing the dataset in the Methods.

We recommend that you upload the step-by-step protocols used in this manuscript to the Protocol Exchange. More details can be found at www.nature.com/protocolexchange/about.

All imaging data should be accompanied by scale bars, which should be defined in the legend. Cropped images of gels/blots are acceptable, but need to be accompanied by size markers, and to retain visible background signal within the linear range (i.e. should not be saturated). The boundaries of panels with low background have to be demarked with black lines. Splicing of panels should only be considered if unavoidable, and must be clearly marked on the figure, and noted in the legend with a statement on whether the samples were obtained and processed simultaneously. Quantitative comparisons between samples on different gels/blots are discouraged; if this is unavoidable, it should only be performed for samples derived from the same experiment with gels/blots were processed in parallel, which needs to be stated in the legend.

All placed images (i.e. a photo incorporated into a figure) should be on a separate layer and independent from any superimposed scale

bars or text. Individual photographic images must be a minimum of 300+ DPI (at actual size) or kept constant from the original picture acquisition and not decreased in resolution post image acquisition. All colour artwork should be RGB format.

The total number of Supplementary Figures (not including the "unprocessed scans" Supplementary Figure) should not exceed the number of main display items (figures and/or tables (see our Guide to Authors and March 2012 editorial <http://www.nature.com/ncb/authors/submit/index.html#suppinfo>; <http://www.nature.com/ncb/journal/v14/n3/index.html#ed>). No restrictions apply to Supplementary Tables or Videos, but we advise authors to be selective in including supplemental data.

GUIDELINES FOR EXPERIMENTAL AND STATISTICAL REPORTING

REPORTING REQUIREMENTS – To improve the quality of methods and statistics reporting in our papers we have recently revised the reporting checklist we introduced in 2013. We are now asking all life sciences authors to complete two items: an Editorial Policy Checklist (found here <https://www.nature.com/authors/policies/Policy.pdf>) that verifies compliance with all required editorial policies and a reporting summary (found here <https://www.nature.com/authors/policies/ReportingSummary.pdf>) that collects information on experimental design and reagents. These documents are available to referees to aid the evaluation of the manuscript. Please note that these forms are dynamic 'smart pdfs' and must therefore be downloaded and completed in Adobe Reader. We will then flatten them for ease of use by the reviewers. If you would like to reference the guidance text as you complete the template, please access these flattened versions at <http://www.nature.com/authors/policies/availability.html>.

STATISTICS – Wherever statistics have been derived the legend needs to provide the n number (i.e. the sample size used to derive statistics) as a precise value (not a range), and define what this value represents. Error bars need to be defined in the legends (e.g. SD, SEM) together with a measure of centre (e.g. mean, median). Box plots need to be defined in terms of minima, maxima, centre, and percentiles. Ranges are more appropriate than standard errors for small data sets. Wherever statistical significance has been derived, precise p values need to be provided and the statistical test used needs to be stated in the legend. Statistics such as error bars must not be derived from n<3. For sample sizes of n<5 please plot the individual data points rather than providing bar graphs. Deriving statistics from technical replicate samples, rather than biological replicates is strongly discouraged. Wherever statistical significance has been derived, precise p values need to be provided and the statistical test stated in the legend.

Version 1:

Decision Letter:

*Please delete the link to your author homepage if you wish to forward this email to co-authors.

Dear Dr Hiiragi,

I apologize for the delay. Reviewer #3 was initially delayed in submitting their report; after this we sought the further advice of an additional reviewer (Reviewer #4 with expertise in mouse development and mechanobiology) to comment on Reviewer #3's current concerns. I do apologize for the extended review process.

Your manuscript, "Optimality of cell, matrix, and tissue dynamics coupling ensures patterning robustness", has now been seen by our original referees, who are experts mouse embryonic development (referee 1); biomechanics and modelling (referee 2); and mechanics and cell fate (referee 3), as well as the additional feedback of referee 4 to comment on referee 3's current concerns. As you will see from their comments (attached below) they find this work of interest, but have raised some important points. Although we are also very interested in this study, we believe that their concerns should be addressed before we can consider publication in Nature Cell Biology.

Nature Cell Biology editors discuss the referee reports in detail within the editorial team, including the chief editor, to identify key referee points that should be addressed with priority, and requests that are overruled as being beyond the scope of the current study. To guide the scope of the revisions, I have listed these points below. We are committed to providing a fair and constructive peer-review process, so please feel free to contact me if you would like to discuss any of the referee comments further.

We are willing to allow one more round of revision to address the referees' concerns. Please be aware that further reconsideration of this manuscript (in the event of re-review) will be conditional on the referees being fully satisfied with the extent of the revisions, as we generally do not encourage multiple review rounds at this journal

In particular, it would be essential to:

A) Experimentally assess the causative links between ECM polarity and PrE cell migration as outlined by Reviewers #3 and #4 (and see experimental suggestion by Reviewer #4).

B) All other referee concerns pertaining to strengthening existing data, providing controls, methodological details, clarifications and textual changes, should also be addressed.

C) Finally please pay close attention to our guidelines on statistical and methodological reporting (listed below) as failure to do so may delay the reconsideration of the revised manuscript. In particular please provide:

We therefore invite you to take these points into account when revising the manuscript. In addition, when preparing the revision please:

- ensure that it conforms to our format instructions and publication policies (see below and <https://www.nature.com/nature/for-authors>).

- provide a point-by-point rebuttal to the full referee reports verbatim, as provided at the end of this letter.

- provide the completed Reporting Summary (found here <https://www.nature.com/documents/nr-reporting-summary.pdf>) <https://www.nature.com/documents/nr-reporting-summary.pdf>). This is essential for reconsideration of the manuscript and will be available to editors and referees in the event of peer review. For more information see <http://www.nature.com/authors/policies/availability.html> or contact me.

Nature Cell Biology is committed to improving transparency in authorship. As part of our efforts in this direction, we are now requesting that all authors identified as 'corresponding author' on published papers create and link their Open Researcher and Contributor Identifier (ORCID) with their account on the Manuscript Tracking System (MTS), prior to acceptance. ORCID helps the scientific community achieve unambiguous attribution of all scholarly contributions. You can create and link your ORCID from the home page of the MTS by clicking on 'Modify my Springer Nature account'. For more information please visit <http://www.springernature.com/orcid>.

This journal strongly supports public availability of data. Please place the data used in your paper into a public data repository, or

alternatively, present the data as Supplementary Information. If data can only be shared on request, please explain why in your Data Availability Statement, and also in the correspondence with your editor. Please note that for some data types, deposition in a public repository is mandatory - more information on our data deposition policies and available repositories appears below.

Link Redacted

We would like to receive the revision within four weeks. If submitted within this time period, reconsideration of the revised manuscript will not be affected by related studies published elsewhere, or accepted for publication in Nature Cell Biology in the meantime. We would be happy to consider a revision even after this timeframe, but in that case we will consider the published literature at the time of resubmission when assessing the file.

We hope that you will find our referees' comments, and editorial guidance helpful. Please do not hesitate to contact me if there is anything you would like to discuss.

Best wishes,

Daryl

Daryl Jason Verzosa David, PhD

Senior Editor, Nature Cell Biology
Advisory Editor, npj Biological Physics and Mechanics
Nature Portfolio

Heidelberger Platz 3, 14197 Berlin, Germany
Email: daryl.david@nature.com
ORCID: <https://orcid.org/0000-0002-9253-4805>

Reviewers' Comments:

Reviewer #1:

Remarks to the Author:

The authors have nicely addressed the point raised by providing additional supporting data.

Reviewer #2:

Remarks to the Author:

The revised paper is substantially improved and the authors have largely addressed my concerns. I do have three minor and easily correctable concerns.

1. I suspect this may be a typo. In the main text on Lines 332-333, the authors state that the proportion of PrE cells isolated from blastocyst and ICMs is 0.606 ± 0.08 for $n = 106$ embryos. The authors also point to Fig 6A, but the caption to that figure says that the mean proportion of PrE cells is 0.605 ± 0.78 for a set of 4 groups with $n = 19, 29, 21$ and 32 embryos (which adds up to 101 embryos). The authors should clarify and/or fix the discrepancy.

2. In the caption to Fig 2D, the slope of the linear regression line is given as -431.58 ± 69.33 . Given the size of the standard error, both values are presented with more than the necessary significant figures. The slope should also have units.

3. The values presented in Table S3 also have more than the necessary significant figures.

Reviewer #3:

Remarks to the Author:

I respectfully disagree with the authors that my concerns can be mostly reduced to a focus 'on integration of prior research into our study'. This was not my primary concern or even a major concern. In laying out my review, I was attempting to make clear that a lot of what the paper was putting forward was already more-or-less covered in the literature. As I said, and the authors reiterated in their response, it is better quantified and characterised than before and the authors really deserve a lot of credit for this. I do believe the authors had already referenced key papers and they have done a better job of discussing them now, and it is more than sufficient.

Better integration into the literature simply was not my main concern. My main point was that, given that a decent amount of the paper content represents a better characterisation of things that are already known, the novelty hinges on: 1. the idea that PrE cells undergo active, directed migration via ECM gradients; and 2. The data presented in Fig. 7 showing that EPI/PrE proportions are optimal for the

size and geometry of the ICM. Fig. 7 is a very interesting result and they have expanded on it with new species. I personally do not believe there is enough there in Fig. 7 in terms of insight and mechanism to carry the whole paper, but if the editor and other reviewers believe it is, I understand that completely.

However, I think almost anyone reading this paper would strongly believe that for this paper to be fully convincing as a major advance, it is essential that the authors have sufficiently shown that PrE is undergoing directed, active migration towards the cavity. My main concern before, which remains after the revisions, is that the idea of active, directed migration of PrE along ECM gradients is not convincing, and I do not see what would make it convincing.

1. The protrusion angle data (Fig. 3F) is relatively unconvincing – it certainly does not look like that alone could convince anyone that PrE cells are migrating to the surface robustly and that this is the main reason they get there. The particular geometry (with more space towards the cavity) and boundary conditions of the ICM would suggest there should be some bias towards the cavity, and it is hard to imagine their data is reflecting anything more than that. In fact, if there was a highway of ECM leading these cells out, I would expect a much stronger bias.

2. All of the perturbations – aPKC, Rac1, latrunculin, CK666 – have many feedbacks with cell surface mechanics and are too confounded to make the case by themselves or collectively. This is always a problem and I do not really know how it could be resolved. It could be good supporting data if there were otherwise strong data supporting their claims. The aPKC polarisation index shown in Fig. 4 is slightly more convincing, but given that polarity is tied up with cell surface mechanics, and there has never been any doubt that EPI cells stick together well and would act as an adhesion barrier for PrE cells, it is very difficult to interpret what that polarity might mean and if it really can be a proxy for migration bias. If EPI cells have more E-cadherin or other surface factor, this could easily explain how the PrE cells might polarise in this manner, but again it does not mean they are actively migrating, much less along an ECM gradient.

3. PrE cells express different integrins than EPI cells and they are more likely to adhere to ECM like laminin, of that there is no doubt. But laminin-integrin binding initiates signalling that is essential for PrE cells, like ERK. I am still unconvinced, even with the new data, that a secreted laminin gradient is causal. As another reviewer pointed out, it could be just a consequence of the PrE cells having an outside bias. The new bead data is not convincing, the PrE cells could just get stuck on them – unless I am missing something there is no evidence they are causing PrE cells to migrate to them. And the rescue they showed is likely to change signalling in the ICM by now having cells that secrete laminin that will likely make the initiation of the PrE identity more robust. The authors did not rule out that very likely possibility. In sum, ECM is extremely important for signalling. It is notoriously difficult to dissect the role of ECM in mechanics/migration etc and its role in intracellular signalling. I do not see evidence here that challenges what should be the null hypothesis: that ECM is integral to the robustness of the mechanical phenotype and identity of PrE cells and it does not necessarily have anything to do with migration.

This may sound like nitpicking. The problem as I see it (again) is not that the authors have not paid proper due to previous literature. It is that previous literature has proposed different sorting mechanisms that revolve around the idea that the cells have different mechanical properties and, given the geometry and the constant perturbations in the ICM like cell divisions and blastocyst contractions, are likely to be sufficient to explain sorting. If the authors want to propose a different mechanism than what has been proposed, the burden of proof is high. In the end, I do not have any reason to believe these cells would actively migrate along an ECM gradient. It seems implausible to me. I believe the ECM gradient is incidental and its role in instructive signalling may indeed be very important. However, I am absolutely open to someone turning my preconceived notions around: I would be delighted for the authors to prove me wrong and I would accept it wholeheartedly if they do. The burden of proof, however, is on the authors. As I said in previous review, this is a very, very tricky problem to solve. I do not have specific recommendations to solve the pleiotropy/confounding problem with ECM particularly in such a size and geometry.

There is, however, very good and well-characterised data in this paper and it has much to offer the field, and the size robustness data is interesting. If the authors used extensive mitigating language around their directed migration data and put it forward as a hypothesis more than a well-supported finding, I would support publication. I just do not know if the journal would be happy with that outcome.

Reviewer #4:

Remarks to the Author:

This is a technically strong manuscript that uses beautiful imaging to quantify and visualize the spatial segregation of ICM cells. The theory is very elegant and the optimality implicated in the theory-driven approach in figure 7 are intriguing but also slightly disconnected from the rest of the manuscript.

A panel of previous work also from the authors themselves have identified cellular mechanisms that contribute to this process. The main new findings of the manuscript, also according to the authors own statements, are the Rac1-mediated polarized migration and migration along a self-generated laminin gradient. While the polarization is convincingly demonstrated, its role in directed migration is very well established across cell biological systems. The migration along a ECM gradient is an exciting new observation but also much harder to demonstrate, and remains somewhat speculative despite the authors efforts. Migrating cells will typically deposit ECM as they go, and therefore the cause and consequence relationship is difficult to establish. The fact that ectopic laminin can promote directed migration it does not provide evidence that this is the mechanism that the cells in fact utilize.

To conclusively demonstrate the causative effect of the ECM, I would recommend that the authors make use of available laminin-reporter lines and perform live imaging experiments to demonstrate that laminin deposition is an early cellular event that occurs at the onset of cell motility, rather than a process that occurs after cells have already translocated their cell bodies along the trajectory.

GUIDELINES FOR SUBMISSION OF NATURE CELL BIOLOGY ARTICLES

ARTICLE FORMAT

ABSTRACT – should not exceed 150 words and should be unreferenced. This paragraph is the most visible part of the paper and should briefly outline the background and rationale for the work, and accurately summarize the main results and conclusions. Key genes, proteins and organisms should be specified to ensure discoverability of the paper in online searches.

TEXT – the main text consists of the Introduction, Results, and Discussion sections and must not exceed 3500 words including the abstract. The Introduction should expand on the background relating to the work. The Results should be divided in subsections with subheadings, and should provide a concise and accurate description of the experimental findings. The Discussion should expand on the findings and their implications. All relevant primary literature should be cited, in particular when discussing the background and specific findings.

REFERENCES – are limited to a total of 70 in the main text and Methods combined,. They must be numbered sequentially as they appear in the main text, tables and figure legends and Methods and must follow the precise style of Nature Cell Biology references. References only cited in the Methods should be numbered consecutively following the last reference cited in the main text. References only associated with Supplementary Information (e.g. in supplementary legends) do not count toward the total reference limit and do not need to be cited in numerical continuity with references in the main text. Only published papers can be cited, and each publication cited should be included in the numbered reference list, which should include the manuscript titles. Footnotes are not permitted.

Methods should be written concisely, but should contain all elements necessary to allow interpretation and replication of the results. As a guideline, Methods sections typically do not exceed 3,000 words. The Methods should be divided into subsections listing reagents and techniques. When citing previous methods, accurate references should be provided and any alterations should be noted. Information must be provided about: antibody dilutions, company names, catalogue numbers and clone numbers for monoclonal antibodies; sequences of RNAi and cDNA probes/primers or company names and catalogue numbers if reagents are commercial; cell line names, sources and information on cell line identity and authentication. Animal studies and experiments involving human subjects must be reported in detail, identifying the committees approving the protocols. For studies involving human subjects/samples, a statement must be included confirming that informed consent was obtained. Statistical analyses and information on the reproducibility of experimental results should be provided in a section titled "Statistics and Reproducibility".

All Nature Cell Biology manuscripts submitted on or after March 21 2016, must include a Data availability statement as a separate section after Methods but before references, under the heading "Data Availability". For Springer Nature policies on data availability see <http://www.nature.com/authors/policies/availability.html>; for more information on this particular policy see <http://www.nature.com/authors/policies/data/data-availability-statements-data-citations.pdf>. The Data availability statement should include:

- Accession codes for primary datasets (generated during the study under consideration and designated as "primary accessions") and secondary datasets (published datasets reanalysed during the study under consideration, designated as "referenced accessions"). For primary accessions data should be made public to coincide with publication of the manuscript. A list of data types for which submission to community-endorsed public repositories is mandated (including sequence, structure, microarray, deep sequencing data) can be found here <http://www.nature.com/authors/policies/availability.html#data>.
- Unique identifiers (accession codes, DOIs or other unique persistent identifier) and hyperlinks for datasets deposited in an approved repository, but for which data deposition is not mandated (see here for details <http://www.nature.com/sdata/data-policies/repositories>).

- At a minimum, please include a statement confirming that all relevant data are available from the authors, and/or are included with the manuscript (e.g. as source data or supplementary information), listing which data are included (e.g. by figure panels and data types) and mentioning any restrictions on availability.
- If a dataset has a Digital Object Identifier (DOI) as its unique identifier, we strongly encourage including this in the Reference list and citing the dataset in the Methods.

We recommend that you upload the step-by-step protocols used in this manuscript to [protocols.io](https://www.protocols.io/help/publish-articles). More details can be found at <https://www.protocols.io/help/publish-articles>.

DISPLAY ITEMS – main display items are limited to 6-8 main figures and/or main tables. For Supplementary Information see below.

FIGURES – Colour figure publication costs \$395 per colour figure. All panels of a multi-panel figure must be logically connected and arranged as they would appear in the final version. Unnecessary figures and figure panels should be avoided (e.g. data presented in small tables could be stated briefly in the text instead).

All imaging data should be accompanied by scale bars, which should be defined in the legend.

Cropped images of gels/blots are acceptable, but need to be accompanied by size markers, and to retain visible background signal within the linear range (i.e. should not be saturated). The boundaries of panels with low background have to be demarked with black lines. Splicing of panels should only be considered if unavoidable, and must be clearly marked on the figure, and noted in the legend with a statement on whether the samples were obtained and processed simultaneously. Quantitative comparisons between samples on different gels/blots are discouraged; if this is unavoidable, it has to be performed for samples derived from the same experiment with gels/blots were processed in parallel, which needs to be stated in the legend.

Regardless of format, all figures must be vector graphic compatible files, not supplied in a flattened raster/bitmap graphics format, but should be fully editable, allowing us to highlight/copy/paste all text and move individual parts of the figures (i.e. arrows, lines, x and y axes, graphs, tick marks, scale bars etc). The only parts of the figure that should be in pixel raster/bitmap format are photographic images or 3D rendered graphics/complex technical illustrations.

Unprocessed scans of all key data generated through electrophoretic separation techniques need to be presented in a supplementary figure that should be labeled and numbered as the final supplementary figure, and should be mentioned in every relevant figure legend. This figure does not count towards the total number of figures and is the only figure that can be displayed over multiple pages, but should be provided as a single file, in PDF or TIFF format. Data in this figure can be displayed in a relatively informal style, but size markers and the figures panels corresponding to the presented data must be indicated.

The total number of Supplementary Figures (not including the "unprocessed scans" Supplementary Figure) should not exceed the number of main display items (figures and/or tables (see our Guide to Authors and March 2012 editorial <http://www.nature.com/ncb/authors/submit/index.html#supinfo>; <http://www.nature.com/ncb/journal/v14/n3/index.html#ed>). No restrictions apply to Supplementary Tables or Videos, but we advise authors to be selective in including supplemental data.

GUIDELINES FOR EXPERIMENTAL AND STATISTICAL REPORTING

REPORTING REQUIREMENTS – We ask authors to complete a Reporting Summary that collects information on experimental design and reagents. We hope this will aid in your evaluation of the paper. The Reporting Summary can be found here <https://www.nature.com/documents/nr-reporting-summary.pdf> Please note that these forms are dynamic 'smart pdfs' and must therefore be downloaded and completed in Adobe Reader. We will then flatten them for ease of use. If you would like to reference the guidance text as you complete the template, please access these flattened versions at <http://www.nature.com/authors/policies/availability.html>.

Version 2:

Decision Letter:

Our ref: NCB-A51940B

7th October 2024

Dear Dr. Hiiragi,

Thank you for submitting your revised manuscript "Optimality of cell, matrix, and tissue dynamics coupling ensures patterning robustness" (NCB-A51940B). It has now been seen by the original referee(s) and their comments are below. The reviewers find that the paper has improved in revision, and therefore we'll be happy in principle to publish it in Nature Cell Biology, pending revisions to satisfy the referees' final requests and to comply with our editorial and formatting guidelines.

In particular, in order for us to proceed further, we will require toning down conclusions about any definitive role of matrix gradients in symmetry breaking with textual revisions throughout your title, abstract, and text (Reviewer #4).

Thank you again for your interest in Nature Cell Biology Please do not hesitate to contact me if you have any questions.

Sincerely,
Daryl

Daryl Jason Verzosa David, PhD

Senior Editor, Nature Cell Biology
Advisory Editor, npj Biological Physics and Mechanics
Nature Portfolio

Heidelberger Platz 3, 14197 Berlin, Germany
Email: daryl.david@nature.com
ORCID: <https://orcid.org/0000-0002-9253-4805>

Reviewer #4 (Remarks to the Author):

I appreciate that the authors have attempted to strengthen the link between matrix deposition and protrusions using immunostainings of fixed tissues. I understand that the live imaging approaches are challenging although I do somewhat disagree with the authors assertion that live imaging would have provided the same level of proof as this chosen approach, as one could have been able to obtain stronger temporal correlations and statistical probabilities. Also, from the images provided it is difficult to see clear protrusions so the question of causality remains somewhat unresolved. I would recommend that the authors tone down the conclusions of the definitive role of "matrix gradients" in the symmetry breaking.

Version 3:

Decision Letter:

Dear Dr Hiiragi,

I am pleased to inform you that your manuscript, "Coupling of cell shape, matrix, and tissue dynamics ensures embryonic patterning robustness", has now been accepted for publication in Nature Cell Biology.

Please note that *Nature Cell Biology* is a Transformative Journal (TJ). Authors may publish their research with us through the traditional subscription access route or make their paper immediately open access through payment of an article-processing charge (APC). Authors will not be required to make a final decision about access to their article until it has been accepted. > Find out more about Transformative Journals

If you have not already done so, we strongly recommend that you upload the step-by-step protocols used in this manuscript to protocols.io (<https://protocols.io>), an open online resource that allows researchers to share their detailed experimental know-how. All uploaded protocols are made freely available and are assigned DOIs for ease of citation. Protocols and Nature Portfolio journal papers in which they are used can be linked to one another, and this link is clearly and prominently visible in the online versions of both. Authors who performed the specific experiments can act as primary authors for the Protocol as they will be best placed to share the methodology details, but the Corresponding Author of the present research paper should be included as one of the authors. By uploading your Protocols onto protocols.io, you are enabling researchers to more readily reproduce or adapt the methodology you use, as well as increasing the visibility of your protocols and papers. You can also establish a dedicated workspace to collect your lab Protocols. Further information can be found at <https://www.protocols.io/help/publish-articles>.

Nature Cell Biology encourages authors presenting evidence for cell, biological, molecular, and genetic interactions to consider communicating these findings using Biofactoid (<https://biofactoid.org/>). This tool helps users share a searchable representation of interactions (e.g. binding, gene expression, post-translational modification) between genes, gene products, or chemicals. Information added to Biofactoid, with author attribution, is shared on social media and public databases, such as Pathway Commons, where it can be discovered and analyzed in the context of a large and growing corpus of knowledge.

With kind regards,

Daryl

Daryl Jason Verzosa David, PhD

Senior Editor, Nature Cell Biology
Advisory Editor, npj Biological Physics and Mechanics
Nature Portfolio

Heidelberg Platz 3, 14197 Berlin, Germany
Email: daryl.david@nature.com
ORCID: <https://orcid.org/0000-0002-9253-4805>

** Visit the Springer Nature Editorial and Publishing website at http://editorial-jobs.springernature.com?utm_source=ejp_NCB_email&utm_medium=ejp_NCB_email&utm_campaign=ejp_NCB for more information about our career opportunities. If you have any questions please click [here](mailto:editorial.publishing.jobs@springernature.com).

Responses to Editorial Comments

I should stress that the referees' concerns point to potential pleiotropic effects and unclear mechanistic links which would need to be addressed with experiments and data, and reconsideration of the study for this journal and re-engagement of referees would depend on strength of these revisions.

In particular, it would be essential to:

A) Experimentally assess the contribution of ECM on embryonic sorting and migration (all Reviewers)

We performed two sets of additional experiments to test the functional role of the ECM gradient in PrE/EPI cell sorting. Specifically:

1. We experimentally generated ectopic ECM accumulation by implanting microbeads coated with laminin into the ICM of the blastocyst, and evaluated the distribution of PrE cells at the stage E4.5. In agreement with our model prediction, PrE cells were misguided by the ectopic laminin localisation (presented as new Figure 5E).
2. We experimentally rescued the incorrectly patterned phenotype of *Lamc1*^{-/-} by generating chimeric embryos that include wild-type (WT) cells. The resulting laminin deposition by these WT cells was sufficient to guide migration of *Lamc1*^{-/-} PrE cells to the cavity-surface monolayer (presented as new Figure 5F).

B) Further assess the effects on cell fate specification, taking into account interspecies considerations and scaling (Reviewer #2). Please note that although Reviewer #2 suggests removing cell fate specification analysis from your manuscript, we consider this essential for consideration at Nature Cell Biology and would ask for further analysis.

We substantially improved the clarity of our presentation of these points including Figure 7, with additional analyses of the surface-to-volume ratios. Note that we are not imposing a linear model, which would indeed be inadequate, for the surface-to-volume scaling in our results on the relationship between embryo size/shape and cell fate proportions (Figure 7). Our point is precisely that the surface-to-volume ratio scales non-linearly with total cell number, as the data show; whereas a fixed fate ratio implies a linear scaling of the range of surface areas that *can* be covered by PrE cells, based on their capacity to stretch, in relation to various embryo sizes depicted by the blue region in Figure 7B. The interesting aspect is that the range of sizes over which an embryo can build a confluent PrE monolayer also depends on embryo shape, with our results suggesting an adaptation between size/shape and fate ratio across different species.

C) Experimentally assess the potential role(s) of mechanics and material properties of the different tissues (Reviewers #2 and #3)

Following Reviewer #2's suggestions, we performed additional micropipette measurement (revised Figure 2D) and included a more detailed discussion of the surface tension parameters and their experimental estimates in the main text.

Reviewer #3 refers to the PrE/EPI sorting mechanisms proposed by Yanagida et al. (2022) (point 3). To assess the distinct role of the isotropic surface fluctuations they focused on, and the role of directed polarity-dependent protrusions we propose in this study, we examined the mechanistic role of the apical polarity in cell migration (Reviewer #3, points 3 and 4) by pharmacologically perturbing aPKC function in the embryo. The aPKC inhibition disrupted the directed movement of PrE cells towards the fluid interface, with PrE cells in these embryos lacking protrusions (presented as new Extended Data Figure 4E and F).

D) All other referee concerns pertaining to strengthening existing data, providing controls, methodological details, clarifications and textual changes, as well as appropriate citation and discussion of relevant literature should also be addressed.

We addressed all other concerns by additional experiments, analyses, and/or discussion accordingly, as specified in detail below.

Specific responses to Reviewers' Comments

Reviewer comments in blue

Author responses in black

Reviewer #1

In this manuscript the authors analyse the mechanisms that help ensure the precision of patterning during mammalian pre-implantation development. For this they study the dynamics of segregation of the epiblast (EPI) and primitive endoderm (PrE). They find that PrE and EPI cells sort by a combination of differential membrane tension, that pushes EPI cells away from the surface, and active migration, that directs. Furthermore, they also observe that as PrE cells move to the surface of the embryo they establish a gradient of ECM. Finally, they provide evidence that the proportion of EPI to PrE cells is species and dependant on the size and geometry of the embryo.

This is a very interesting study that sheds important light on the mechanics of pre-implantation development. The dissection of the different contributing factors is elegant and provides a convincing model. There is a point that would however benefit from clarification. The authors argue that the graded ECM deposition guides PrE migration. As the data currently stands, the differences in ECM localization could also be interpreted as being a consequence of PrE sorting, but not contributing to their migration. If the authors would like to make this statement, they would need to provide some functional data.

We appreciate the reviewer's interest in our study and thank the reviewer for the suggestion to dissect the role of ECM localisation. Accordingly, we have performed two additional experiments to address the functional role of the ECM distribution in PrE cell sorting.

First, we experimentally generated ectopic ECM accumulation in the ICM by implanting polymethyl methacrylate (PMMA) microbeads coated with laminin (see new Extended Data Figure 5F, copied below, for validation) into the ICM of the blastocyst, and evaluated the distribution of PrE cells at stage E4.5. Our model predicts that some PrE cells would be misguided by the ectopic laminin localisation, and we tested this with experiments. In control embryos with E-cadherin-coated microbeads, the final positioning of EPI and PrE cells is not perturbed (see below, Revised Figure 5E, top row). In contrast, when microbeads coated with E-cadherin and laminin are implanted into the ICM of the blastocyst, ectopic PrE cells localise to the coated microbead in E4.5 blastocysts (Revised Figure 5E, bottom row). We quantify the ectopic PrE cells localised at the coated microbead in the ICM but not in contact with the fluid-cavity, and compare them across the two groups. These data are in agreement with the prediction and demonstrate that laminin deposition is functionally sufficient for guiding the PrE cell migration. These data are presented as a new panel in Figure 5.

Revised Figure 5E

Figure 5E: Schematic for the experimental strategy to introduce ectopic laminin localisation in the ICM of the blastocyst using coated microbeads. Brightfield and immunofluorescence images of 2x E4.5 blastocysts with implanted beads coated with E-cadherin or E-cadherin+Laminin. Yellow asterisks and yellow dashed circles indicate microbead position in the ICM. Quantification of ectopic PrE cells localised at the coated beads in 2x E4.5 blastocysts. Mann-Whitney U-test, $p=0.0319$, $n=4,3$

embryos with successfully integrated E-cadherin coated beads and E-cadherin+Laminin coated beads, respectively.

Extended Data Figure 5F

F

Extended Data Figure 5F: Representative immunofluorescence and brightfield images of control and coated PMMA microbeads incubated with E-cadherin-Fc chimeric protein (left) and laminin protein (right).

Second, we performed experiments to rescue the incorrectly patterned phenotype of *Lamc1*^{-/-} previously reported in (Kim et al., 2022) by generating chimeric embryos that include several wild-type (WT) cells in the ICM. If the ECM plays a functional role in PrE cell migration, the resulting laminin deposition by these WT cells should be sufficient to guide *Lamc1*^{-/-} PrE cells to form a monolayer at the fluid interface. To test this prediction, we generated chimeric embryos in which the ICM comprises of both WT and *Lamc1*^{-/-} cells by aggregating blastomeres at the 4-cell stage (Revised Figure 5F). While PrE cells in late-stage *Lamc1*^{-/-} blastocysts tend to clump together (Kim et al., 2022 and revised Figure 5F), those in the chimeric embryos successfully form a segregated monolayer at the fluid interface (Revised Figure 5F), supporting the functional role of laminin in guiding PrE cell migration. These data are included as revised Figure 5F.

Revised Figure 5F

F

Figure 5F: Schematic diagram for the experimental strategy to rescue the incorrectly patterned phenotype of *Lamc1*^{-/-} blastocysts. Representative immunofluorescence images of late-stage blastocysts from *Lamc1*^{-/-} (left), and chimeric blastocysts between comprising *Lamc1*^{-/-} + WT cells (right). Bottom, zoomed-in images of the ICM. White arrowheads indicate *Lamc1*^{-/-} cells that are successfully segregated to the PrE monolayer at the fluid interface in the presence of WT cells in the ICM. Quantification of ectopic PrE cells in *Lamc1*^{-/-} blastocysts and *Lamc1*^{-/-} + WT chimeric blastocysts, n=8,5 embryos for the two groups, respectively. One-way ANOVA, $p=8.88e^{-03}$.

Reviewer #2

A. Summary of the key results

This paper focuses on the mechanisms by which internal embryonic epiblast (EPI) cells and primitive endoderm (PrE) cells organize themselves in mammalian blastocysts: an initially well-mixed population of the two cell types in the inner cell mass (ICM) sorts itself out so that the PrE cells form a monolayer in contact with a fluid cavity and the EPI cells cluster together between the PrE cells and the overlying trophectoderm. There are two primary sets of key results. The first focuses on the cellular / biophysical mechanisms by which the sorting takes place. These include key roles for long actin-based protrusions on PrE cells, polarization and reduced apical surface tension of PrE cells (that makes it more favorable for them to position themselves on the ICM surface), and a positive feedback mechanism in which secretion of extracellular matrix (ECM) from PrE cells gradually forms a gradient of ECM density that provides cues for the innermost PrE cells to direct their motility toward the ICM surface. The second set of key results focuses on cell fate specification within the ICM. One observation is that very few cells change from PrE to EPI or vice-versa based on their position within the ICM, i.e., there is little to no feedback from positional information to cell-fate specification. Another is that different species (mouse, monkey, human) with different-sized blastocysts seem to have evolved toward different PrE and EPI cell-fractions in the ICM that are optimal for forming monolayers of PrE cells over EPI cells – given that species’ specific blastocyst geometry. The cell-fractions are stable within a species even when blastocyst size varies, so there does not appear to be feedback between ICM size and cell-fractions. The robustness of the system with respect to making PrE surface monolayers for a wide range of surface-area-to-volume ratios is instead made possible by variations in the average surface area of PrE cells.

B. Originality and significance: if not novel, please include reference

The questions of how self-organization is accomplished in early blastocysts is clearly significant. There have of course been decades of studies on a variety of cell-sorting phenomenon, but the authors undertake a particularly detailed and thorough examination of the cellular / biophysical mechanisms here. The finding of a positive feedback between current cell position, ECM secretion, ECM density gradients, and directed motility of PrE cells is novel, as is the finding of an early role for PrE cell polarization even before PrE cells reach the ICM surface. The second set of key findings related to cell-fate specification within the ICM are less significant. In many ways, they are negative results showing that there is not really any influence on cell-fate specification from position within the ICM or size of the ICM. That does bring up an interesting question of whether the system is then ‘brittle’ in the face of intraspecies ICM- and blastocyst-size variation, but the flexibility of PrE cells to adopt a wide range of more- or less-flattened geometries, and thus surface areas, provides robustness. That is a useful observation, but it makes the initial question a bit less interesting.

We agree with the reviewer that the flexibility of PrE cell shape and surface area contributes to the robustness of PrE monolayer formation. However, we specifically demonstrate that this mechanism alone fails for larger size variations, resulting in ectopic EPI or PrE cells and disrupted patterning (Figure 6). These findings indeed initially seemed like a “negative result” to us too; however more in-depth quantitative analyses led us to a second, evolutionary mechanism to ensure patterning robustness – optimal coupling of fate proportion, cell dynamics, and cell and tissue size and shape. While feedbacks to cell fate specification have been studied and proposed in various developmental contexts (Chan et al., 2019; Korotkevich et al., 2017; Maître et al., 2016; Plusa et al., 2008), this optimal coupling of cellular and tissue parameters appears to be less discussed (Itzkovitz et al., 2012), therefore may represent an original finding.

C. Data & methodology: validity of approach, quality of data, quality of presentation

The experimental approaches are sound and well described, as are the detailed and quantitative analyses of the image sets. The figures are well constructed, and the writing is mostly clear.

There is one issue that decreases the quality of presentation: lots of relevant information is distributed to the figure captions and the Extended Data. This presentation makes the reader do a lot of work to put together the complete story line. Such decisions are often driven by length constraints, and I would

suggest, as I do below, that the authors consider focusing this paper on the cellular and biophysical mechanisms of PrE/EPI cell sorting, which would free up space to more clearly present these key findings.

We have revised the text in our manuscript and included essential information in the main text as much as possible, while retaining the cross-species analysis of cell fate proportions - as per editor's request - that we agree represents a novel aspect of this study.

Some other minor points:

Fig. 4E: The schematic describing how polarization index is calculated is not at all clear. What are the two ends of the arrow pointing to?

We have improved the polarisation index schematic to clarify how the parameter is calculated. The polarisation index is calculated as a ratio between the average fluorescence intensities in the outer cytoplasmic region and inner cytoplasmic region of the cell, with outer and inner defined along the radial axis of the ICM. The modified schematic is included in the main figure as revised Figure 4E, as shown below:

Revised Figure 4E

Figure 4E: Schematic description of polarisation index. Polarisation index is calculated from the measurements in (D) as the ratio between mean aPKC intensity until 1/4th distance from the outer edge and mean aPKC intensity until 1/4th distance from the inner edge of the cell. Boxplots for comparison of the polarisation index in PrE (GATA6-high) versus EPI cells (GATA6-low). GATA6 expression level is categorised as high or low by thresholding the bimodal distribution of GATA6 fluorescence intensity. Colour of the line indicates GATA6-expression level of the cell. n=136 GATA6-high and 124 GATA6-low cells from 32 ICMs. One-way ANOVA, $p=6.03e^{-20}$.

Section on ECM deposition (lines 238-275). Since many of the earlier experiments were conducted on isolated ICMs, are radial ECM gradients observable in these isolated ECMs?

We have performed additional immunostaining to analyse ECM in isolated ICMs at stage E3.75 for the radial gradient. We quantified the laminin intensity in isolated 3x ICMs along the radial axis from the centre to the outer surface, as in Revised Figure 5C,D, which indeed showed the laminin gradient. The data are presented as new Extended Data Figure 5D.

Revised Extended Data Figure 5D

Extended Data Figure 5D: Immunofluorescence images and quantification of total laminin distribution from the centre to the ICM-cavity interface in 3x isolated ICMs at stage E3.75. Solid line and shaded region, mean \pm SD of the laminin fluorescence intensity. Yellow dotted arrow indicates the segment along which fluorescence intensity was measured in the equatorial plane of the ICM. Lines of the same colour correspond to measurements from the same embryo. Data from n=8 embryos.

D. Appropriate use of statistics and treatment of uncertainties

Lines 120-124 and Fig. 2B: The authors conclude that “PrE cells located at the ICM surface were more stretched, in significant contrast to the more rounded EPI cells.” The scatter plots of cell aspect ratio in Fig. 2B do show that the distribution of aspect ratios is slightly skewed to larger values (flatter cells) for PrE cells; however, the authors evaluate significance here with a Mann-Whitney U test, so they should be a little more exact in describing the findings. The null hypothesis of this test is just that comparing a pair of randomly chosen surface PrE and EPI cells will with equal probability find the PrE or EPI cell being flatter (having larger aspect ratio). Rejecting the null hypothesis means that the distribution of PrE cell aspect ratios is different from that of EPI cells. Both cell types have lots of rounded cells, but PrE-cell distribution has a longer tail of flatter cells.

We have accordingly revised the text to clarify the differences in cell shape among EPI and PrE cells.

Fig. 2D: This appears to be a fairly weak correlation of measured surface tension with *Pdgfra*(H2B-GFP) intensity. There should be some quantification offered for this correlation, e.g., a Pearson correlation coefficient or a slope \pm standard error for the best fit line.

We have performed additional micropipette aspiration measurements to increase the statistical power of our analysis. Revised Figure 2D now presents data collected from 40 measurements from 5 independent experiments, strengthening our analysis and conclusions. We observed that surface tension of cells better correlates with logarithm of *Pdgfra*^{H2B-GFP} intensity, with Pearson correlation coefficient $R = -0.71$, $p = 2.8 \times 10^{-7}$, and slope -431.58 ± 69.33 .

Revised Figure 2D

Figure 2D: Micropipette aspiration of E3.5 ICMs expressing *Pdgfra*^{H2B-GFP} (green) and membrane td-tomatato (mT, magenta), and scatter plot of measured surface tension of outer cells versus logarithm of *Pdgfra*^{H2B-GFP} fluorescence intensity of the cell. White arrowhead marks the site of cell aspiration and the white dotted line indicates cell surface contour. n= 40 cells from 24 ICMs. Black dotted line, linear

regression with slope -431.58 ± 69.33 , Pearson's $R = -0.71$, $p = 2.8e^{-07}$. Interfacial tension is calculated using Young-Laplace equation with γ_{cm} , cell-medium interfacial tension, P_c , aspiration pressure, R_p , radius of pipette and R_c , curvature radius of cell surface.

Line 289-291: When the PrE cell-fraction is noted in the main text, it is reported with a very tight constraint as 0.606 ± 0.08 . Is the 0.08 a standard deviation or standard error of the mean? The graphical representation in Fig. 6A seems to show much broader error bars.

We have revised the text to clarify this point in the manuscript and in the figure legend. The 0.606 ± 0.08 is mean \pm standard deviation, in both the text and in the graph.

Extended Data Fig 6B: The data are fit to a linear model, but the data for ICM radius versus size factor appear to be non-linear, with the slope flattening at larger size factors. In accord with expected dimensional scaling (see below), and given that size factor is based on a comparison of cell numbers or ICM volume, one would expect the ICM radius to scale as $(\text{size factor})^{1/3}$ or $(\text{size factor})^{1/2}$ depending on the blastocyst shape. The author's conclusion that "ICM radius scales linearly with embryo size factor in size-manipulated E4.5 blastocysts" is not well supported by this data. The p-value is very low, but that p-value only tells you that the best-fit slope is significantly different from zero, not that a linear model is the most appropriate.

We apologize for this misleading presentation, and have corrected Ext. Data Fig 6B as well as 6C. These linear trend lines were only shown to indicate the increasing or decreasing trend, and were not used anywhere else in our present work. Indeed, a linear relation of ICM base radius with size factor or with the IA/volume ratio is not compatible with fundamental scaling laws. We emphasize that this relation did not form the basis of our analyses in Fig. 7, as we believe the reviewer was understandably led to think. We have now corrected Ext. Data Fig. 6B and 6C by removing the linear regression, and instead present only the scatter plot to show the measurements.

Further, to avoid misunderstanding, revised Figure 7B now highlights the divergence between PrE coverage area (blue shaded region) and the ICM-fluid interface area for a simplified hemispherical shape (black dotted line) with increasing cell numbers.

Revised Extended Data Figure 6B

Extended Data Figure 6B: Scatter plot depicting how ICM radius increases non-linearly with total cell number in the ICM in size-manipulated E4.5 blastocysts.

Revised Extended Data Figure 6C

Extended Data Figure 6C: Scatter plot depicting how ratio of ICM interface area to volume decreases with increasing ICM radius.

Revised Figure 7B

Figure 7B: Scatter plot of $A_{\text{Interface}}$ as a function of total cell number in the ICM for size-manipulated E4.5 mouse embryos, colour of the dots indicates size ratio of the embryos. Monolayer formation is predicted between the minimal and maximal bounds of A_{PrE} based on the fixed fate ratio (blue region). The black dotted line shows the surface area of a hemispherical PrE as a function of the volume corresponding to the respective number of cells. This curve illustrates how nonlinear surface-to-volume scaling permits monolayer formation only within a particular range of embryo sizes for a simplified shape. Inset shows zoom in for x0.25 and x0.375 size ratio. $n = 25$ embryos for 2/8x, 26 for 3/8x, 24 for 4/8x, 29 for 1x, 17 for 2x, 18 for 3x, and 10 embryos for 4x size ratios.

E. Conclusions: robustness, validity, reliability

There is one section of the paper where I have strong doubts about the validity of the conclusions. In “Differential surface tension can sort EPI and PrE cells, but only for normal-size ICM,” the first half of the conclusion is well-supported, but the second half is not. The authors measure the sorting dynamics for normal-size ICMs and tune a cellular Potts model to match those dynamics. They then run that model for larger ICMs and conclude that surface tension alone is not sufficient to sort the larger ICMs within the developmental/physiological window available; however, the physiological window is ~12 hrs, sorting of normal-size embryos took 2.5 hrs and the authors state that “larger embryos took 4-5 hours longer and achieved slightly lower sorting scores.” Assuming the extrapolation of the model is valid, then the timing seems to work out to allow surface-tension-based sorting within 12 hrs, albeit perhaps to slightly less complete sorting. All of this assumes that the model and its non-unique fitted parameters can be extrapolated appropriately. I just do not find this argument convincing.

That said, the argument above is only used to justify looking for additional mechanisms beyond surface-tension-based sorting. The authors do show that there are additional mechanisms (ECM-

gradient-based directed cell migration), so perhaps the model-based justification to look for additional mechanisms just isn't needed in this paper.

We thank the reviewer for their incisive comments, and we agree that extrapolation of the parameters to the large-sized ICMs may not be accurate. Therefore, we have removed the model-based sorting analysis from the manuscript. We instead highlight the outward movement of inner PrE cells from Figure 1H as justification to analyse cell behaviour inside the bulk of the ICM.

The author's conclusions regarding the importance and optimality of PrE/EPI cell proportions seem fairly weak, especially when those species-specific proportions are held up as being key to robustness with respect to natural size variations. It appears that the key to robustness is not the fixed proportions (which actually hinder robustness), but the flexibility in PrE cell shape and how the same number of PrE cells can cover a wide range of ICM surface area by changes in how flat or rounded these cells are.

We appreciate that substantial clarifications are necessary here. We agree that, while flexibility in PrE cell shape indeed contributes to the robust formation of a PrE monolayer, a fixed EPI/PrE ratio limits this robustness. Figure 7B shows the range of possible monolayer areas compatible with the measured fixed fate ratio—the limits of this range increase linearly with the total number of cells (minimal/maximal apical PrE area times number of PrE cells). However, since the surface-to-volume ratio of the ICM scales non-linearly with the number of cells, this mechanism cannot produce a confluent monolayer for very small ICM size (black, red and yellow dots above blue shaded region) or very large ICMs (blue dots below blue shaded region) in size-manipulated blastocysts.

To account for the robustness of PrE monolayer formation across species with differently sized/shaped ICMs, we propose an evolutionary adaptation or coupling between the EPI/PrE fate ratio and the respective surface-to-volume ratio.

We have improved our explanations in text and figures, and the revised Figure 7B now highlights the divergence between PrE coverage area (Figure 7B, blue shaded region) and the increase in ICM-fluid interface area for a simplified ICM shape (Figure 7B, black dotted line). Specifically, for a spherical ICM with a hemispherical $A_{\text{Interface}}$, we estimated the increase in interface area with increasing cell number, illustrating that non-linear surface-to-volume scaling permits PrE monolayer formation only within a particular range of embryo sizes.

Revised Figure 7B

Figure 7B: Scatter plot of $A_{\text{Interface}}$ as a function of total cell number in the ICM for size-manipulated E4.5 mouse embryos, colour of the dots indicates size ratio of the embryos. Monolayer formation is predicted between the minimal and maximal bounds of A_{PrE} based on the fixed fate ratio (blue region). The black dotted line shows the surface area of a hemispherical PrE as a function of the volume corresponding to the respective number of cells. This curve illustrates how nonlinear surface-to-volume scaling permits monolayer formation only within a particular range of embryo sizes for a

simplified shape. Inset shows zoom in for x0.25 and x0.375 size ratio. n = 25 embryos for 2/8x, 26 for 3/8x, 24 for 4/8x, 29 for 1x, 17 for 2x, 18 for 3x, and 10 embryos for 4x size ratios.

F. Suggested improvements: experiments, data for possible revision

Given that there are two distinct sets of key findings and limited space to describe them all, the authors should give careful consideration to focusing this paper on the first set related to cellular and biophysical mechanisms of PrE/EPI cell sorting. The second set of findings on cell-fate specification are not as strong and detract from the paper's ability to fully and clearly explore the first set of findings.

If the parts about cell-fate specification remain, the analysis and discussion of how PrE-monolayer formation is maintained in different sized embryos really should have some discussion of dimensional scaling. The plots in Fig 7B, G and J all compare interface area to number of cells in ICM. Since the latter will be proportional to the volume of the ICM (assuming average cell volume doesn't change), then this is a question of surface-area-to-volume ratio, where one would naturally expect something like a $1/r$ scaling. With the current plots, one would thus expect surface area to scale as $(\text{volume})^{2/3}$ power. In accord with that expectation, there is some flattening of the data trendlines at larger numbers of cells; the trend in Fig. 7B is certainly not linear. It would be very useful to plot the results on a log-log scale to see if the expected power laws hold or not. The scaling may hold within a species, but not among species. It is worth noting that the blastocysts get larger from mouse to monkey to human, but the PrE cell fraction does not follow the same trend (rising from 0.606 to 0.702 from mouse to monkey, but then falling to 0.554 for human). Scaling analysis and arguments could add substantial clarity to this part of the paper.

We appreciate that a better explanation of Figure 7 is needed. The linear relations shown in Fig. 7B, G and J refer to the range of possible areas that can be covered by PrE cells given their capacity to stretch and the respective number of cells at the fixed fate ratio. They are calculated as the range between 10th and 90th percentiles of the apical area distribution for the PrE cells multiplied by the total cell number and the PrE fate ratio. We have not imposed any linear surface-to-volume scaling of ICMs, in fact our point is precisely that a fixed fate ratio must lead to failure of monolayer formation given extreme size perturbations because of the non-linearity of surface-to-volume scaling. Lastly, we do not propose a specific scaling relation in the manuscript because we found the ICM shapes to be quite variable.

It would also help to label panels B, G and J of Fig. 7 with the species from which the data was obtained.

We have revised the Figure 7 B, G, and J to include the species information in the figure panels, as indicated below:

Revised Figure 7B

Figure 7B: Scatter plot of $A_{\text{Interface}}$ as a function of total cell number in the ICM for size-manipulated E4.5 mouse embryos, colour of the dots indicates size ratio of the embryos. Monolayer formation is predicted between the minimal and maximal bounds of A_{PrE} based on the fixed fate ratio (blue region). The black dotted line shows the surface area of a hemispherical PrE as a function of the volume corresponding to the respective number of cells. This curve illustrates how nonlinear surface-to-volume scaling permits monolayer formation only within a particular range of embryo sizes for a simplified shape. Inset shows zoom in for $\times 0.25$ and $\times 0.375$ size ratio. $n = 25$ embryos for $2/8x$, 26 for $3/8x$, 24 for $4/8x$, 29 for $1x$, 17 for $2x$, 18 for $3x$, and 10 embryos for $4x$ size ratios.

Revised Figure 7G

Figure 7G: Scatter plot of $A_{\text{Interface}}$ as a function of total cell number in the ICM for monkey blastocysts 7-8 days post-ICSI. Monolayer formation is predicted between the maximal and minimal bounds of A_{PrE} , for 70% PrE proportion within the ICM (blue region), compared to prediction for 60% PrE proportion within the ICM (orange hatched region).

Revised Figure 7J

Figure 7J: Scatter plot of $A_{\text{Interface}}$ as a function of total cell number in the ICM for human blastocysts at stages late D6 and D7. Monolayer formation is predicted between the maximal and minimal bounds of A_{PrE} , for 55% PrE proportion within the ICM (blue region). $n=15$ embryos.

Lines 397-399: “Despite the fixed proportion of EPI/PrE cells, patterning is robust against naturally variable sizes of the embryo because this proportion is species-specific and optimal for embryo size

and geometry.” This sentence overstates the importance of having a species-specific optimal EPI/PrE proportion for robustness to natural variations in embryo size. The robustness is not really from the fixed proportion, but from the adjustability of PrE cell shape (flattening) at the ICM surface.

We agree with the reviewer on the role of PrE cell shape change in robust patterning. However, as discussed above, this mechanism can only operate within a limited range of size and shape variations. We substantiate this claim with additional analysis of a hemispherical ICM shape.

Therefore, from the data presented in Figure 7D-J, we propose an adaptation of the fate ratio on an evolutionary timescale by doing a comparative analysis of mammalian blastocysts, to account for robust monolayer formation in species with differently sized and shaped ICMs.

The surface tension values used in the cellular Potts model (Table S2) are surprisingly large compared to kT , which suggests that cell movements under this model would be quite deterministic. I can see the value in placing the exact parameter values in a supplemental table, but it would be helpful for the main text to include some mention of these parameters, including their order of magnitude relative to kT and the relative ordering of $J_{EPI:EPI}$, $J_{EPI:PrE}$ and $J_{PrE:PrE}$, which would influence the sorting dynamics.

The temperature and the tension values were adopted from the experimental data. The novel framework of Poissonian cellular Potts models, which is explained in a companion paper (ref. (Belousov et al., 2023), now accepted for publication in PRL), allows for an unambiguous interpretation of these values and has no fictitious parameters such as the fluctuation allowance (effective temperature) in the traditional cellular Potts models. Yet, as we show in the companion paper [Fig. 4(b) and (c)], these large tension values are not sufficient to drive the sorting in a deterministic fashion and the system may actually get stuck in a metastable state. Therefore, other sources of noise are necessary and present in our model due to the active growth and division. Because the Poissonian framework does not rely on Metropolis sampling with the Boltzmann factor, the temperature is not the only source of fluctuations. Two alternatives are discussed in the companion paper.

G. References: appropriate credit to previous work?

The references appear to give appropriate credit to previous work.

H. Clarity and context: lucidity of abstract/summary, appropriateness of abstract, introduction and conclusions.

The paper’s Introduction does not contain a clear thesis statement about the paper’s important findings. Instead, it says that “In this study, we systematically analyse cellular dynamics, position, fate, and polarity using reduced systems and blastocyst manipulation to gain mechanistic insights into ICM patterning robustness during mouse pre-implantation development.” This statement makes the paper seem more descriptive than it actually is once you get into the meat of the results. The abstract does a much better job summarizing the key findings.

We have revised the text to edit the last paragraph of the Introduction to present the thesis of this study more clearly.

Reviewer #3

In this paper, Hiiragi and colleagues investigated the segregation of PrE cells to the outside of the ICM in mouse embryos, supplemented with additional analysis in monkey and human blastocysts to ensure robustness of the results. The authors perform a number of analyses, mostly with genetic/small molecule perturbations followed by imaging and advanced quantitative analysis, and some physical modeling, to make several claims about PrE segregation. The main claims are that the segregation is due to Rac1/aPKC – dependent directed migration towards the outside of the ICM, and that they are stuck there due to a decreased tension. The authors also show a gradient of laminin secreted by the PrE cells and claim that the PrE cells are using this as a pathway to crawl out of the ICM using the aforementioned mechanisms. The authors claim this is the first paper to show a coupling between the acquisition of PrE fate and spatial segregation, though this is not really the case. Indeed, there are very few claims in this paper that are genuinely novel and it's difficult to see a major conceptual advance. What is true is that this paper does a better job than its predecessors to quantify dynamics within embryos, and though that is to be commended, I am not sure that fact is enough to carry this paper for consideration for Nature Cell Biology. Following are a few comments on the claims.

We feature a few novel concepts in our paper, which are appreciated by Reviewer #1 and Reviewer #2: Rac1-dependent migration and directed protrusions in PrE cells, the role of extracellular matrix proteins in guiding PrE migration, and the optimal coupling of the fixed cell fate ratio in the ICM with embryo size and shape.

The only sentence where we used “first” for our findings is in Discussion “...we report for the first time, that PrE cells undergo Rac1-dependent active migration.” We are not aware of any earlier study that showed this, and would appreciate it if this reviewer specifies it, so that we could cite it here.

1. (Fig. 1) The *ex vivo* segregation of blastocysts has been shown numerous times, including cited papers such as Wigger et al, and the Plusa et al paper led a number of papers that have shown high-fidelity sorting of PrE from the ICM.

While previous cited works have indeed shown *ex vivo* segregation of ICMs, comprehensive tracking of the ICM cells in 3D and its quantitative analysis, as presented in Figure 1, has never been reported. This allowed us to dissect and reveal the underlying mechanisms that are presented in the following Results.

2. (Fig. 2) Differential surface tension as a means of either sorting cells or keeping them on the outside of an aggregate has been shown numerous times across many different organisms, and AFM has been performed on EPI and PrE to show differences in surface mechanics previously, and active myosin stains have been shown before too. The current paper uses a better technique to show differences in tension using micropipette aspiration but overall differential surface tension is not a new concept. The correlation shown in Fig. 2D is also unconvincing.

We have performed additional micropipette aspiration measurements to increase the statistical power of our analysis. Revised Figure 2D now presents data collected from 40 measurements from 5 independent experiments, strengthening our analysis and conclusions. We observed that surface tension of cells better correlates with logarithm of *Pdgfr α ^{H2B-GFP}* intensity, with Pearson correlation $R = -0.71$, $p = 2.8e^{-07}$, and slope -431.58 ± 69.33 .

Note that AFM measurements and active myosin staining were previously conducted on dissociated ICM cells in (Yanagida et al., 2022), both of which actually failed to detect differences between EPI and PrE cells. Dissociated ICM cells may not fully recapitulate the physiological condition, as they are fragile due to the enzymatic dissociation treatment, and do not survive *in vitro* until E4.5 stage. For this reason, in our study we used isolated ICMs, which faithfully recapitulate the sorting process.

Revised Figure 2D

D

Figure 2D: Micropipette aspiration of E3.5 ICMs expressing *Pdgfra*^{H2B-GFP} (green) and membrane td-tomato (mT, magenta), and scatter plot of measured surface tension of outer cells versus logarithm of *Pdgfra*^{H2B-GFP} fluorescence intensity of the cell. White arrowhead marks the site of cell aspiration and the white dotted line indicates cell surface contour. n= 40 cells from 24 ICMs. Black dotted line, linear regression with slope -431.58 ± 69.33 , Pearson's R= -0.71 , and $p=2.8e^{-07}$. Interfacial tension is calculated using Young-Laplace equation with γ_{cm} , cell-medium interfacial tension, P_c , aspiration pressure, R_p , radius of pipette and R_c , curvature radius of cell surface.

3. (Fig. 3) The protrusion analysis has been shown at least in part by Meilhac et al (also cited in the paper) and migration has been proposed before in that paper and others. Again this paper does a better job of quantifying it, but it's not a new concept. And the shape analysis was performed in Yanagida et al – indeed this paper doesn't preclude the possibility this is enhanced fluidity of PrE as claimed in that paper. Much of what is shown in this paper could suggest a tension gradient through the whole ICM, which together with a change in cell mechanics (fluidity or otherwise) would explain why the extensions of the cell would go outward. Indeed, one might expect due to geometry that there is a stress gradient in an aggregate, particularly if the cells on the inside are more adhesive. The effects of Latrunculin are way too messy, and Rac1 affects many things (including surface tension itself, and it negatively regulates RhoA activity, among a great many things) to make this convincing.

Our findings of directed protrusions formed in PrE cells within the ICM are most likely distinct from the EPI/PrE sorting mechanism proposed by Yanagida et al., (2022), because a) the enhanced fluidity of PrE cells, shown therein, would generate surface fluctuations in all directions without directionality, and b) their sorting model overlooks the consideration of boundary conditions, particularly the presence of two distinct interfaces around the ICM—a cellular interface on the polar trophectoderm (TE) side and a fluid interface on the blastocyst cavity side. Therefore, their model could only explain segregation of isolated ICMs and cellular aggregates, but not the whole blastocyst. In contrast, our model takes into account the asymmetric boundary conditions of the ICM and the directed migration of PrE cells, to offer a more complete explanation of how the PrE cells specifically move towards a fluid interface as opposed to the polar TE side.

Furthermore, although the earlier studies did not show how the enhanced fluidity arises only in PrE cells, we now demonstrate by additional experiments that the apical polarisation drives PrE cell protrusion as well as migration, as presented to address the next comment.

4. (Fig. 4) The potentially most novel aspect of this paper is connection to aPKC and polarity, but the analysis isn't that thorough. Each of the perturbations can affect many things, and of course they do see smaller ICMs with the genetic perturbations. Overall, this isn't a convincing contribution to what's known, particularly in light of the work of Clare Chazaud and lab on the importance of polarity in PrE. Moreover, aPKC and polarity does not explain changes in mechanics in itself. What mechanical changes are associated? The authors may claim it's to facilitate migration but this reviewer is not particularly convinced by that data.

We appreciate the reviewer's interest in the connection between cell fate segregation and polarity.

We have performed an additional experiment and analysis to provide a functional link between apical polarisation and cell migratory activity, by perturbing aPKC function with Gö6983 in the blastocysts. Specifically, we generated large, mosaic blastocysts using fluorescent reporters marking cell fate with *Pdgfra*^{H2B-GFP}, and membrane with mTmG (as in Figure 3A, B), performed live-imaging of these

blastocysts under aPKC inhibition with Gö6983, and evaluated the migratory activity of PrE cells by their movement and changes in cell shape.

First, measurement of the distance of PrE cells from the fluid cavity surface showed that treatment with Gö6983 disrupted the directed movement of PrE cells towards the fluid interface (revised Extended Data Figure 4E) in contrast to Figure 3C (see below). Second, PrE cells in the blastocysts clearly lack membrane protrusions upon inhibition of aPKC. As a result, PrE cell shape is more rounded, and not significantly different from EPI cells in the ICM (Extended Data Figure 4F). Together, these data indicate that acquisition of apical polarity triggers the formation of membrane protrusions in PrE cells and their migration towards the fluid cavity surface. We present these new data in the manuscript as new panels, Extended Data Figure 4E,F.

Note that genetic KO of two isoforms of aPKC showed an increase in acto-myosin levels at the apical domain in the 8-cell embryo, suggesting that aPKC antagonizes cortical contractility in the apical domain (Maître et al., 2016, cited in this study). In our revised manuscript, we elaborated this reasoning so that it is clear that we experimentally demonstrated the mechanical role of aPKC and the apical domain in EPI/PrE sorting at the ICM surface.

Revised Extended Data Figure 4E:

Extended Data Figure 4E: Single-cell tracking and analysis of distance of fluorescence-labelled EPI and PrE cells from the cavity surface in chimeric blastocysts treated with the aPKC inhibitor Gö6983. Distance is measured from the centre of nucleus/cell to the cavity interface. Individual cell position curves were smoothed using a rolling average. Grey dotted lines mark the average position of ICM-TE interface, and ICM-cavity interface set at $d=0$. $n=7$ EPI cells, 19 PrE cells from 7 embryos.

Figure 3C

Figure 3C: Single-cell tracking and analysis of distance of fluorescence-labelled EPI and PrE cells from the cavity surface. Distance is measured from the centre of nucleus/cell to the cavity interface, which was visualised in brightfield. Individual cell position curves were smoothed using a rolling average. Grey dotted lines mark the average position of ICM-TE interface and the ICM-cavity interface. $n=14$ EPI cells, 31 PrE cells from 13 embryos.

Revised Extended Data Figure 4F:

Extended Data Figure 4F: Representative images of EPI (top) and PrE (bottom) cell shapes and cell shape quantification in mosaic-labelled E3.75 blastocysts upon inhibition of aPKC with Gö6983. White asterisks, EPI cells of interest. White arrowheads, PrE cells. Mann-Whitney U-test, $p=0.6$, $n=48$, 80 measurements for EPI and PrE cells, respectively.

5. (Fig. 5) The differences in integrins and ECM have been noted by these same authors previously in a very nice paper (E.J.Y. Kim et al, 2022). However, ECM and integrin sensing of it will also affect cell fate, and there is very little evidence of the fact that the ECM is forming the sort of scaffolding that would act as a ‘highway’ for cells, instead of acting to promote a different kind of cell-cell adhesion or signalling hub for PrE. Indeed, ECM-integrin sensing is very difficult to prove as a mechanism for active migration in jammed tissue, particularly since it plays such a profound role in mitogenic signalling such as FGF signalling. Unfortunately, this part is not convincing and I do not see how it could become convincing, because even disrupting the ECM would cause many effects.

We agree with the reviewer that it is technically challenging to test the specific role of the ECM in fate segregation since, for example, chemically disrupting the ECM throughout the blastocyst would have many effects. Therefore, we have performed two additional experiments to manipulate the ECM distribution locally and test the functional role – sufficiency – of the ECM gradient in PrE cell sorting, in addition to its requirement our earlier study showed (Kim et al. 2022).

First, we experimentally generated ectopic ECM accumulation in the ICM by implanting polymethyl methacrylate (PMMA) microbeads coated with laminin (see new Extended Data Figure 5F, copied below, for validation) into the ICM of the blastocyst, and evaluated the distribution of PrE cells at stage E4.5. Our model predicts that some PrE cells would be misguided by the ectopic laminin localisation, and we tested this with experiments. In control embryos with E-cadherin-coated microbeads, the final positioning of EPI and PrE cells is not perturbed (see below, Revised Figure 5E, top row). In contrast, when microbeads coated with E-cadherin and laminin are implanted into the ICM of the blastocyst, ectopic PrE cells localise to the coated microbead in E4.5 blastocysts (Revised Figure 5E, bottom row). We quantify the ectopic PrE cells localised at the coated microbead in the ICM but not in contact with the fluid-cavity, and compare them across the two groups. These data are in agreement with the prediction and demonstrate that laminin deposition is functionally sufficient for guiding the PrE cell migration. These data are presented as a new panel in Figure 5.

Revised Figure 5E

Figure 5E: Schematic for the experimental strategy to introduce ectopic laminin localisation in the ICM of the blastocyst using coated microbeads. Brightfield and immunofluorescence images of 2x E4.5 blastocysts with implanted beads coated with E-cadherin or E-cadherin+Laminin. Yellow asterisks and yellow dashed circles indicate microbead position in the ICM. Quantification of ectopic PrE cells localised at the coated beads in 2x E4.5 blastocysts. Mann-Whitney U-test, $p=0.0319$, $n=4,3$ embryos with successfully integrated E-cadherin coated beads and E-cadherin+Laminin coated beads, respectively.

Extended Data Figure 5F

Extended Data Figure 5F: Representative immunofluorescence and brightfield images of control and coated PMMA microbeads incubated with E-cadherin-Fc chimeric protein (left) and laminin protein (right).

Second, we performed experiments to rescue the incorrectly patterned phenotype of *Lamc1*^{-/-} previously reported in (Kim et al., 2022) by generating chimeric embryos that include several wild-type (WT) cells in the ICM. If the ECM plays a functional role in PrE cell migration, the resulting laminin deposition by these WT cells should be sufficient to guide *Lamc1*^{-/-} PrE cells to form a monolayer at the fluid interface. To test this prediction, we generated chimeric embryos in which the ICM comprises of both WT and *Lamc1*^{-/-} cells by aggregating blastomeres at the 4-cell stage (Revised Figure 5F). While PrE cells in late-stage *Lamc1*^{-/-} blastocysts tend to clump together (Kim et al., 2022 and revised Figure 5F), those in the chimeric embryos successfully form a segregated monolayer at the fluid interface (Revised Figure 5F), supporting the functional role of laminin in guiding PrE cell migration. These data are included as revised Figure 5F.

Revised Figure 5F

Figure 5F: Schematic diagram for the experimental strategy to rescue the incorrectly patterned phenotype of *Lamc1*^{-/-} blastocysts. Representative immunofluorescence images of late-stage blastocysts from *Lamc1*^{-/-} (left), and chimeric blastocysts between comprising *Lamc1*^{-/-} + WT cells (right). Bottom, zoomed-in images of the ICM. White arrowheads indicate *Lamc1*^{-/-} cells that are successfully segregated to the PrE monolayer at the fluid interface in the presence of WT cells in the ICM. Quantification of ectopic PrE cells in *Lamc1*^{-/-} blastocysts and *Lamc1*^{-/-} + WT chimeric blastocysts, n=8,5 embryos for the two groups, respectively. One-way ANOVA, $p=8.88e^{-03}$.

6. The remainder of the paper about size and robustness is potentially interesting, but doesn't hold up in light of the rest of the paper.

We agree with the reviewer that the last part of this study on the optimality for patterning robustness is interesting and proposes a new mechanism potentially operating on the evolutionary timescale. This became evident to us, owing to the quantitative analyses we performed in the earlier parts of the study. With our comparative analysis of mouse, monkey, and human blastocysts, we propose an evolutionary adaptation between cell fate proportion in the ICM and embryo size and shape. We substantially revised the analysis and description for better clarification of this part.

I do believe this paper has something to offer it and represents an incremental advance in what is known on this difficult and important topic. I simply do not see the conceptual advance the authors appear to be claiming. I am also not convinced that doing a significant number of experiments to address the above concerns would overcome the sense that the paper is not adding a great deal to what is already known.

We feature a few novel concepts in our paper, which are appreciated by Reviewer #1 and Reviewer #2: directed protrusions of PrE cells guided by the deposition of extracellular matrix proteins, and the optimal coupling of the fixed cell fate ratio in the ICM with embryo size and shape. From Reviewer #3, whose report seems to be focused on integration of the prior research into our study, these major advancement in concepts did not receive substantial comments to act upon.

References

- Belousov, R., Savino, S., Moghe, P., Hiiragi, T., Rondoni, L., & Erzberger, A. (2023). *Poissonian cellular Potts models reveal nonequilibrium kinetics of cell sorting* (arXiv:2306.04443). arXiv. <http://arxiv.org/abs/2306.04443>
- Chan, C. J., Costanzo, M., Ruiz-Herrero, T., Mönke, G., Petrie, R. J., Bergert, M., Diz-Muñoz, A., Mahadevan, L., & Hiiragi, T. (2019). Hydraulic control of mammalian embryo size and cell fate. *Nature*, *571*(7763), 112–116. <https://doi.org/10.1038/s41586-019-1309-x>
- Iitzkovitz, S., Blat, I. C., Jacks, T., Clevers, H., & van Oudenaarden, A. (2012). Optimality in the Development of Intestinal Crypts. *Cell*, *148*(3), 608–619. <https://doi.org/10.1016/j.cell.2011.12.025>
- Kim, E. J. Y., Sorokin, L., & Hiiragi, T. (2022). ECM-integrin signalling instructs cellular position sensing to pattern the early mouse embryo. *Development (Cambridge)*, *149*(1). <https://doi.org/10.1242/dev.200140>
- Korotkevich, E., Niwayama, R., Courtois, A., Friese, S., Berger, N., Buchholz, F., & Hiiragi, T. (2017). The Apical Domain Is Required and Sufficient for the First Lineage Segregation in the Mouse Embryo. *Developmental Cell*, *40*(3), 235-247.e7. <https://doi.org/10.1016/j.devcel.2017.01.006>
- Maître, J. L., Turlier, H., Illukkumbura, R., Eismann, B., Niwayama, R., Nédélec, F., & Hiiragi, T. (2016). Asymmetric division of contractile domains couples cell positioning and fate specification. *Nature*, *536*(7616), 344–348. <https://doi.org/10.1038/nature18958>
- Plusa, B., Piliszek, A., Frankenberg, S., Artus, J., & Hadjantonakis, A. K. (2008). Distinct sequential cell behaviours direct primitive endoderm formation in the mouse blastocyst. *Development*, *135*(18), 3081–3091. <https://doi.org/10.1242/dev.021519>
- Yanagida, A., Corujo-Simon, E., Revell, C. K., Sahu, P., Stirparo, G. G., Aspalter, I. M., Winkel, A. K., Peters, R., De Belly, H., Cassani, D. A. D., Achouri, S., Blumenfeld, R., Franze, K., Hannezo, E., Paluch, E. K., Nichols, J., & Chalut, K. J. (2022). Cell surface fluctuations regulate early embryonic lineage sorting. *Cell*, *185*(5), 777-793.e20. <https://doi.org/10.1016/j.cell.2022.01.022>

Responses to Editorial Comments

A) Experimentally assess the causative links between ECM polarity and PrE cell migration as outlined by Reviewers #3 and #4 (and see experimental suggestion by Reviewer #4).

We have performed an additional experiment and analysis of immunofluorescence images to evaluate the correlation between asymmetry in laminin distribution around PrE cells and the location of membrane protrusions.

B) All other referee concerns pertaining to strengthening existing data, providing controls, methodological details, clarifications and textual changes, should also be addressed.

We have addressed all other remaining comments from the Reviewers regarding clarifications and textual changes, as mentioned in detail below.

Specific responses to Reviewers' Comments

Reviewer comments in blue

Author responses in black

Reviewer #1

Remarks to the Author:

The authors have nicely addressed the point raised by providing additional supporting data.

Reviewer #2:

Remarks to the Author:

The revised paper is substantially improved and the authors have largely addressed my concerns. I do have three minor and easily correctable concerns.

1. I suspect this may be a typo. In the main text on Lines 332-333, the authors state that the proportion of PrE cells isolated from blastocyst and ICMs is 0.606 ± 0.08 for $n = 106$ embryos. The authors also point to Fig 6A, but the caption to that figure says that the mean proportion of PrE cells is 0.605 ± 0.78 for a set of 4 groups with $n = 19, 29, 21$ and 32 embryos (which adds up to 101 embryos). The authors should clarify and/or fix the discrepancy.

We apologise for this discrepancy, indeed these were typos. The mean PrE proportion and sample numbers are 0.605 ± 0.078 (mean \pm SD, $n=101$ embryos). The typos have been corrected in the main text and are now consistent with the figure legend.

2. In the caption to Fig 2D, the slope of the linear regression line is given as -431.58 ± 69.33 . Given the size of the standard error, both values are presented with more than the necessary significant figures. The slope should also have units.

We now present the slope value as -432 ± 69 pN/ μ m in the figure panel and legend.

3. The values presented in Table S3 also have more than the necessary significant figures.

We have updated the values in Table S3 to have the appropriate number of significant figures.

Reviewer #3:

Remarks to the Author:

I respectfully disagree with the authors that my concerns can be mostly reduced to a focus 'on integration of prior research into our study'. This was not my primary concern or even a major concern. In laying out my review, I was attempting to make clear that a lot of what the paper was putting forward was already more-or-less covered in the literature. As I said, and the authors reiterated in their response, it is better quantified and characterised than before and the authors really deserve a lot of credit for this. I do believe the authors had already referenced key papers and they have done a better job of discussing them now, and it is more than sufficient.

Better integration into the literature simply was not my main concern. My main point was that, given that a decent amount of the paper content represents a better characterisation of things that are already known, the novelty hinges on: 1. the idea that PrE cells undergo active, directed migration via ECM gradients; and 2. The data presented in Fig. 7 showing that EPI/PrE proportions are optimal for the size and geometry of the ICM. Fig. 7 is a very interesting result and they have expanded on it with new species. I personally do not believe there is enough there in Fig. 7 in terms of insight and mechanism to carry the whole paper, but if the editor and other reviewers believe it is, I understand that completely.

However, I think almost anyone reading this paper would strongly believe that for this paper to be fully convincing as a major advance, it is essential that the authors have sufficiently shown that PrE is undergoing directed, active migration towards the cavity. My main concern before, which remains after the revisions, is that the idea of active, directed migration of PrE along ECM gradients is not convincing, and I do not see what would make it convincing.

1. The protrusion angle data (Fig. 3F) is relatively unconvincing – it certainly does not look like that alone could convince anyone that PrE cells are migrating to the surface robustly and that this is the main reason they get there. The particular geometry (with more space towards the cavity) and boundary conditions of the ICM would suggest there should be some bias towards the cavity, and it is hard to imagine their data is reflecting anything more than that. In fact, if there was a highway of ECM leading these cells out, I would expect a much stronger bias.
2. All of the perturbations – aPKC, Rac1, latrunculin, CK666 – have many feedbacks with cell surface mechanics and are too confounded to make the case by themselves or collectively. This is always a problem and I do not really know how it could be resolved. It could be good supporting data if there were otherwise strong data supporting their claims. The aPKC polarisation index shown in Fig. 4 is slightly more convincing, but given that polarity is tied up with cell surface mechanics, and there has never been any doubt that EPI cells stick together well and would act as an adhesion barrier for PrE cells, it is very difficult to interpret what that polarity might mean and if it really can be a proxy for migration bias. If EPI cells have more E-cadherin or other surface factor, this could easily explain how the PrE cells might polarise in this manner, but again it does not mean they are actively migrating, much less along an ECM gradient.
3. PrE cells express different integrins than EPI cells and they are more likely to adhere to ECM like laminin, of that there is no doubt. But laminin-integrin binding initiates signalling that is essential for PrE cells, like ERK. I am still unconvinced, even with the new data, that a secreted laminin gradient is causal. As another reviewer pointed out, it could be just a consequence of the PrE cells having an outside bias. The new bead data is not convincing, the PrE cells could just get stuck on them – unless I am missing something there is no evidence they are causing PrE cells to migrate to them. And the rescue they showed is likely to change signalling in the ICM by now having cells that secrete laminin that will likely make the initiation of the PrE identity more robust. The authors did not rule out that very likely possibility. In sum, ECM is extremely important for signalling. It is notoriously difficult to dissect the role of ECM in mechanics/migration etc and its role in intracellular signalling. I do not see evidence here that challenges what should be the null hypothesis: that ECM is integral to the robustness of the mechanical phenotype and identity of PrE cells and it does not necessarily have anything to do with migration.

This may sound like nitpicking. The problem as I see it (again) is not that the authors have not paid proper due to previous literature. It is that previous literature has proposed different sorting

mechanisms that revolve around the idea that the cells have different mechanical properties and, given the geometry and the constant perturbations in the ICM like cell divisions and blastocyst contractions, are likely to be sufficient to explain sorting. If the authors want to propose a different mechanism than what has been proposed, the burden of proof is high. In the end, I do not have any reason to believe these cells would actively migrate along an ECM gradient. It seems implausible to me. I believe the ECM gradient is incidental and its role in instructive signalling may indeed be very important. However, I am absolutely open to someone turning my preconceived notions around: I would be delighted for the authors to prove me wrong and I would accept it wholeheartedly if they do. The burden of proof, however, is on the authors. As I said in previous review, this is a very, very tricky problem to solve. I do not have specific recommendations to solve the pleiotropy/confounding problem with ECM particularly in such a size and geometry.

There is, however, very good and well-characterised data in this paper and it has much to offer the field, and the size robustness data is interesting. If the authors used extensive mitigating language around their directed migration data and put it forward as a hypothesis more than a well-supported finding, I would support publication. I just do not know if the journal would be happy with that outcome.

We appreciate the reviewer's consideration and have carefully modified our presentation of the role of ECM in guiding PrE cell migration. According to Reviewer 4's suggestion, we have now performed additional experiments that show that the bias in the ECM deposition correlates with the direction in which migratory PrE cells extend protrusions. Specifically, sites of membrane protrusions in PrE cells, detected as regions of high local membrane curvature along the cell contour, overlap with the distribution of secreted laminin at the onset of migratory activity (Extended Data Fig. 5G). In contrast to PrE cells at stage E3.5, the distribution of cell membrane regions enriched for laminin and those with high curvature are both oriented towards the fluid cavity at stage E3.75 (Extended Data Fig. 5H). These findings are consistent with the hypothesis that graded laminin distribution could potentially guide PrE cell migration.

Extended Data Figure 5G

Ext. Data Fig. 5G: Representative immunofluorescence images of non-migratory and migratory PrE cells in 3x blastocysts at stages E3.5 and E3.75, respectively. Accompanying line profiles show the local membrane curvature (blue) and laminin distribution (magenta) along the PrE cell boundary. Yellow dotted lines, cell contour for measurement of laminin distribution and local membrane curvature, traced starting at the yellow circle clockwise until the flat arrowhead. Line profiles were smoothed using a rolling average. Scale bars, 20 μ m.

Extended Data Figure 5H

Ext. Data Fig. 5H: Polar histograms depicting orientation of PrE cell membrane regions with highest average curvature (blue) and highest average laminin distribution (magenta) along the cell contour from (G) at stages E3.5 and E3.75, respectively. The cell contour was binned into 6 length intervals and the curvature and laminin intensity were averaged for each bin. n=19, 17 cells for E3.5 and E3.75 stages, respectively.

Further, we agree that, alternatively or in addition, there could be signalling interactions between PrE cells and ECMs which could stabilise cell fate. Accordingly, we have modified the text to discuss the signalling role of cell-ECM interactions in regulating PrE cell fate in the blastocyst, and do not claim an exclusively 'mechanical' role for the PrE-ECM interaction for patterning.

Reviewer #4:

Remarks to the Author:

This is a technically strong manuscript that uses beautiful imaging to quantify and visualize the spatial segregation of ICM cells. The theory is very elegant and the optimality implicated in the theory-driven approaches in figure 7 are intriguing but also slightly disconnected from the rest of the manuscript.

A panel of previous work also from the authors themselves have identified cellular mechanisms that contribute to this process. The main new findings of the manuscript, also according to the authors own statements, are the Rac1-mediated polarized migration and migration along a self-generated laminin gradient. While the polarization is convincingly demonstrated, its role in directed migration is very well established across cell biological systems. The migration along a ECM gradient is an exciting new observation but also much harder to demonstrate, and remains somewhat speculative despite the authors efforts. Migrating cells will typically deposit ECM as they go, and therefore the cause and consequence relationship is difficult to establish. The fact that ectopic laminin can promote directed migration it does not provide evidence that this is the mechanism that the cells in fact utilize.

To conclusively demonstrate the causative effect of the ECM, I would recommend that the authors make use of available laminin-reporter lines and perform live imaging experiments to demonstrate that laminin deposition is an early cellular event that occurs at the onset of cell motility, rather than a process that occurs after cells have already translocated their cell bodies along the trajectory.

We appreciate the reviewer's interest in our study and thank the reviewer for the suggestion to dissect the causative effect of the ECM. Live-imaging of laminin, however, would only show the temporal sequence of the relevant processes (laminin deposition vs. cell migration) and we are not sure if this would show the causality more convincingly than the functional perturbation experiments that we have performed at the first revision and that Reviewer #1 and #2 are convinced with. Furthermore, it is uncertain if the "available laminin reporter line" actually has the expression high/clear enough in the blastocyst to allow for the simultaneous imaging with cell fate and membrane markers, let alone the challenge of importing the mice and performing the experiment within the duration of the final revision.

Instead, we have performed an equivalent experiment using immunofluorescence - to examine whether the asymmetry in laminin deposition surrounding PrE cells correlates with that in PrE cell protrusions indicative of cellular migratory activity at the onset of PrE cell migration. These data are now presented as new panels in Extended Data Figure 5, as detailed below.

First, immunofluorescence images of PrE cells at stage E3.75 show that inner PrE cells extend protrusions towards laminin around preceding PrE cells at the cavity surface in 3x blastocysts (Extended Data Fig. 5C), in agreement with the model in which retention of the PrE cells near the cavity surface breaks symmetry in ECM distribution.

Extended Data Figure 5C

Ext. Data Fig. 5C: Representative immunofluorescence images of an inner PrE cell extending protrusions towards laminin around a preceding PrE cell at the onset of migratory activity. Yellow arrowhead marks the site of protrusion, and yellow dotted line indicates cell boundary. Scale bars, 20 μ m.

Further, sites of membrane protrusions in PrE cells, detected as regions of high local membrane curvature along the cell contour, overlap with the distribution of secreted laminin at the onset of migratory activity (Extended Data Fig. 5G). In contrast to PrE cells at stage E3.5, the distribution of cell membrane regions enriched for laminin and those with high curvature are both oriented towards the fluid cavity at stage E3.75 (Extended Data Fig. 5H). These findings are consistent with the hypothesis that graded laminin distribution could potentially guide PrE cell migration.

Extended Data Figure 5G

Ext. Data Fig. 5G: Representative immunofluorescence images of non-migratory and migratory PrE cells in 3x blastocysts at stages E3.5 and E3.75, respectively. Accompanying line profiles show the local membrane curvature (blue) and laminin distribution (magenta) along the PrE cell boundary. Yellow dotted lines, cell contour for measurement of laminin distribution and local membrane curvature, traced starting at the yellow circle clockwise until the flat arrowhead. Line profiles were smoothed using a rolling average. Scale bars, 20µm.

Extended Data Figure 5H

Ext. Data Fig. 5H: Polar histograms depicting orientation of PrE cell membrane regions with highest average curvature (blue) and highest average laminin distribution (magenta) along the cell contour from (G) at stages E3.5 and E3.75, respectively. The cell contour was binned into 6 length intervals and the curvature and laminin intensity were averaged for each bin. n=19, 17 cells for E3.5 and E3.75 stages, respectively.

Reviewer #4:

Remarks to the Author:

I appreciate that the authors have attempted to strengthen the link between matrix deposition and protrusions using immunostainings of fixed tissues. I understand that the live imaging approaches are challenging although I do somewhat disagree with the authors assertion that live imaging would have provided the same level of proof as this chosen approach, as one could have been able to obtain stronger temporal correlations and statistical probabilities. Also, from the images provided it is difficult to see clear protrusions so the question of causality remains somewhat unresolved. I would recommend that the authors tone down the conclusions of the definitive role of "matrix gradients" in the symmetry breaking.

We appreciate the reviewer's consideration, and have toned down conclusions regarding a definitive role of matrix gradients in symmetry breaking.